# FloatSOM: GPU Accelerated, Distributed, Topology-Flexible Self-Organizing Maps

## Abstract

GPU-accelerated Self-Organizing Map (SOM) implementations are among the most competitive options for large-scale SOM analysis, but growing dataset sizes increasingly challenge their practical use because workloads no longer fit cleanly within device-memory limits. We introduce FloatSOM, a SOM framework for scalable training and deployment that supports multi-GPU execution, out-of-memory disk-backed streaming, and scalable training-time graph topologies beyond regular lattices. We evaluate FloatSOM on 14 synthetic and real benchmark datasets together with controlled speed scaling benchmarks, and show that these graph-based topologies, combined with topology-aware hyperparameter fine-tuning, yield lower quantization error than current state-of-the-art SOM baselines. FloatSOM also sustains this performance at large scale with high-throughput distributed execution; in the largest benchmark, it trains a 1024-node SOM network on 1,000,000,000 samples with 50 features in 6.16 minutes on 8 GPUs across two separate high-performance-computing nodes.

## 1 Introduction

Self-Organizing Map (SOM), originally introduced by Kohonen (Kohonen, 1990), is an unsupervised machine-learning method that uses competitive learning to organize nodes such that they capture the topology of the data. In practice, this topology-preserving representation means SOMs are commonly used to map dataset topology, produce dimensionality-reduced visualizations, and conduct clustering at scale (Kangas et al., 1990). As dataset size and heterogeneity increase, the computational requirements of SOMs grow in both time and memory. Recent tools such as aweSOM have improved single-node serial-online SOM execution through CPU/GPU acceleration and ensemble stacking (Ha et al., 2025). However, many current implementations remain constrained to single-device workloads that must fit within video random-access memory (VRAM, GPU memory), with limited support for distributed compute, out-of-core execution, and modern GPU orchestration.

SOM topology presents a second limitation. Classical SOMs are traditionally trained on regular rectangular or hexagonal lattices because fixed grids make neighborhood definition, visualization, and optimization straightforward. However, these regular lattices also impose a strong geometric prior on the learned representation. Accordingly, prior work on dynamic, growing, and graph-structured SOM variants reflects a long-standing recognition that fixed lattices are not always the best match for irregular data geometry (Alahakoon et al., 2000; Vasighi & Amini, 2017; Kangas et al., 1990). However, these alternatives have generally not been developed or evaluated in the high-throughput, large-sample regime targeted by modern GPU-enabled applications and are broadly impractical for deployment at scale.

FloatSOM is designed to address these combined systems and topology limitations. Within FloatSOM, we implement distributed multi-GPU execution, out-of-memory disk-backed streaming, and scalable topology-flexible training beyond standard fixed lattices. Specifically, we implement highly scalable minimum-spanning-tree (MST) and relative-neighborhood-graph (RNG) topologies that define the SOM neighborhood during training rather than being overlaid on a completed map. We also evaluate multiple sampling strategies as a route to further computational acceleration and derive fine-tuned hyperparameter configurations across

diverse datasets and recommended operating regimes. FloatSOM is suitable for both high-performance-computing (HPC) operation and consumer-grade desktop GPUs.

## 2 Related Work

Related work on practical SOM deployment spans software implementations, sampling-efficient training, topology design, and model selection.

### 2.1 Open-Source SOM Implementations and Systems

Open-source SOM libraries range from lightweight implementations to systems-oriented packages. Mini-Som implements classical serial-online training (Vettigli, 2018), while aweSOM adds CPU/GPU acceleration and ensemble stacking for large single-node workloads (Ha et al., 2025). XPySOM instead provides GPU-accelerated batch training and is the closest executable comparator to FloatSOM's Python/GPU training pathway (Mancini et al., 2020).

Somoclu and GigaSOM provide additional parallel-systems context (Wittek et al., 2017; Kratochvíl et al., 2020) and are further elaborated on in Section 8.

### 2.2 Sampling Methodologies for SOM Training

Classical online and batch SOM training schemes traditionally sample every data-point in every training iteration (Kohonen, 2001; Liu et al., 2023). Consequently, one obvious route to improving SOM speed and scalability is subsampling, which reduces the amount of data used to train the final SOM.

To date, numerous subsampling strategies have been proposed. Beyond naive random sampling, guided subsampling methods include hierarchical dynamic subset selection SOM (HDSSSOM), which concentrates computation on difficult or stale regions of the data (Wetmore et al., 2005), and adaptive sampling strategies for design-space exploration (Ito et al., 2016). However, systematically benchmarked and openly maintained implementations that compare full-data, random, and guided sampling within a modern high-performance SOM workflow remain limited.

### 2.3 SOM Lattice and Adaptive Topologies

Most practical SOM implementations retain regular rectangular or hexagonal lattices because they simplify neighborhood indexing, visualization, and vectorized updates (Kohonen, 1990; 2013). Hexagonal lattices are often preferred in the literature and serve as the regular-topology baseline in this manuscript (White & Kiester, 2008; Kohonen, 2013; Forest et al., 2020).

The SOM literature has also explored alternatives to fixed lattices, including dynamic maps and graph-structured neighborhoods (Vasighi & Amini, 2017; Spanakis & Weiss, 2016; Kangas et al., 1990; Jang et al., 2009). MSTs have previously appeared in SOM analyses, but prior uses generally treat the MST as an interpretive structure over an already trained map rather than as the neighborhood relation that drives SOM learning. For example, Jang et al. use MSTs for interpretation, subnode embedding, and map-shape assessment, while FlowSOM overlays an MST on trained SOM codes to visualize relationships among metaclusters (Jang et al., 2009; Van Gassen et al., 2015). This is distinct from the training-time role used here: in FloatSOM, the MST is the operative neighborhood graph during SOM updates, is recalculated from the evolving node-weight geometry, and directly changes the update influence matrix used during learning. Thus, current deployed MST uses in SOM workflows generally configure post hoc connections on trained regular lattice maps, whereas FloatSOM uses MST/RNG graphs as the training neighborhood itself. Earlier SOM variants also discussed MST-defined neighborhoods during learning (Kangas et al., 1990), but these alternatives have not been assessed on large-scale datasets and do not have implementations that are either publicly available or suitable for distributed GPU computation. FloatSOM's topology contribution is therefore a scalable GPU-compatible implementation and large-scale quantification of refreshed graph-based SOM training, rather than a post hoc MST overlay on a conventional trained SOM.

Relative Neighborhood Graphs (RNGs) (Toussaint, 1980) are of particular interest here. To our knowledge, dynamically refreshed RNG neighborhoods have not previously been used as a SOM training topology; we return to the full rationale and implementation for RNG in Section 3.2.2.

## 2.4 Hyperparameter Optimization and Fair Comparison

SOM performance depends strongly on parameters such as map size, initialization, learning-rate schedule, and neighborhood schedule. Prior work shows that these choices can substantially affect observed performance (Akinduko et al., 2016; Forest et al., 2020). Because recent SOM system comparisons often emphasize implementation-level performance against existing tools or reference configurations (Mancini et al., 2020; Kratochvíl et al., 2020), comparisons based only on untuned settings can be difficult to interpret.

More generally, modern machine-learning workflows increasingly rely on automated hyperparameter optimization rather than manual tuning alone. Tools such as Optuna provide bounded search over large hyperparameter spaces and can support multi-objective optimization, allowing parameter settings to be selected with respect to several benchmark criteria simultaneously rather than collapsed into a single score (Akiba et al., 2019). Accordingly, we test both tuned and untuned reference and FloatSOM implementations to distinguish performance under untuned settings from performance obtained under more suitable hyperparameters.

## 3 Methods

FloatSOM combines four methodological components: sample selection, topology definition, SOM updates, and the compute architecture used to execute them. The framework supports random, full, and HDSSSOM sampling; rectangular, hexagonal, MST, and RNG topologies; and execution modes ranging from local GPU training to single-node and multi-node multi-GPU operation. Hyperparameter optimization is treated as a separate layer applied across these configurations. Figure 1 summarizes this design.

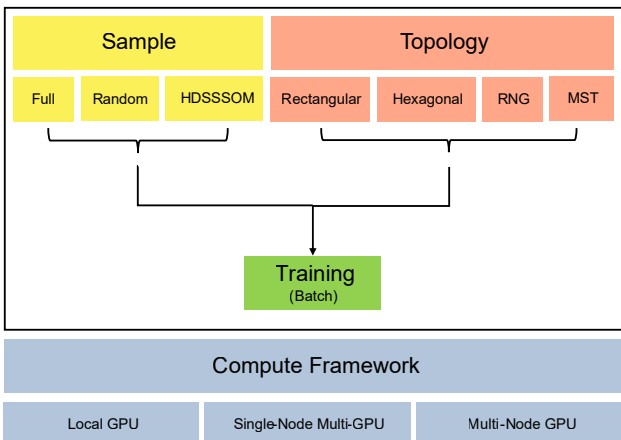

Figure 1: Schematic overview of the FloatSOM methods framing used in this manuscript. Sample selection and topology definition have configurable components that feed into the standard SOM training step, while the compute architecture determines how that same training procedure is executed in practice. The options shown here summarize the FloatSOM configurations discussed in the following subsections.

## 3.1 Sampling Selector Mathematics

This section formalizes the sampling policies evaluated in this manuscript.

Let the full dataset be $X = \{x_i\}_{i=1}^{N}$, where $N$ is the total number of samples in the dataset. Let $m$ denote the number of samples presented to the SOM in a given training iteration, or the sampling budget. This budget is either fixed directly or determined as a proportion $\rho$ of the dataset:

$$m = \begin{cases} m_0, & \text{if a fixed budget is specified,} \\ \max(1, \lfloor N\rho \rfloor), & \text{if a dataset proportion is used.} \end{cases} \tag{1}$$

For index sets $\mathcal{I}_t^{(s)}$ at iteration $t$, the implemented full and random selectors are:

$$\mathcal{I}_t^{\text{full}} = \{1, \dots, N\}, \qquad \mathcal{I}_t^{\text{random}} \sim \text{Unif}(\{I \subseteq \{1, \dots, N\} : |I| = m\}), \; m < N. \tag{2}$$

If $m \geq N$, random returns the full dataset (no subsampling). Otherwise, to avoid biasing training toward a single fixed subsample, the random selector redraws $\mathcal{I}_t^{\text{random}}$ from the entire dataset at every training iteration. The selected training subset is $X_t^{(s)} = \{x_i : i \in \mathcal{I}_t^{(s)}\}$. Because random subsampling draws uniformly from size-$m$ subsets, we are able to better obtain coverage over the full dataset.

HDSSSOM is implemented in multi-GPU-compatible form as an adaptive sampler that preferentially revisits difficult, under-trained, or stale samples (Wetmore et al., 2005).

### 3.2 Topology Definition

After initialization, neighborhood relations are defined by the selected topology. For regular-lattice baselines, we support both grid and hexagonal layouts. Consistent with prior SOM guidance, we treat hexagonal as the standard reference topology in this manuscript. We implement the hexagonal lattice topology in accordance with (Vettigli, 2018). For MST and RNG, neighborhood structure is derived from the current node-weight geometry using the methodologies below.

Thus, regular-lattice neighborhoods are fixed before training, whereas MST and RNG neighborhoods are recalculated from the evolving node weights during training.

Note that regardless of the selected topology, FloatSOM uses the same node-weight initialization options. Initialization determines only the starting node weights; neighborhood relations are applied afterward according to the selected topology to establish node connections, keeping the initial state comparable across regular-lattice and graph-based runs.

#### 3.2.1 MST Topology Implementation

The MST topology replaces fixed lattice neighborhood distance with graph hop distance on a minimum spanning tree built from current SOM nodes. The MST is therefore part of the SOM training rule: it determines which nodes receive neighborhood influence during each topology-refresh interval, rather than serving as a visualization or analysis layer after training. For $P$ nodes in feature dimension $d$, the pairwise node-distance matrix is formed with the standard squared-distance Gram identity, which avoids 3D broadcast tensors and preserves $O(P^2 d)$ dense linear-algebra structure.

In a standard lattice SOM, neighborhood distance is determined by fixed node grid coordinates, which are assigned before training and are independent of the learned node-weight geometry in data space. MST instead uses the node weight vectors to locate the nodes in data space, then calculates the minimum spanning tree between those node locations (Kruskal, 1956). The node distances therefore determine which tree edges are selected, while the neighborhood distance used by the Gaussian update is the graph hop distance along the resulting tree. Topology-derived influence matrices are cached and refreshed according to the topology-refresh schedule.

At the beginning of training, the topology is refreshed every iteration: the previous tree edges are discarded, pairwise distances are recalculated from the updated node weights, and the MST is recomputed from those updated node locations. The refresh rate then decays as training proceeds, so topology updates become less frequent later in training. If the current iteration does not trigger recomputation, the previous graph state and cached influences are reused. Otherwise, the MST is recalculated and the influence cache is rebuilt.

---

**Algorithm 1** Dynamic MST topology update with refresh-triggered recomputation and cached influence reuse.

---
 1: **Input:** node weights $W_t$, iteration $t$, topology-refresh policy
 2: **Output:** topology state $(E_t, g_t, \text{cached influences})$
 3: Query the refresh policy for the current iteration
 4: **if** no topology refresh is due **then**
 5:     return previous topology state
 6: **end if**
 7: Compute pairwise squared node distances on GPU
 8: Transfer distances to CPU and run Kruskal to obtain MST edges $E_t$
 9: Build adjacency from $E_t$
10: Compute all-pairs graph hop distances $g_t$ with chunked GPU Floyd-Warshall
11: Deduplicate active radii and rebuild/update cached influence maps
12: Commit $E_t$, $g_t$, and cache state
13: return topology state

---

### 3.2.2   RNG Topology Implementation

Our RNG topology constructs a Relative Neighborhood Graph over current node distances using the standard RNG criterion (Toussaint, 1980). Unlike MST, RNG is not restricted to a single spanning-tree backbone, so multiple locally supported neighborhood relations can be retained. Conceptually, the absence of this restriction permits the RNG topology to represent both tree-like and denser mesh-like local structures simultaneously throughout the training process. Consequently, we hypothesise that this greater freedom in topology will allow RNG-based SOM maps to better represent a variety of distributions.

---

**Algorithm 2** Dynamic RNG topology update with refresh-triggered recomputation and cached influence reuse.

---
 1: **Input:** node weights $W_t$, iteration $t$, topology-refresh policy
 2: **Output:** topology state $(E_t, g_t, \text{cached influences})$
 3: Query the refresh policy for the current iteration
 4: **if** no topology refresh is due **then**
 5:     return previous topology state
 6: **end if**
 7: Compute pairwise squared node distances on GPU
 8: Evaluate RNG candidate elimination in chunks using the blocker test
 9: Retain surviving RNG edges $E_t$ and build adjacency
10: Compute all-pairs graph hop distances $g_t$ with chunked GPU Floyd-Warshall
11: Deduplicate active radii and rebuild/update cached influence maps
12: Commit $E_t$, $g_t$, and cache state
13: return topology state

---

## 3.3   Multi-GPU + Larger-Than-Memory Methodology and Implementation

FloatSOM is implemented for distributed GPU execution, with support for RAM (Random Access Memory, CPU memory) spilling and disk-backed execution when datasets exceed available VRAM and RAM, respectively.

### 3.3.1   General Multi-GPU Logic

Distributed execution uses Ray actors with one GPU per worker and NCCL collectives for synchronous aggregation (Moritz et al., 2018). At each iteration, every worker processes its assigned shard locally in `n_chunks`, accumulates worker-local statistics across those chunks, and synchronizes once per iteration.

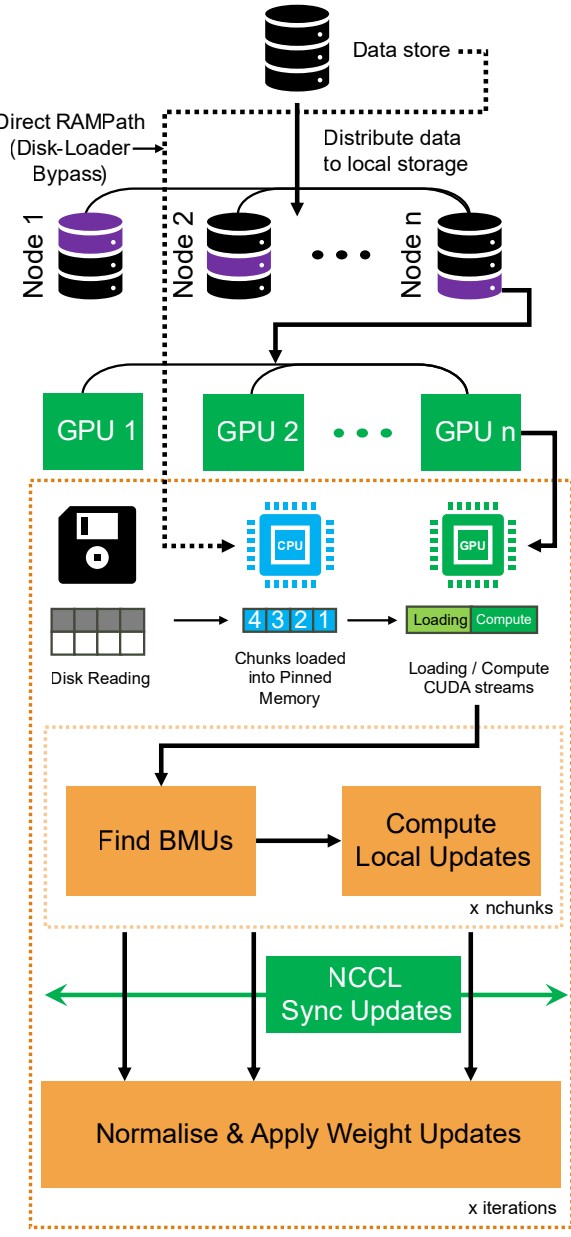

Figure 2: Multi-GPU data loading and NCCL synchronization schematic. In disk-backed mode, data are sharded to worker-local storage, loaded chunk-wise into pinned host memory, transferred to GPU with transfer overlapped with compute, processed locally, and synchronized by NCCL all-reduce before one weight update per iteration. In RAM mode, data are instead pre-sharded into worker-local CPU RAM and follow the same pinned-memory-to-GPU path without disk reads.

For worker $g \in \{1, \ldots, G\}$, let $X_{t,g}^{(s)}$ denote the worker-local shard of the selected iteration subset $X_t^{(s)}$. Let $j$ index SOM nodes, let $w_j^{(t)}$ denote the weight vector of node $j$ at iteration $t$, let $b(x)$ denote the best-matching unit (BMU) of sample $x$ under the current weights, let $h_{j,b(x)}^{(t)}$ denote the iteration-$t$ neighborhood influence between node $j$ and the BMU (Best Matching Unit)of $x$, and let $\eta_t$ denote the learning rate at iteration $t$. The local accumulators are:

$$
\begin{aligned}
U_j^{(g)} &= \sum_{x \in X_{t,g}^{(s)}} \eta_t \, h_{j,b(x)}^{(t)} (x - w_j^{(t)}), \\
H_j^{(g)} &= \sum_{x \in X_{t,g}^{(s)}} h_{j,b(x)}^{(t)}.
\end{aligned}
\tag{3}
$$

$U_j^{(g)}$ is worker $g$'s summed learning-rate-scaled displacement for node $j$, and $H_j^{(g)}$ is the corresponding summed neighborhood influence used to normalize that displacement.

Global synchronized accumulators are:

$$
\begin{aligned}
U_j &= \sum_{g=1}^{G} U_j^{(g)}, \\
H_j &= \sum_{g=1}^{G} H_j^{(g)}.
\end{aligned}
\tag{4}
$$

In implementation, $U_j^{(g)}$ and $H_j^{(g)}$ are accumulated across the worker's `n_chunks` and synchronized once per iteration. After synchronization, $U_j$ and $H_j$ define the global update numerator and normalization denominator for node $j$. Under weighted normalization, the normalized update is given by $U_j/H_j$ up to numerical safeguards; optional momentum is then added to this normalized update before applying the resulting change to the node weight $w_j^{(t)}$.

For $P$ nodes and $d$ features in float32, the two update accumulators contain $Pd + P$ values, so their logical all-reduce payload is $4P(d + 1)$ bytes per worker per iteration, plus one 4-byte sample count. A ring all-reduce sends and receives approximately $2(G - 1)/G$ times that payload per worker for $G$ workers. This communication is independent of the number of observations $N$; increasing $N$ instead increases worker-local computation and data staging. With observation chunks of at most $C$ rows, the dominant bounded worker arrays scale as $O(Cd + Pd)$ rather than requiring the full $O(Nd)$ dataset in GPU memory. Graph-topology distance and influence structures add both computation and storage terms that scale with $P^2$. Their construction and refresh therefore become more expensive as node count increases, particularly for RNG. FloatSOM tiles these structures and can spill them from VRAM to system RAM when necessary. Spilling is a memory-management response that may add transfer overhead, but it is not the sole source of the topology-dependent runtime increase.

### 3.3.2 Multi-GPU Implementation Details

As shown in Fig. 2, each worker uses a chunked loading path from CPU memory to GPU memory. In streaming mode, data are distributed to worker-local disk shards and then read chunk-by-chunk into pinned host memory before transfer to GPU; in RAM mode, data are pre-sharded into worker-local CPU RAM and follow the same pinned-memory path without disk reads. FloatSOM overlaps data transfer and compute across CUDA streams, uses JIT-compiled kernels (Just-In-Time) for the core BMU and update steps, and synchronizes worker-local accumulators once per iteration with NCCL all-reduce. Weights therefore remain resident on worker GPUs across iterations. For large topologies that would otherwise exceed VRAM, Float-SOM also supports spilling the associated graph-distance or influence structures from VRAM to system RAM.

### 3.3.3 Larger-Than-Memory Capable Topology Updates

Additional larger-than-memory support for topology updates is provided through topological chunking. Topological chunking applies the same idea to topology-side computations within each worker. When graph-distance or influence structures would otherwise exceed a worker's VRAM, those computations are tiled and evaluated in bounded pieces rather than materialized at once. This topology-side chunking is not depicted in the Fig. 2 data path, but it follows the same per-worker bounded-memory execution rule.

### 3.4 Multi-Objective Hyperparameter Optimization

We use Optuna as an automated multi-objective hyperparameter optimization tool to derive near-optimal performance and corresponding hyperparameters for each FloatSOM configuration under a given sampling and topology combination (Akiba et al., 2019).

## 4 Experimental Setup

We use two benchmark protocols: an Optuna quality benchmark and a speed scaling benchmark. The first evaluates algorithmic quality and tuned attainable performance, and the second evaluates runtime and distributed scaling behavior. All production Optuna and speed benchmarks reported in this manuscript were executed on Gadi at the National Computational Infrastructure (NCI), Australia, on gpuvolta nodes (National Computational Infrastructure, n.d.), using a consistent multi-GPU environment across runs.

### 4.1 Optuna benchmark protocol

We use Optuna-based multi-objective optimization to determine the best attainable performance and corresponding hyperparameters for each sampling (full and random) and topology (hexagonal, MST, and RNG) combination, optimizing jointly for train and holdout quantization error ($QE_T$ and $QE_H$). In this protocol, each dataset-topology-sampling configuration is optimized for 200 trials across 10 seeds under variant-appropriate search constraints. The reported runs used the standard in-memory batch path rather than the Ray-distributed execution stack. This corresponds to:

$$14_{\text{Datasets}} \times 10_{\text{Seeds}} \times 3_{\text{Topologies}} \times 2_{\text{SamplingMethods}} \times 200_{\text{Trials}} = 168{,}000 \text{ Runs}$$

The neighborhood-radius search space was shared across topology families. In particular, `initial_radius` was an Optuna-optimized parameter for hexagonal, MST, and RNG runs, with the same search interval of 0.5 to 10.0 in each case. Thus, the hexagonal topology comparisons below use a tuned hexagonal comparator rather than a default-radius hexagonal baseline.

Sampling comparisons in Section 5.2 compare full versus random paired analyses pooled across all topologies. Topology comparisons in Sections 5.3-5.4 use full sampling runs across hexagonal, MST, and RNG. Otherwise, the remaining analyses use full sampling. A separate focused HDSSSOM pilot is reported in Section 5.2 and summarized in Supplementary Table S2. This pilot additionally included the HDSSSOM sampling methodology alongside the full and random sampling methodologies.

### 4.1.1 Optuna benchmark datasets and preprocessing

The Optuna benchmark uses 14 synthetic and real scikit-learn datasets spanning low- and high-dimensional regimes (Pedregosa et al., 2011); dataset metadata are listed in Supplementary Table S1. Synthetic datasets are generated according to the random seed, whereas real datasets are loaded directly from sklearn. Across the Optuna protocol, inputs are standardized feature-wise to zero mean and unit variance using `StandardScaler`, then deterministically permuted and split into fixed 70/30 train-holdout partitions. We retain both partitions to evaluate both training-set representation and transfer to unseen samples.

### 4.1.2  Optuna quality metrics

Our primary quality metric is Quantization Error ($QE$) (Kohonen, 2001). We report $QE$ on both the training partition ($QE_T$) and the holdout partition ($QE_H$). The training value reflects how well the fitted SOM represents the samples it was trained on, whereas the holdout value evaluates the same trained map after projecting previously unseen samples onto it. The holdout partition is therefore not used during fitting; it is used only to evaluate the trained map, analogous to a test set in conventional model-training workflows.

Balanced $QE$, denoted $QE_B$, is defined as the mean of $QE_T$ and $QE_H$. $QE_B$ is therefore a composite endpoint that weights data representation fidelity (train) and generalizability (holdout) equally. Note that in the Optuna runs, $QE_T$ and $QE_H$ are optimized jointly as a two-objective vector, with $QE_B$ only calculated *post hoc*.

We quantify local BMU-neighborhood ordering using Mean Tied Rank (MTR), following its application to comparisons among alternative SOM grid topologies (López-Rubio & Díaz Ramos, 2014). For each sample $x_i$, let $b_i^{(1)}$ and $b_i^{(2)}$ denote the first and second best-matching units (BMUs), respectively. Starting from $b_i^{(1)}$, we group all non-winning units by the number of hops along the shortest path in the SOM graph. Let $d_i$ be the number of hops from $b_i^{(1)}$ to $b_i^{(2)}$, let $S_{i,d_i}$ be the set of units exactly $d_i$ hops away, and let $L_{i,d_i}$ be the number of units fewer than $d_i$ hops away. Because every unit in $S_{i,d_i}$ is tied at the same hop distance, the second BMU is assigned their average ordinal rank:

$$
\tau_i = \underbrace{L_{i,d_i}}_{\text{units fewer hops away}} + \underbrace{\frac{|S_{i,d_i}|+1}{2}}_{\text{average rank at the same hop distance}},
$$

$$
\text{MTR} = \underbrace{\frac{1}{N}\sum_{i=1}^{N}\tau_i}_{\text{mean tied rank across the } N \text{ evaluated samples}}.
$$

Hexagonal, MST, and RNG maps define adjacency differently: hexagonal connections are fixed by the lattice, MST connections form a sparse tree derived from the node weights, and RNG connections are also weight-derived but can give nodes different numbers of neighbors. MTR applies the same graph-distance ranking procedure to each trained map while accounting for topology-specific differences in node degree and tied hop distances. The $L_{i,d_i}$ term counts all units fewer hops away, while the second term assigns the average rank across the $|S_{i,d_i}|$ units at the same hop distance as the second BMU. It therefore supports consistent comparison among the fixed hexagonal, MST, and RNG adjacency structures. Lower MTR indicates that first and second BMUs tend to occur earlier in this graph-specific ordering and therefore reflects better local neighborhood ordering.

We additionally report node utilization, defined as the fraction of nodes selected as a BMU, and its complement, dead-node fraction. These post hoc diagnostics are computed on training and holdout splits and balanced as for $QE$.

### 4.1.3  XPySOM calibration protocol

We calibrated the FloatSOM batch pathway against XPySOM using the 14 benchmark datasets, 10 shared seeds, full sampling, a $32 \times 32$ map, and one run per dataset–seed condition without top-$k$ selection. Both implementations used the same deterministic 70/30 train–holdout split, number of batch epochs, and XPySOM-like untuned hyperparameters. XPySOM's initialized node weights were transferred into FloatSOM after reconciling the two implementations' node-coordinate ordering, so paired runs began from corresponding prototypes. The primary implementation-concordance comparison used the hexagonal topology in both implementations; the associated MST and RNG FloatSOM paths retained the same XPySOM hexagonal reference to quantify the additional effect of changing the FloatSOM training topology.

We recorded train, holdout, and balanced $QE$ together with training time. Within each dataset, paired percentage effects across the 10 seeds were summarized using the Wilcoxon signed-rank location estimate, its exact distribution-free 95% interval, and a two-sided paired Wilcoxon test; global rows pool the 140 dataset–seed pairs. Supplementary Figs. S1–S3 and Supplementary Tables S4–S6 report these results.

### 4.2 Speed scaling benchmark protocol

The speed scaling benchmark evaluates runtime and distributed scaling behavior in different compute and algorithm configurations. Speed scaling is evaluated with runs across $G \in \{1, 2, 4, 8\}$ GPUs under a series of fixed scaling protocols. Runtime summaries are computed from repeated executions per configuration and reported as both absolute training time and efficiency relative to the corresponding 1-GPU comparison. We compute scaling efficiency as $E_G = (T_1/T_G)/G \times 100\%$, where $T_1$ is the 1-GPU runtime and $T_G$ is the runtime on $G$ GPUs. A value of 100% corresponds to ideal linear scaling; lower values indicate that communication, orchestration, or I/O overhead has reduced the realized speedup, while values above 100% can occur when additional GPUs also change the memory or data-staging regime. These scaling runs use the Ray-orchestrated distributed execution layer built on top of the standard FloatSOM training path (Moritz et al., 2018). Accordingly, the scaling figures in Sections 6.1-6.2 and the runtime/scaling comparison reported later against XPySOM should be interpreted as distributed-execution results rather than the in-memory Optuna path.

Some 1-GPU baselines were unavailable because the corresponding runs did not complete within the fixed benchmark timeout, particularly for workloads entering disk-backed execution. Because efficiency is defined relative to the 1-GPU runtime, we estimated these missing denominators by local extrapolation from the last successful 1-GPU point on the same curve. These extrapolated denominators are plotting references rather than measured runtimes and are interpreted only descriptively.

Topology speed comparisons include hexagonal, MST, and RNG, with harmonized workload settings so ratios isolate topology-associated runtime effects. A dedicated random versus full comparison, run on $G \in \{1, 2, 4\}$ GPUs, is additionally included to isolate sampling-specific runtime effects independently of the multi-GPU scaling runs.

XPySOM was selected as the executable external baseline because it most closely matches FloatSOM's Python/GPU batch-training regime. We also attempted aweSOM on the standard speed workload, but all five runs reached the 1800-s timeout before completing 60% of $N$ online updates. Matching FloatSOM's 10 batch iterations would require $10N$ pointwise updates, so aweSOM was not included in the timed comparison.

#### 4.2.1 Scaling benchmark datasets

The speed benchmark uses synthetic random matrices with uniform values in $[0, 1]$. No train-holdout split is used here because the objective is runtime rather than generalization, so all generated data are used for training. We use a shared default configuration of $10^7$ samples, 50 dimensions, a $32 \times 32$ SOM grid, and 10 training iterations, and then vary one workload axis at a time: sample count ($10^6$–$10^9$), feature dimension (50–5000), or grid side length (8–64). The exact axis values used for each scaling panel are provided in Supplementary Table S3.

Additional CPU and RAM resources attached to each GPU are scaled linearly with GPU count while keeping the software environment and benchmark procedure consistent across runs. For scaling benchmarks, each run is timed out if it exceeds 30 minutes of wall time. Per-GPU resource allocation is fixed at one NVIDIA V100 (32 GB VRAM), 12 CPU cores, 90 GB system RAM, with one full node comprising 4 GPUs and their associated CPUs, and with 400 GB of associated local disk storage.

### 4.3 Hyperparameter Tuning and Stability

To quantify parameter-tuning benefit, we performed an explicit paired analysis between the tuned configuration and the XPySOM untuned default reference. For each sampling mode and topology combination, we first extracted the parameter settings from the best-performing Optuna runs under the benchmark objective

for that combination. We then distilled these per-seed best-performing tuned settings into deployable default configurations by taking the mean of numeric parameters and the mode of categorical parameters.

The stability analysis focuses on four tuned hyperparameters that govern SOM training dynamics. These settings determine how the map is initialized and how updates evolve over training: the initial radius sets the early neighborhood scale around each BMU, the initialization method sets the starting node weights, the radius decay type controls the shift from broad global organization toward local refinement, and the momentum-use parameter determines whether each update retains part of the previous update direction.

### 4.3.1 Tuned Configuration versus Untuned Reference Analysis

Fixed configurations were rerun under matched dataset, seed, topology, and split keys. The final topology diagnostic comprised all 14 benchmark datasets, 20 shared random seeds, full sampling, and each of the hexagonal, MST, and RNG topologies, giving 280 dataset–seed units per paired contrast. Preprocessing and the deterministic 70/30 train–holdout split followed Section 4.1.1. The untuned profile used the same XPySOM-like reference settings for all topologies, whereas the tuned profile used the fixed topology-specific full-sampling configurations derived above. Deployment analyses compare these tuned FloatSOM configurations with untuned hexagonal XPySOM, while topology diagnostics compare tuned and untuned FloatSOM within and between topologies. These reruns estimate deployable fixed-configuration performance; the top-$k$ Optuna summaries instead estimate attainable performance within the search budget.

For Supplementary Fig. S5, tuned and untuned values were paired within dataset, seed, topology, and split. Supplementary Table S12 pools the 280 matched dataset–seed differences for each prespecified profile–topology–metric contrast and reports the mean paired effect, two-sided one-sample paired $t$-test confidence interval, Cohen's $d_z$, raw p-value, and win/loss/tie counts. These raw p-values are descriptive diagnostics rather than a multiplicity-adjusted family; effect sizes and confidence intervals are the primary summaries. Supplementary Table S13 instead reports the corresponding mean observed metric within each profile, dataset, and topology across the 20 seeds. Dataset-level forest-plot results use the convention described above, and overall rows pool all matched pairs with the same paired $t$-test and confidence-interval calculation.

### 4.3.2 Hyperparameter stability and dataset-type stratification

Hyperparameter stability is important because it determines whether a method can be deployed reliably without bespoke retuning. We therefore extracted the top-ranked tuned Optuna trial separately for each seed and compared the recovered hyperparameter values across seeds. We define hyperparameter stability as the variation in these recovered settings. For numeric parameters, stability is measured by the relative difference between runs, $|a - b| / \max(|a|, |b|, \varepsilon)$, where $a$ and $b$ are the values of a given parameter for the two compared seeds and $\varepsilon = 10^{-12}$. These relative differences are then averaged across numeric parameters, with lower values indicating higher stability. For categorical parameters, stability is defined as the mismatch rate across the same seed pairs. We report both mean per-parameter stability and an equal-weight overall stability summary across datasets.

### 4.3.3 Matched initial-radius sensitivity analysis

To determine whether topology differences could be explained by topology-specific treatment of neighborhood radius, we performed a controlled initial-radius sweep. Hexagonal, MST, and RNG maps were evaluated at $r \in \{0.5, 0.75, 1.0266, 1.5, 2, 3, 5\}$ using the same 20 random seeds on each of the 14 benchmark datasets. The value $r = 1.0266$ was the selected hexagonal full-sampling radius and is denoted `Hex optimal` in Fig. 9. Apart from topology and the deliberately varied initial radius, the training configuration was fixed: full sampling, random initialization, asymptotic radius decay, momentum enabled with initial momentum 0.6069, and XPySOM-compatible normalization. Runs were therefore matched by dataset and seed, ensuring that topology families received the same radius values rather than different search ranges or topology-specific radius schedules.

We evaluated balanced $QE$, balanced MTR, and balanced node utilization. Because absolute $QE$ scales differ substantially among datasets, each dataset–seed $QE_B$ value was divided by the matched hexagonal $QE_B$ at $r = 1.0266$. Ratios were averaged on the log scale across seeds within each dataset and then across datasets with equal dataset weight; exponentiating this mean yields the geometric-mean ratio shown in Fig. 9A. Balanced MTR and balanced node utilization are retained in their observed units in Fig. 9B and Fig. 9C, respectively. Along each topology's response curve, direction-aware percentage changes from $r = 1.0266$ were calculated for all 280 matched dataset–seed units and tested against zero using two-sided one-sample $t$-tests; Benjamini–Hochberg adjustment was applied across the 54 non-anchor radius–topology–metric tests. These tests assess within-topology radius response.

We additionally conducted post hoc cross-topology $QE_B$ contrasts at each radius. For every topology pair, dataset, seed, and radius, we calculated the log ratio of the comparator $QE_B$ to the reference $QE_B$, averaged log ratios across the 20 seeds within each dataset, and applied a two-sided one-sample $t$-test to the 14 equally weighted dataset effects. Exponentiated means and confidence limits are geometric-mean $QE_B$ ratios, with values below 1 favouring the comparator. Benjamini–Hochberg correction was applied across all 21 topology-pair–radius contrasts. Supplementary Table S15 reports the estimates, intervals, raw p-values, and adjusted q-values.

### 4.3.4 Execution-path concordance analysis

To determine whether the quality diagnostics depended on execution pathway, we compared local CuPy and Ray streaming under matched inputs and fixed tuned configurations. The complete protocol is provided in Supplementary Methods S1, and the paired results are reported in Supplementary Table S14.

### 4.4 Statistical analysis

All Optuna comparisons use matched pairs within dataset, seed, and split units to control for substantial between-run heterogeneity. Within each matched unit, trials were ranked by the target metric, the top five were retained, and each condition was summarized by the median of those retained trials, yielding a top-$k$ summary with $k = 5$ intended to estimate near-optimal attainable performance under a fixed number of tuning trials, where each trial is one candidate hyperparameter configuration evaluated by Optuna. Paired effects were then computed as simple condition differences, with negative values favoring the first condition for lower-is-better metrics. Dataset-level and global summaries are shown as forest plots with 95% confidence intervals from two-sided paired one-sample $t$-tests. Supplementary Fig. S4 repeats the topology comparisons at $k \in \{1, 3, 5, 10\}$, always using the median of the retained trials within each matched unit, to assess sensitivity to the number of near-optimal trials retained.

We report Benjamini-Hochberg q-values alongside raw p-values for related dataset-level tests, and use q-values for figure markers and significance counts. Each topology contrast in Figs. 6-7 forms a separate family of 42 tests (14 datasets across $QE_B$, $QE_H$, and $QE_T$). Pooled overall tests are reported separately with raw p-values.

## 5 Results

### 5.1 XPySOM calibration

Under matched-configuration XPySOM versus FloatSOM calibration on hexagonal $QE$ (Fig. S3), no implementation-associated $QE$ difference was detected. Accordingly, we use hexagonal FloatSOM batch as the operational XPySOM reference in the benchmarks that follow, while not claiming formal statistical equivalence. The runtime differences are attributable to FloatSOM's JIT kernels, which incur a small startup cost. This overhead is progressively amortized as workload size increases, after which FloatSOM runs faster than XPySOM on larger datasets.

**Figure 3: Full versus HDSSSOM on Pilot Dataset**

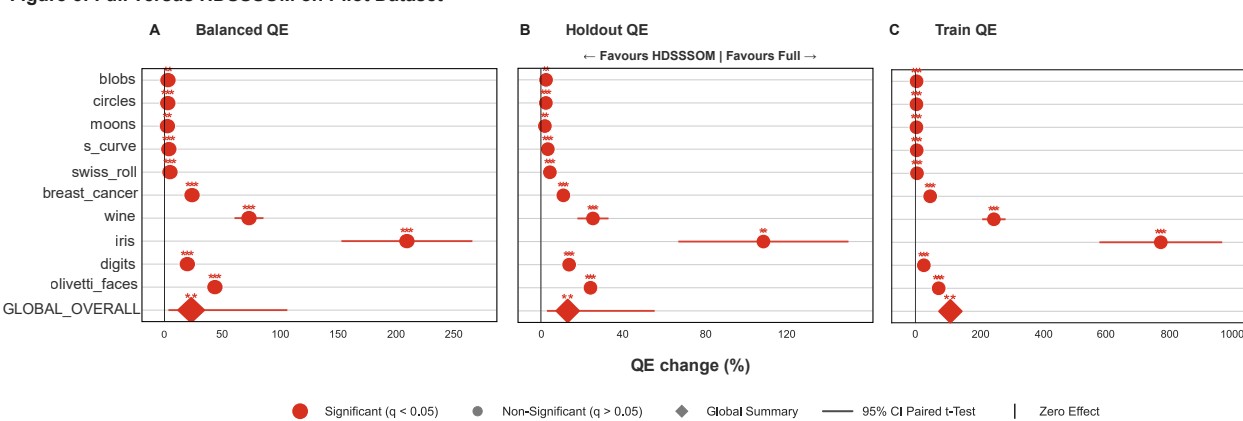

Figure 3: HDSSSOM screening pilot on $QE_B$ (hexagonal topology): full vs HDSSSOM, using the smaller pilot configuration summarized in Supplementary Table S2 (10 datasets, 5 seeds, with all other pilot settings held fixed). Panels report dataset-matched paired top-$k$ within-unit medians plus dataset-level paired-effect summaries (Section 4.4). Forest whiskers denote 95% paired $t$-test confidence intervals around the mean paired effect.

## 5.2 Comparison of Different Sampling Methods

We first report a focused HDSSSOM pilot as an elimination comparison rather than as part of the broader sampling benchmark. This pilot used a smaller, more limited configuration than the later full versus random analysis and is summarized in Supplementary Table S2.

Under that pilot configuration, HDSSSOM was materially worse than full sampling. In the hexagonal view shown in Fig. 3, full outperformed HDSSSOM in all 50 paired comparisons (10 datasets × 5 seeds; 50 wins, 0 losses, 0 ties), with dataset-level median balanced-$QE$ improvements ranging from 2.5% to 209.7% and a global median improvement of 38.7%. These HDSSSOM results should therefore be interpreted as a pilot screen under the stated configuration rather than as a comprehensive evaluation of all possible HDSSSOM schedules. Given these large differences, we focused on the full and random sampling methodologies for the remainder of the manuscript.

The full versus random analysis in Fig. 4 was generated from matched hexagonal Optuna runs in which the sampling selector was switched from full to random. Effects were computed within matched seed-specific units and then summarized across seeds. Fig. 4A-C summarize the matched full versus random paired effects for balanced, holdout, and train $QE$, respectively, showing that full sampling generally provides equal or better $QE$ than random sampling. Notably, this full versus random separation is much smaller than the full versus HDSSSOM pilot effect shown in Fig. 3; in most dataset-level comparisons, the full versus HDSSSOM improvement is at least twice as large - as evidenced by the difference in axis scales between Fig. 3 and Fig. 4.

We find the effectiveness of random sampling relative to full sampling is scale-dependent. Above 10,000 samples, paired $QE$ differences are not meaningfully detected, whereas in smaller datasets the random arm shows higher variability and less stable outcomes (Fig. 4D-F). This is consistent with reduced per-iteration sample support under random subsampling (Fig. 4D-F). In this benchmark, the $> 10{,}000$ regime is therefore a useful practical proxy for more stable random sampling behavior.

Nevertheless, full sampling remains the best sampling strategy for optimal $QE$ results. Accordingly, all remaining analyses reported below rely on full sampling unless specified.

## Figure 4: Full vs Random Outcomes and Dataset-Size Regression (Hexagonal)

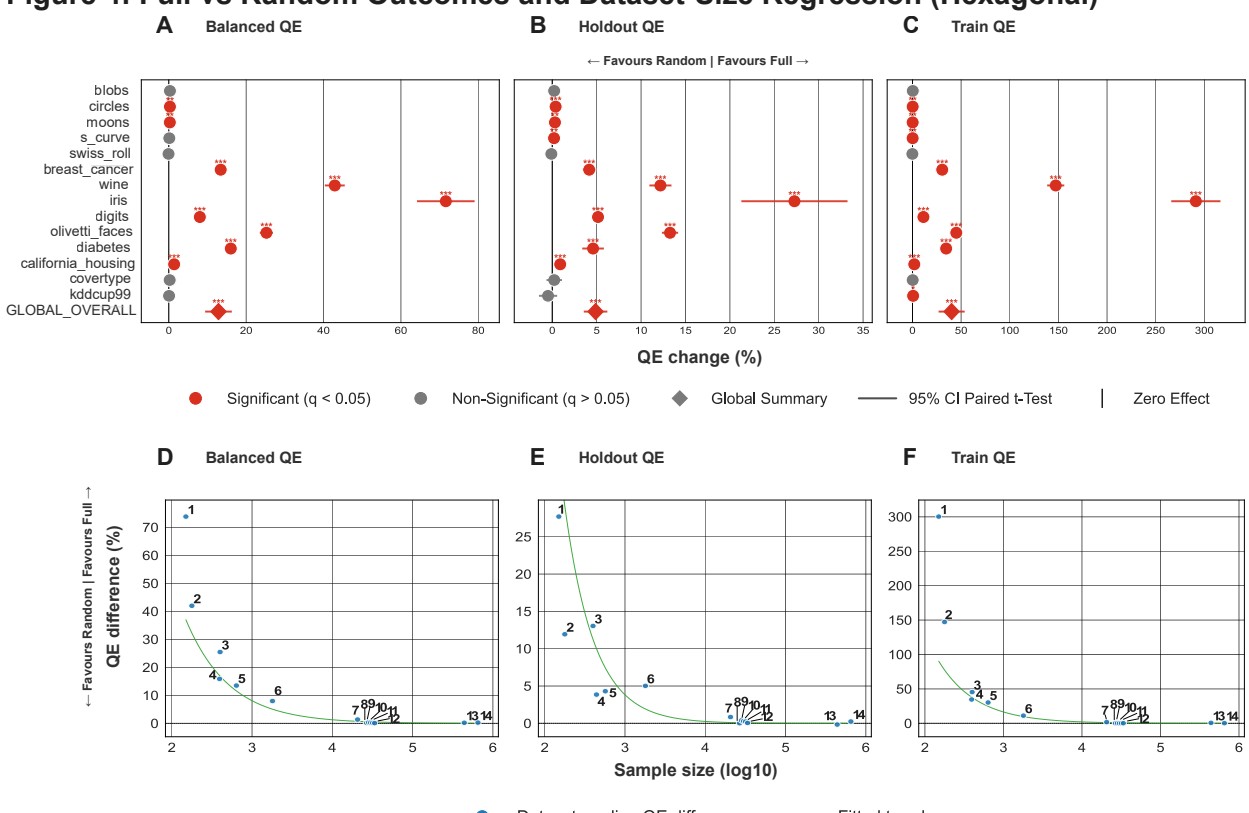

Figure 4: Full versus random sampling under the hexagonal topology. Panels A-C report paired full-versus-random effects for $QE_B$, $QE_H$, and $QE_T$. Panels D-F show the corresponding dataset-size regressions for the same three metrics, with numbered datasets keyed in Supplementary Table S1. Figure significance markers use Benjamini-Hochberg adjusted q-values. In the larger-dataset regime observed here ($>10{,}000$ samples), little paired $QE$ separation is detected; at smaller dataset scales, random is more variable and less stable.

### 5.3 Topology Results

Topology comparisons use $QE$ as the optimized endpoint and hexagonal batch SOM as the regular-topology baseline; Fig. 5 illustrates the three neighborhood structures. `initial_radius` was optimized over the same interval for every topology. The selected full-sampling values were 1.03 for hexagonal, 1.46 for MST, and 1.41 for RNG; under random sampling they were 1.17, 1.82, and 1.77. Thus, the graph-topology gains are not comparisons against a broader default-radius hexagonal map, although radius was not isolated from the other tuned parameters.

In the KDD Cup 99 projection (Fig. 5D–F), the hexagonal topology contains several apparently nonlocal connections spanning separated regions of the projected prototype distribution, whereas the MST and RNG connections more closely follow its local geometry. This pattern illustrates how predetermined lattice neighbors can couple the updates of prototypes that are not locally adjacent in the displayed data structure, restricting how independently the nodes can redistribute.

#### 5.3.1 MST

We report dataset-wise paired improvement summaries (hexagonal over MST) using the statistical procedure in Section 4.4 in Fig. 6.

**Figure 5: Representative Topologies**

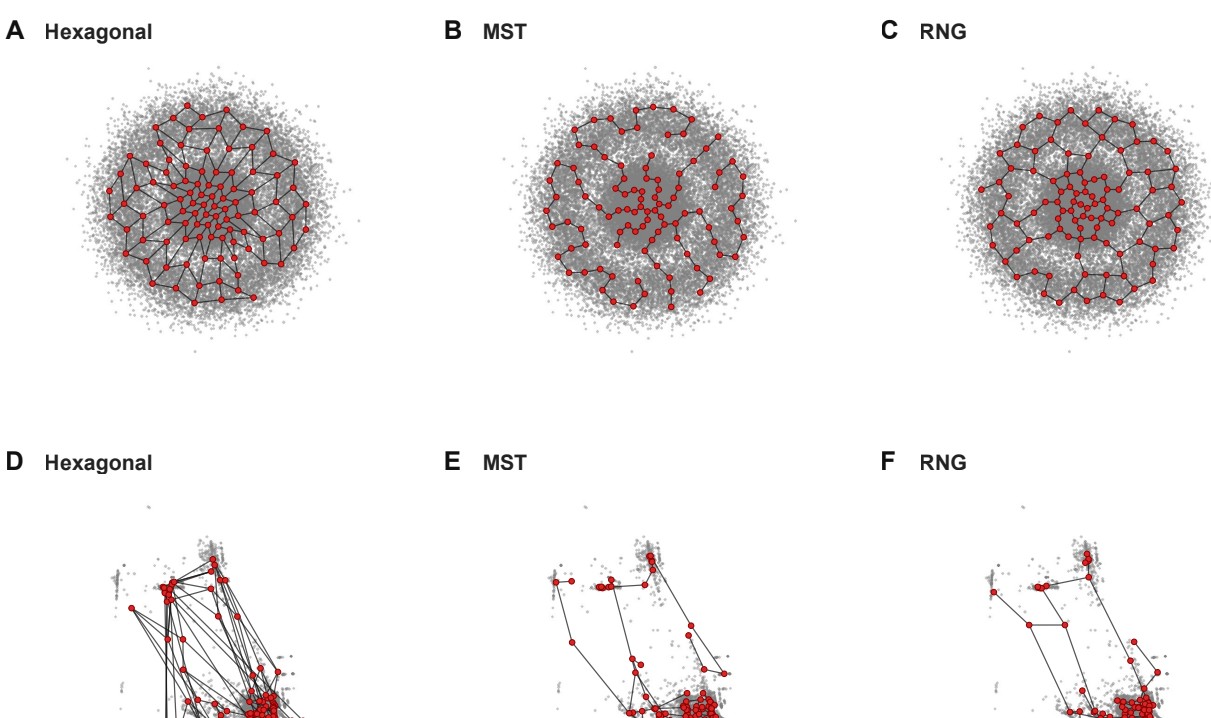

Figure 5: Representative node and connection overlays for matched 100-node hexagonal, MST, and RNG SOMs. A–C: circles in native 2D. D–F: KDD Cup 99, trained on standardized 41-dimensional observations and displayed using one shared 2D PCA projection fitted to the observations and applied to all three sets of prototypes. Training used seed 42, full sampling, random initialization, 50 iterations, and the untuned XPySOM-like settings used at this point in the manuscript: initial learning rate 0.5, initial radius 5 with exponential decay, momentum disabled, and XPySOM-compatible normalization. Grey observation clouds show a deterministic maximum of 30,000 points; black lines are topology connections and red markers are SOM nodes. PCA can distort graph geometry in the original 41-dimensional space.

Overall, MST outperforms matched hexagonal on Balanced QE (Fig. 6A), indicating a net advantage across train and holdout performance. This aggregate gain is driven more clearly by Train QE (Fig. 6C), while Holdout QE is more mixed across datasets (Fig. 6B) and shows no clear overall holdout advantage. The overall paired t-test p-values are Balanced QE (p=1.12e-05); Holdout QE (p=0.15); and Train QE (p=0.0064). Supplementary Table S7 lists the corresponding per-dataset and overall hexagonal-comparison effect estimates, 95% confidence intervals, raw p-values, and Benjamini-Hochberg q-values for MST and RNG.

### 5.3.2   RNG

To evaluate RNG topology performance, we reuse the paired reporting logic on hexagonal versus RNG, again centered on $QE_B$ with $QE_H$ and $QE_T$ in Fig. 7.

**Figure 6: Hexagonal vs MST Across QE Metrics**

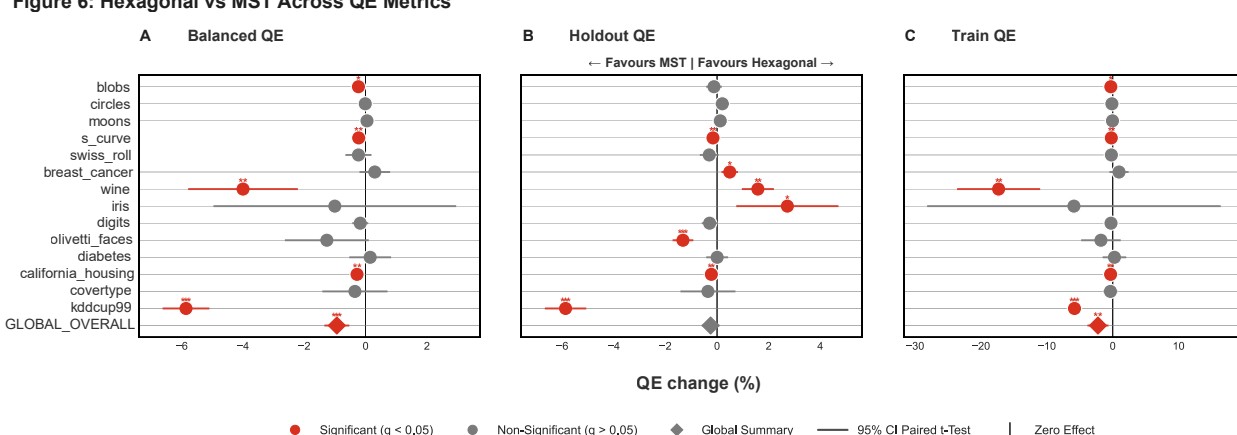

Figure 6: Hexagonal versus MST topology on $QE$ metrics under full sampling only. Panels A-C report paired full sampling only $QE$ effects for $QE_B$, $QE_H$, and $QE_T$ across the available full sampling datasets. Forest whiskers denote 95% paired $t$-test confidence intervals around the mean paired effect. Supplementary Table S7 reports the per-dataset effect estimates, 95% confidence intervals, raw p-values, and dataset-level Benjamini-Hochberg adjusted q-values.

**Figure 7: Hexagonal vs RNG Across QE Metrics**

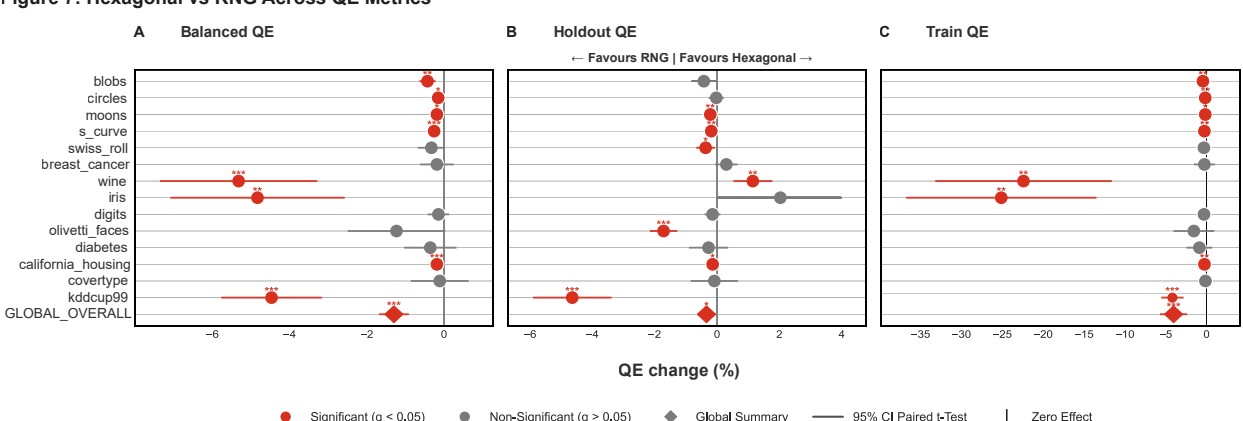

Figure 7: Hexagonal versus RNG topology on $QE$ metrics under full sampling only. Panels A-C report paired full sampling only $QE$ effects for $QE_B$, $QE_H$, and $QE_T$ across the available full sampling datasets. Forest whiskers denote 95% paired $t$-test confidence intervals around the mean paired effect. Supplementary Table S7 reports the per-dataset effect estimates, 95% confidence intervals, raw p-values, and dataset-level Benjamini-Hochberg adjusted q-values.

RNG has lower QE than matched hexagonal on the reported QE endpoints (Fig. 7A-7C), with overall paired t-test p-values of Balanced QE (p=7.4e-10); Holdout QE (p=0.0232); and Train QE (p=4.69e-06). Supplementary Table S7 lists the corresponding per-dataset and overall hexagonal-comparison effect estimates, 95% confidence intervals, raw p-values, and Benjamini-Hochberg q-values for MST and RNG.

The main trend in Fig. 7 is that RNG improves on hexagonal most clearly in balanced QE and especially in train QE, with the separation most apparent in the real and larger datasets where the added flexibility of the graph neighborhood appears more useful than the fixed regular lattice.

### 5.3.3 Direct MST–RNG comparison

**Figure 8: MST vs RNG Across QE Metrics**

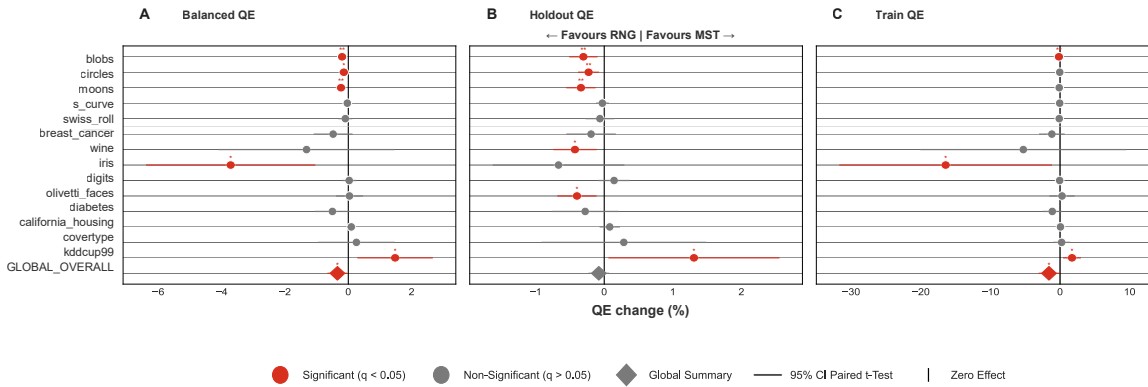

Figure 8: MST versus RNG topology on $QE$ metrics under full sampling only. Panels A-C report paired full sampling only $QE$ effects for $QE_B$, $QE_H$, and $QE_T$ across the available full sampling datasets. Forest whiskers denote 95% paired $t$-test confidence intervals around the mean paired effect.

Fig. 8 directly compares MST and RNG under full sampling. In the Optuna matched top-$k$ runs, RNG achieved significantly lower balanced and train $QE$ than MST, while no significant difference was detected for holdout $QE$. Across the topology comparisons in Figs. 6–8, RNG therefore showed the highest quantization capacity among the evaluated topologies.

### 5.3.4   Initial-radius sensitivity

Because initial radius was selected during tuning and differed across topology families, we isolated its effect in a fixed-configuration sweep. All non-radius hyperparameters were held at the tuned hexagonal full-sampling configuration, and each of seven radii was evaluated for hexagonal, MST, and RNG maps using the same 20 seeds on all 14 datasets. Fig. 9A reports $QE_B$ after normalizing every dataset–seed condition to its matched hexagonal value at the selected radius ($r = 1.027$). We summarize these ratios by averaging log ratios across seeds within each dataset, weighting the 14 datasets equally, and exponentiating the cross-dataset mean. The resulting geometric-mean ratio is dimensionless: 1 denotes matched hexagonal performance at the selected radius, values below 1 indicate lower $QE_B$, and values above 1 indicate higher $QE_B$.

Across the seven tested radii, MST and RNG maintained lower observed $QE_B$ than hexagonal, with adjusted differences evident from $r = 1.027$ onward.

Increasing radius eventually exchanged higher $QE_B$ for lower MTR in every topology, but this trade-off emerged earlier and more sharply for hexagonal maps. RNG maintained the lowest observed MTR throughout the sweep, while the node-utilization curves did not indicate that the graph-topology results were driven by poorer use of the available map nodes. The significance symbols in Fig. 9 represent within-topology radius comparisons.

The fixed-configuration diagnostics provide the corresponding deployable tuned-default comparison (Table 1). Unlike the Optuna matched top-$k$ results, no balanced-$QE$ difference was detected between the tuned RNG and MST defaults ($p = 0.126$). RNG nevertheless produced significantly better local ordering, lowering mean balanced MTR from 7.87 to 4.91, a difference of 2.96 tied-rank positions ($p = 1.75 \times 10^{-65}$). Thus, when non-winning nodes were ordered by graph distance from the first BMU, the second BMU appeared approximately three positions earlier under RNG than under MST. Finally, RNG also produced significantly higher balanced node utilization, increasing it by 0.0040 ($p = 6.07 \times 10^{-4}$).

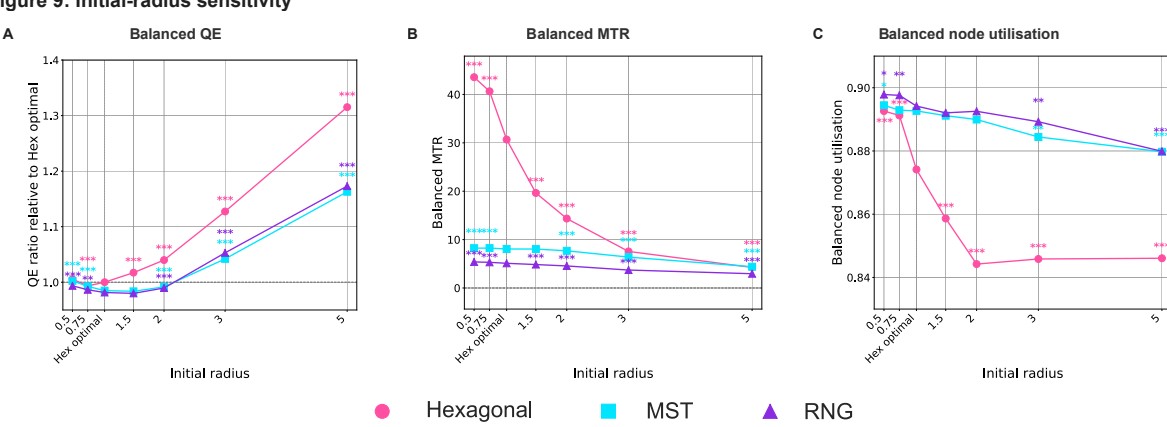

Figure 9: Initial-radius sensitivity under matched fixed configurations. A: dataset-balanced geometric-mean $QE_B$ ratio relative to the matched hexagonal configuration at its selected radius ($r = 1.027$); lower values are better and the dashed horizontal line marks a ratio of 1. B: observed balanced Mean Tied Rank (MTR; lower is better). C: observed balanced node utilization (higher is better). All panels summarize 14 datasets with 20 matched seeds per topology and radius. The dotted vertical line and `Hex optimal` tick mark $r = 1.027$. Significance markers use Benjamini–Hochberg adjusted tests comparing each non-anchor radius with $r = 1.027$ within topology across the 54 radius–topology–metric tests. Confidence intervals are omitted from the plotted panels for readability; Supplementary Table S15 reports the post hoc matched cross-topology $QE_B$ estimates, intervals, and adjusted tests at every radius.

## 5.4 Hyperparameter Tuning and Stability

### 5.4.1 Performance Gains from Hyperparameter Tuning

Having shown that the graph-topology effects persist when radius is varied under a common tuned non-radius configuration, we next quantify the broader effect of complete topology-specific tuning relative to the untuned reference.

Fig. 10 compares the fixed tuned and untuned configurations using pooled topology-level $QE$ summaries; topology-stratified results are provided in Supplementary Fig. S5.

The fixed tuned configurations had lower values for all three pooled $QE$ metrics than the untuned references, but also had higher balanced MTR: by 24.34 tied-rank positions for hexagonal maps compared with 1.43 for MST and 1.11 for RNG. This profile-associated local-ordering trade-off was therefore much stronger for the fixed lattice and was not accompanied by lower node utilization.

The $QE$ improvement was observed in every topology family.

Table 1: Matched topology diagnostics for balanced quantization error and Mean Tied Rank. Positive paired effects favor the second topology in each contrast after applying metric directionality; lower raw values are better for both $QE_B$ and $MTR_B$. Full confidence intervals, raw paired-test p-values, split-specific metrics, node utilization, and dead-node fraction are reported in Supplementary Table S12.

| Profile | $QE_B$ | | | $MTR_B$ | | |
| --- | --- | --- | --- | --- | --- | --- |
| | MST vs hex | RNG vs hex | RNG vs MST | MST vs hex | RNG vs hex | RNG vs MST |
| Untuned | +0.269 | +0.249 | -0.020 | -0.20 | +2.44 | +2.64 |
| Tuned | +0.064 | +0.052 | -0.012 | +22.71 | +25.67 | +2.96 |

**Figure 10: Tuned vs Default Across QE Metrics**

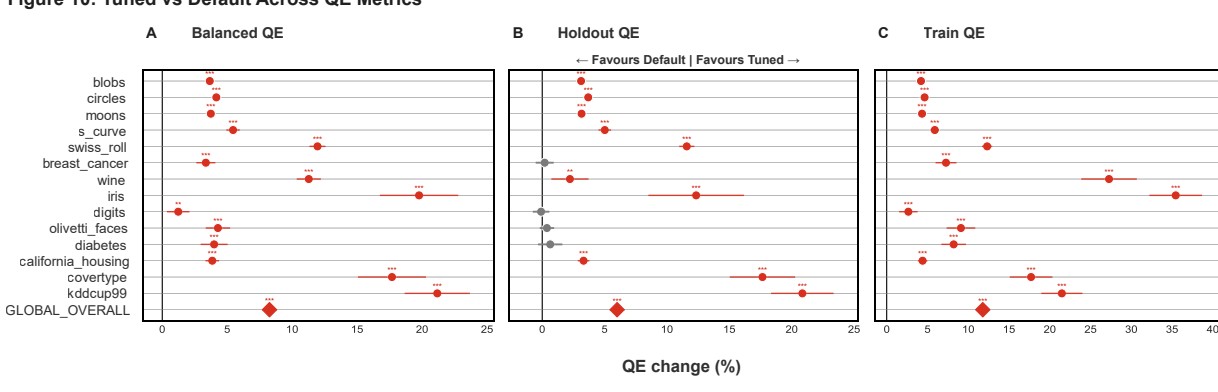

Figure 10: Tuned configuration versus untuned reference $QE$ comparison across $QE_B$, $QE_H$, and $QE_T$, pooled across all topology runs under the matched pairing keys. Positive values indicate the tuned configuration outperforms the untuned reference; the global overall row pools all matched tuned configuration/untuned reference pairs across datasets. Forest whiskers denote 95% paired $t$-test confidence intervals around the mean paired effect.

**Figure 11: Hyperparameter Stability**

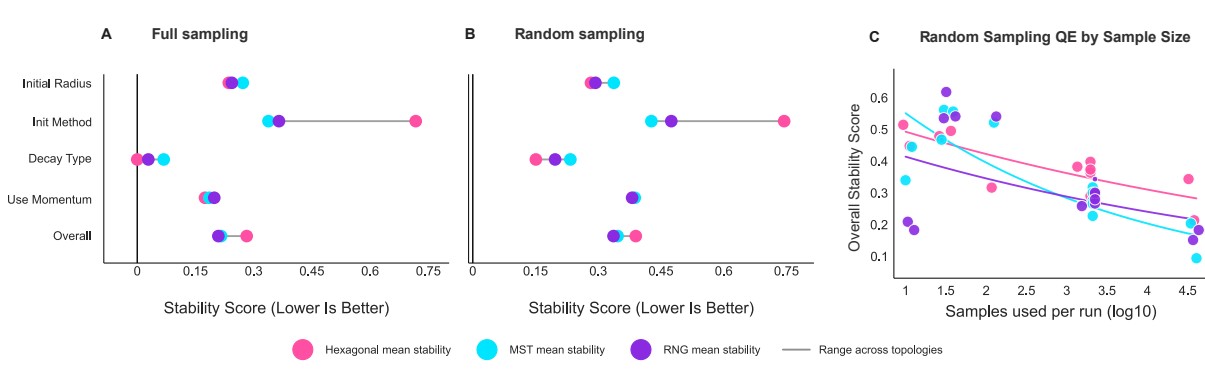

Figure 11: Hyperparameter stability by sampling mode. A: selected parameter stability under full sampling for hexagonal, MST, and RNG topologies (lower stability score is better). B: selected parameter stability under random sampling for the same topologies. C: dataset-size stability regression under random sampling, using the selected parameter stability score against sample size (log10) across the included topology families.

### 5.4.2 Hyperparameter Stability Across Topology and Sampling

To assess whether the derived hyperparameters are robust across runs and datasets, we compare the within-topology seed-to-seed tuned-parameter drift (Section 4.3.2). Overall, MST and RNG exhibit less variability than hexagonal maps under both full and random sampling (Fig. 11A-B). Across these panels, full sampling is generally the more stable setting. Mirroring the sample-size-dependent $QE$ performance, Fig. 11C shows that random-sampling hyperparameter stability also improves with increasing sample size.

### 5.5 Execution-Path Diagnostic

The quality analyses above used the in-memory pathway. Repeating the tuned diagnostics through Ray streaming with matched data and configurations produced small differences, none of which reached significance before or after Benjamini-Hochberg correction (Supplementary Table S14). The remaining differences may reflect floating-point accumulation order. The following speed benchmarks use the Ray pathway.

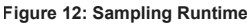

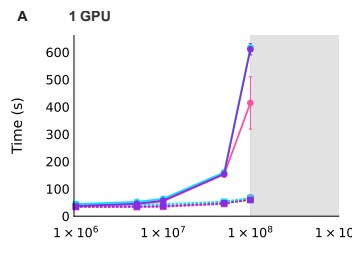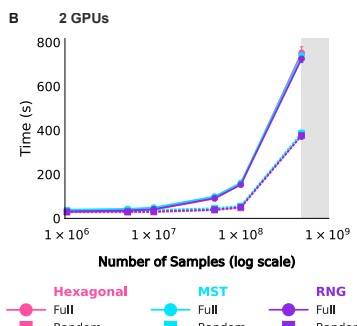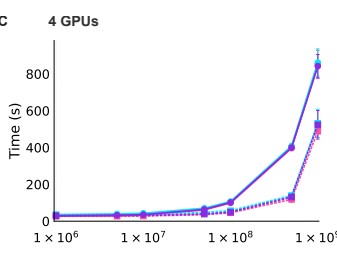

Figure 12: Sample scaling runtime comparison of full versus random sampling across $G \in \{1, 2, 4\}$ GPUs (A,B,C). Curves report mean wall-clock training time (s) under harmonized settings; error bars denote $\pm 1$ standard deviation across $n = 3$ repeated runs per configuration. Color encodes topology (hexagonal, MST, RNG), and line style encodes sampling mode (full vs. random). Shaded x-axis regions indicate sample-size ranges that could not be run in that panel relative to the shared axis maximum due to timeouts. Lower values indicate faster execution.

# 6 Speed Scaling

We next examine three aspects of FloatSOM runtime performance: the cost of full relative to random sampling, the scaling behavior obtained with additional GPUs, and the runtime implications of MST and RNG relative to a hexagonal topology.

## 6.1 Random versus Full Sampling Runtime

We report sample scaling runtime comparisons for full versus random sampling using the synthetic datasets as outlined in Section 4.2.1. We stratify by hexagonal, MST, and RNG, for 1, 2, and 4 GPUs (Fig. 12).

Across topologies, random sampling is faster than full sampling across all 1-, 2-, and 4-GPU comparisons, with similar proportional reductions at a given dataset size. With more GPUs, larger datasets can also be processed before timing out, although the runtime advantage of random over full narrows at the largest sample counts.

For the 1-GPU random sampling runs, the last successful 1-GPU run was at 100M samples, the same number of samples as run on full samples, despite random being much faster. We reason that the failed subsequent runs on larger samples were due to the transition from RAM operation to disk-backed operation. Even in random sampling mode, the implementation still stages the full dataset initially and loads full chunks before the subsampler selects the training samples. Therefore, the relevant cost is no longer only the reduced number of selected samples. We therefore treat the 1-GPU random failure at 500M samples as a disk-mode systems limitation rather than as evidence against the general random versus full runtime ordering.

## 6.2 Multi-GPU topology scaling and Larger-Than-Memory context

To examine parallel scaling, we benchmarked FloatSOM under full sampling across $G \in \{1, 2, 4, 8\}$ GPUs using workload-scaling benchmarks that extend from standard in-memory settings to larger workloads that exceed available memory.

**Figure 13: Multi-GPU Full-Batch Scaling (RNG)**

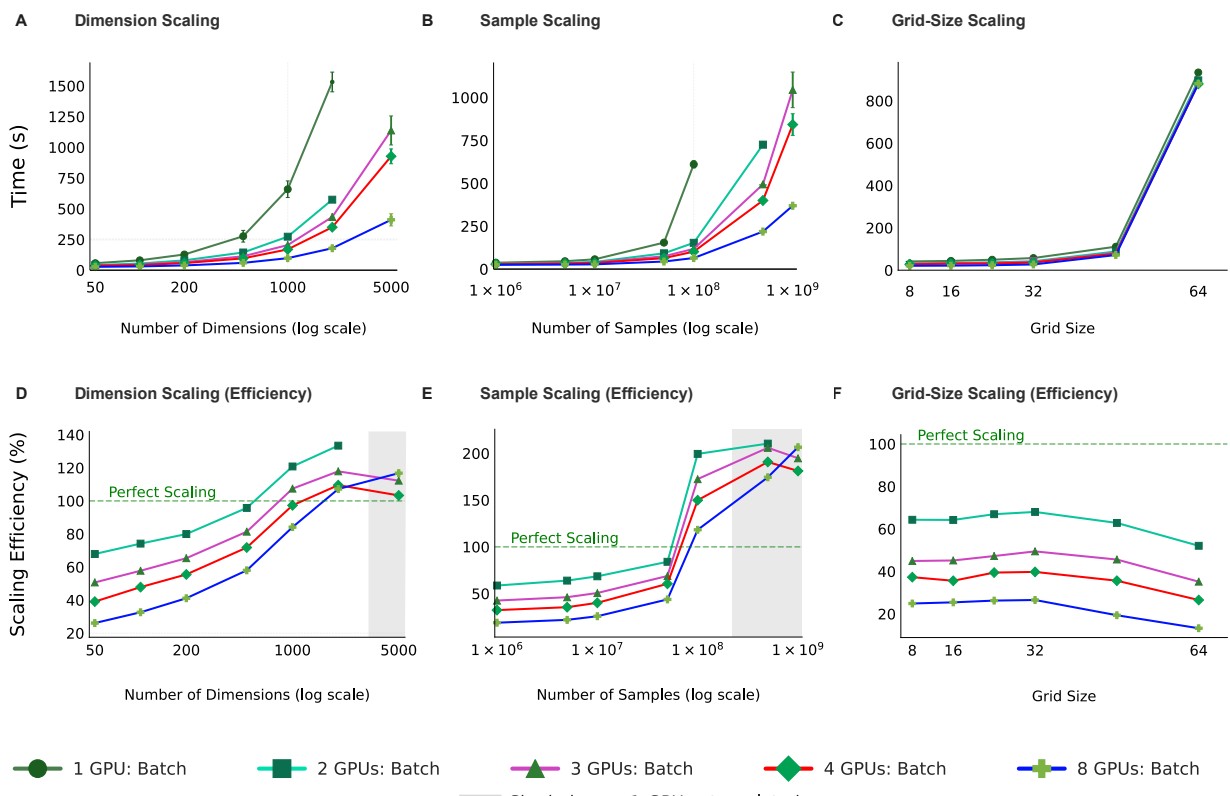

Figure 13: Multi-GPU scaling across $G \in \{1, 2, 4, 8\}$ GPUs. Panels A-C show runtime (s) for dimension, sample, and grid size scaling workloads, respectively. Panels D-F show scaling efficiency for the same workloads, computed from the single-GPU baseline and the corresponding $G$-GPU runtime. Runtime error bars denote $\pm 1$ standard deviation across $n = 3$ repeated runs per configuration; the 100% efficiency reference line indicates ideal linear scaling.

### 6.2.1   GPU Scaling and Larger-Than-Memory Runtime Performance

Fig. 13 shows that increasing GPU count improves performance in the sample scaling regime by increasing parallel throughput and delaying the transition to disk-backed execution. In the sample scaling benchmark, the 500,000,000 sample dataset requires disk backing under the 2-GPU configuration, whereas the 8-GPU configuration remains in RAM mode until the 1,000,000,000 sample dataset; when staging is still required, the disk-to-GPU path is distributed across more nodes. The 8-GPU RNG configuration processes 1,000,000,000 samples in 369.41 s (6.16 min). Note that this speed includes the time required to transfer data from shared storage to node-local shards, alongside the overhead associated with operating across multiple HPC nodes.

Grid-size scaling is the main exception: at the largest tested grid size (64), runtime decreases from 934.01 s (15.57 min) on 1 GPU to 880.83 s (14.68 min) on 8 GPUs, a 5.69% reduction. Once map-size and topology-refresh costs dominate, additional GPUs therefore contribute little further speedup.

### 6.2.2   GPU Scaling Efficiency

We next consider GPU efficiency under strong scaling, relative to ideal linear scaling. Efficiency values with directly measured 1-GPU baselines summarize observed strong scaling. Values using extrapolated denominators are shown as descriptive estimates and are not used to establish a general efficiency improve-

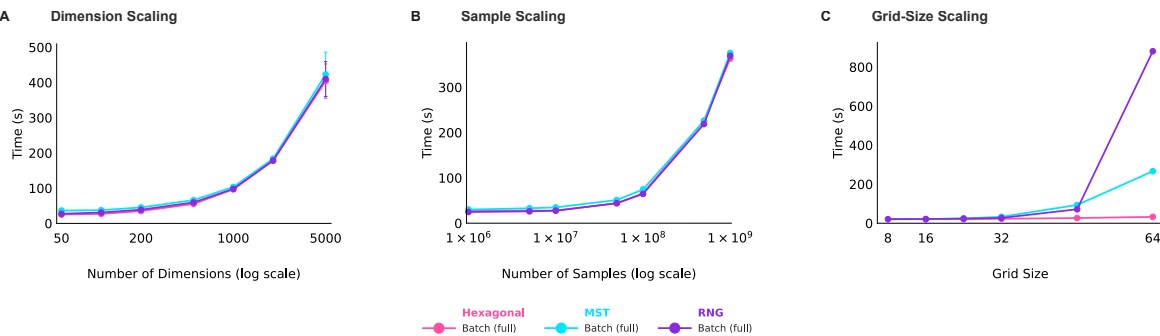

Figure 14: Topology runtime comparison at fixed $G = 8$ GPUs under full sampling. Panels A-C report mean wall-clock runtime (s) for dimension, sample, and grid size scaling workloads, respectively, with topology traces for hexagonal, MST, and RNG. Error bars denote $\pm 1$ standard deviation across $n = 3$ repeated runs per configuration. The largest-axis 8-GPU topology runtime summaries are listed in Supplementary Table S11.

ment across workload sizes. Efficiencies above 100% may reflect both parallel throughput and a changed memory/data-staging regime, rather than superlinear computation. The absolute runtime curves and directly measured-baseline comparisons therefore provide the primary scaling evidence.

### 6.3 Topology Runtime Comparisons

With that systems context in place, we next compare topology runtimes across hexagonal, MST, and RNG configurations on 8 GPUs (Fig. 14).

In Fig. 14A-B, the topologies scale similarly as input complexity and data volume increase: even at the largest tested axis values, the maximum pairwise runtime spread remains modest at dimension scaling (4.70% at 5,000 dimensions) and sample scaling (3.26% at 1,000,000,000 samples).

However, when the grid itself is enlarged in Fig. 14C, topology-dependent runtime differences become readily evident. At the largest tested grid size (grid size 64), the 8-GPU mean runtimes are 32.54 s (0.54 min) for hexagonal, 266.45 s (4.44 min) for MST, and 880.83 s (14.68 min) for RNG, corresponding to 8-GPU MST and RNG runtimes that are 8.19x and 27.07x the hexagonal runtime, respectively.

To further examine the RNG runtime spike in grid-size scaling, we compared the topology-construction operations. For each of the $\binom{P}{2}$ candidate node pairs, RNG checks every possible third node in its relative-neighborhood blocker test, giving up to $\binom{P}{2}(P-2) = O(P^3)$ pair–blocker comparisons per topology refresh. At $P = 4096$, this is approximately 34.3 billion unordered pair–blocker checks. MST instead constructs and sorts approximately $\binom{4096}{2} = 8.39$ million candidate edges using Kruskal's algorithm, an $O(P^2 \log P)$ edge-construction step after pairwise distances are calculated. Both topologies then perform the shared graph-distance and influence calculations. The additional $O(P^3)$ RNG blocker-test work therefore provides a quantitative algorithmic explanation for the sharp RNG cost increase at large grid sizes.

## 7 Final FloatSOM RNG Comparison with XPySOM

Fig. 15 compares untuned hexagonal XPySOM with tuned FloatSOM RNG and therefore combines implementation, tuning, and topology effects (Mancini et al., 2020). Sections 5.1, 5.3, and 5.4 separate these components; Supplementary Figs. S7-S8 provide the corresponding tuned hexagonal and MST comparisons.

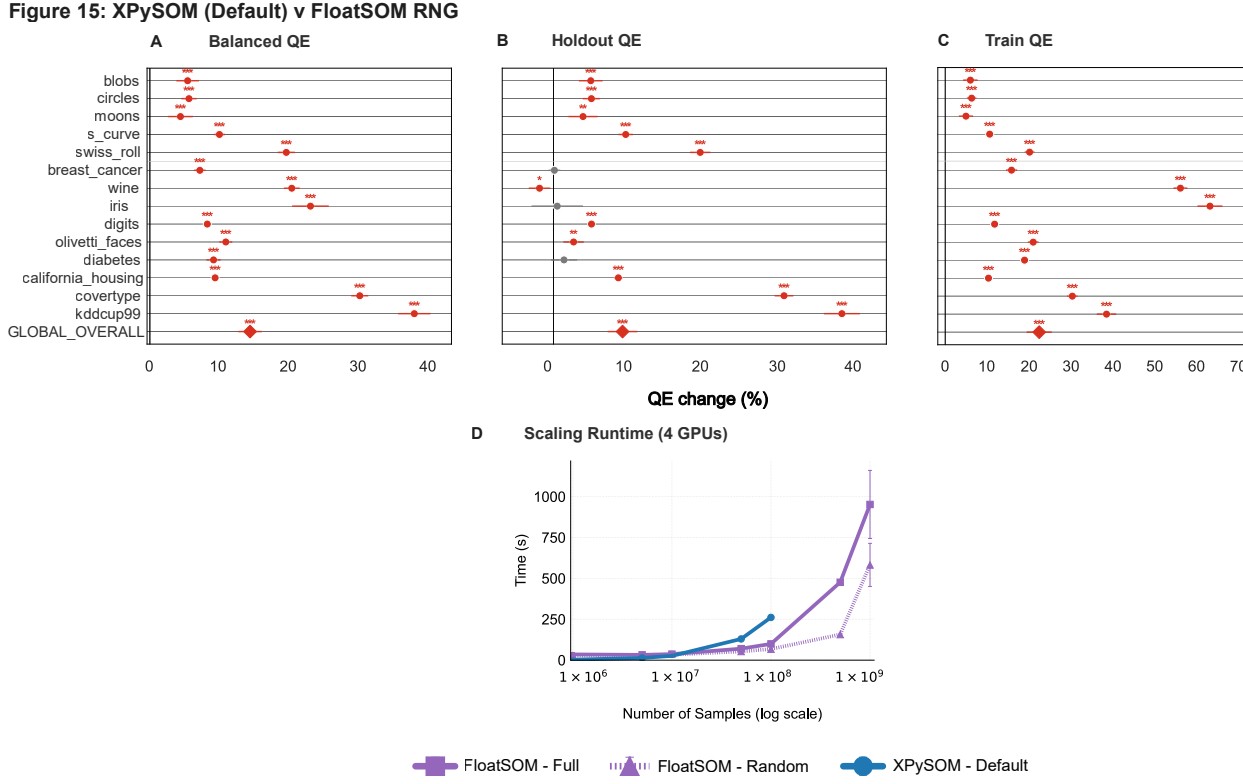

Figure 15: Integrated deployment comparison of untuned hexagonal XPySOM versus tuned FloatSOM RNG, combining implementation, tuning, and topology effects. Panels A-C compare $QE_B$, $QE_H$, and $QE_T$ under full sampling; panel D provides the scaling/runtime context.

At the overall level, Fig. 15 shows median percentage improvements of $QE_B$ (14.5%); $QE_H$ (9.1%); and $QE_T$ (22.5%) for tuned FloatSOM RNG relative to untuned hexagonal XPySOM, capturing the combined deployment effect of implementation, topology choice, and tuning on $QE$.

For the untuned hexagonal XPySOM reference in Fig. 15, workloads beyond the $10^8$-sample case were not processed because they exceeded available VRAM and XPySOM requires the full dataset to be loaded into memory. Taken together, Fig. 15 shows the point at which the workload size is large enough to warrant the extra startup overhead incurred by tuned FloatSOM RNG: tuned FloatSOM RNG delivers better $QE$ than the untuned hexagonal XPySOM baseline, while also running faster and scaling to larger workloads.

## 8 Discussion

FloatSOM combines graph-based SOM topologies with distributed, out-of-memory GPU training. The benchmarks show how sampling, topology, tuning, and workload size affect both $QE$ and runtime.

The trade-off between full and random sampling is scale dependent. Random sampling is less stable on small datasets, while above 10,000 samples we detect little paired $QE$ difference. Full sampling is therefore preferable when stability is critical; random sampling offers higher throughput on larger datasets, although disk-backed operation limits its I/O advantage.

Across the Optuna matched top-$k$ runs, RNG showed significantly better balanced and train $QE$ than MST, while no significant holdout-$QE$ difference was detected. RNG also had better overall hyperparameter stability. In the tuned defaults, no balanced-$QE$ difference was detected, possibly reflecting the relatively smaller fixed-default sample compared with the broader Optuna runs; however, these runs did readily demonstrate

RNG's superiority in topology, where RNG exhibited significantly lower MTR and better node utilization than both MST and hexagonal.

A geometric interpretation consistent with these results is that a fixed lattice imposes uniform, predetermined coupling: when a prototype moves, it can unnecessarily influence lattice neighbors that are unrelated in the learned data space, potentially displacing useful prototypes and increasing dead nodes. The apparently nonlocal hexagonal connections in the KDD Cup 99 projection (Fig. 5D) provide a visual illustration of this coupling and the associated restriction on independent prototype redistribution, subject to the distortions inherent in the two-dimensional PCA display. MST minimizes such coupling, allowing prototypes to redistribute more freely along irregular or elongated data structures. This freedom is also its limitation, because a tree cannot retain multiple locally appropriate connections where a concentrated region is better represented by a mesh. RNG occupies an intermediate position. Its nodes can influence several neighbors, but those connections arise from the evolving prototype geometry rather than being imposed before training; RNG can remain sparse where appropriate and form multiple connections in dense regions. This interpretation is supported by RNG achieving the lowest MTR and highest node utilization in Fig. 9 (Kohonen, 2013; Kangas et al., 1990; Toussaint, 1980).

$QE$ and MTR capture different properties: $QE$ measures vector-quantization fidelity, while MTR measures local ordering between the first and second BMUs. The fixed QE-tuned profiles are associated with higher MTR in every topology, but the topological cost is far larger for hexagonal maps: balanced MTR is 24.34 tied-rank positions higher than in the untuned profile, compared with 1.43 for MST and 1.11 for RNG. The matched radius sweep reinforces this distinction across the full tested range: MST and RNG maintained lower observed $QE$ than hexagonal, while RNG maintained the lowest observed MTR throughout. Thus, the QE-tuned MST and RNG profiles are associated with far lower ordering penalties than the fixed hexagonal lattice. Topology and hyperparameter choice should therefore be considered jointly rather than treating topology as a post-training visualization choice. The lower seed-to-seed variability of MST and RNG may likewise reflect their ability to rebuild connectivity from evolving node locations, unlike the fixed hexagonal lattice (Kohonen, 2013; Kangas et al., 1990).

The quality experiments use the in-memory pathway, whereas the scaling experiments use Ray. Within the distributed benchmark, scaling depends on workload size: overhead dominates small problems, while larger workloads benefit from parallel throughput and may remain in RAM when the 1-GPU case requires disk backing. Efficiencies above 100% reflect this change in memory regime rather than super-linear computation.

Regarding existing distributed SOM implementations, Somoclu and GigaSOM are important parallel-systems references. However, Somoclu was not benchmarked against because both Somoclu and SomocluGPU were already compared directly with XPySOM in the XPySOM study and found to be more than 10 times slower (Mancini et al., 2020). GigaSOM is a CPU-based algorithm implemented in Julia with perhaps the strongest demonstrated speed and size scale-ups. However, considering GigaSOM's hardware requirements, programming language difference, and dataset availability, a full systems benchmark was outside the scope of this paper. For context though, in GigaSOM's large dataset example completed a workflow that trained a 1024 node SOM on 1,167,129,317 events with 18 features in $< 25$ minutes on an 11-node, 256-core CPU cluster (Kratochvíl et al., 2020). Comparatively, FloatSOM trained a same-sized 1024-node SOM on 1,000,000,000 samples with 50 features, a dataset ~x2.4 larger than GigaSOM's, in 6.16 minutes using 8 GPUs across two HPC nodes.

Taken together, the topology results support RNG as the default topology for most datasets. This recommendation is caveated for workloads requiring very large grid sizes, and hence very large numbers of nodes. At grid size 64, hexagonal, MST, and RNG required 32.54, 266.45, and 880.83 s on 8 GPUs, respectively. RNG's graph-construction and all-pairs path calculations therefore scale much more sharply with grid size, so MST is preferable when a large graph is required and this topology overhead is prohibitive. More generally, full sampling favors stability, random sampling favors throughput on large datasets, and additional GPUs are most useful when they keep the workload in RAM.

FloatSOM brings these topology and systems choices together in a single large-scale SOM framework.

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

# A  Supplementary Methods and Tables (End Matter)

## A.1  Supplementary Methods S1. Execution-path concordance protocol

The execution-path analysis used all 14 datasets, the 10 shared seeds 42–51, full sampling, and the hexagonal, MST, and RNG topologies on one NVIDIA V100 GPU. For each dataset–seed–topology unit, local CuPy and Ray streaming received the same standardized data, deterministic train–holdout split, topology-specific tuned parameters from Section 4.3.1, and evaluation metrics; execution profile was the only deliberately changed factor. Differences were defined as Ray streaming minus local CuPy.

For each topology, Supplementary Table S14 reports paired differences for balanced $QE$, MTR, node utilization, and dead-node fraction across 140 matched dataset–seed units. Mean differences and 95% confidence intervals were calculated from two-sided one-sample paired $t$-tests against zero, with Benjamini–Hochberg correction applied jointly across the 12 topology–metric tests.

## A.2  Supplementary Tables

Supplementary Table S1: Dataset metadata and numbered point key for the Figure 4 sampling mode analysis.

| dataset_index | dataset | dimension_count | sample_size | dataset_type |
|---|---|---|---|---|
| 1 | iris | 4 | 150 | real |
| 2 | wine | 13 | 178 | real |
| 3 | olivetti_faces | 4096 | 400 | real |
| 4 | diabetes | 10 | 442 | real |
| 5 | breast_cancer | 30 | 569 | real |
| 6 | digits | 64 | 1797 | real |
| 7 | california_housing | 8 | 20640 | real |
| 8 | blobs | 2 | 30000 | synthetic |
| 9 | circles | 2 | 30000 | synthetic |
| 10 | moons | 2 | 30000 | synthetic |
| 11 | s_curve | 3 | 30000 | synthetic |
| 12 | swiss_roll | 3 | 30000 | synthetic |
| 13 | kddcup99 | 41 | 494021 | real |
| 14 | covertype | 54 | 581012 | real |

Supplementary Table S2: HDSSSOM pilot configuration summary for Figure 3.

| field | value |
|---|---|
| pilot purpose | Initial HDSSSOM screening before the broader sampling comparison |
| configuration scope | Smaller hexagonal Optuna pilot |
| datasets | swiss_roll, moons, circles, blobs, s_curve, breast_cancer, wine, iris, digits, olivetti_faces |
| dataset count | 10 |

Supplementary Table S2 (continued)

| field | value |
|---|---|
| seed count | 5 |
| sampling methods present | full, random, HDSSSOM |
| topology | `hexagonal` |
| algorithm family in campaign | `full_batch` |
| optimization split setup | `evaluation-split=both`, yielding `QE_H` and `QE_T`; Figure 3 reports paired `QE_B` |
| trials per scenario | 200 |
| main text figure slice | hexagonal / full vs HDSSSOM |

Supplementary Table S3: Exact axis values used for the sample count, feature dimension, and grid size scaling benchmarks in Section 4.2.1.

| scaling axis | exact values |
|---|---|
| sample count | $10^6$, $5 \times 10^6$, $10^7$, $5 \times 10^7$, $10^8$, $5 \times 10^8$, $10^9$ |
| feature dimension | 50, 100, 200, 500, 1000, 2000, 5000 |
| grid side length | 8, 16, 24, 32, 48, 64 |

Supplementary Table S4: Untuned FloatSOM MST versus hexagonal XPySOM.

The `dataset_index` column matches the numbered points in Supplementary Figure S1 panel D. Positive percentage values and positive signed effects favor FloatSOM; the compact embedded table lists wins as FloatSOM/XPySOM/ties.

| idx | dataset | metric | split | wins F/X/tie | median % | mean % | 95% CI | p | n | signed effect |
|---|---|---|---|---|---|---|---|---|---|---|
| 8 | blobs | QE | train | 0/10/0 | -3.6521 | -3.5612 | [-4.6291, -2.687] | 0.002 | 10 | -1 |
| 5 | breast_cancer | QE | train | 10/0/0 | 6.0461 | 5.7839 | [5.0244, 6.3682] | 0.002 | 10 | 1 |
| 7 | california_ housing | QE | train | 10/0/0 | 4.2199 | 4.3194 | [3.7617, 4.9195] | 0.002 | 10 | 1 |
| 9 | circles | QE | train | 0/10/0 | -1.7629 | -2.1879 | [-3.655, -1.0873] | 0.002 | 10 | -1 |
| 14 | covertype | QE | train | 10/0/0 | 15.3837 | 15.3479 | [13.9347, 16.925] | 0.002 | 10 | 1 |
| 4 | diabetes | QE | train | 10/0/0 | 7.0091 | 7.0306 | [6.1295, 7.9882] | 0.002 | 10 | 1 |
| 6 | digits | QE | train | 10/0/0 | 5.3038 | 5.3987 | [5.0889, 5.8056] | 0.002 | 10 | 1 |
| 1 | iris | QE | train | 10/0/0 | 28.4174 | 28.3671 | [24.2551, 32.3913] | 0.002 | 10 | 1 |
| 13 | kddcup99 | QE | train | 10/0/0 | 19.4133 | 19.693 | [16.1432, 23.2133] | 0.002 | 10 | 1 |
| 10 | moons | QE | train | 1/9/0 | -3.7058 | -3.3568 | [-5.5241, -1.575] | 0.0098 | 10 | -0.8 |
| 3 | olivetti_faces | QE | train | 10/0/0 | 8.4513 | 8.5474 | [7.1779, 9.6246] | 0.002 | 10 | 1 |
| 11 | s_curve | QE | train | 10/0/0 | 3.3827 | 3.4166 | [1.906, 5.1934] | 0.002 | 10 | 1 |
| 12 | swiss_roll | QE | train | 10/0/0 | 5.271 | 5.0983 | [3.8948, 6.3949] | 0.002 | 10 | 1 |
| 2 | wine | QE | train | 10/0/0 | 20.0992 | 20.1816 | [18.2671, 22.1871] | 0.002 | 10 | 1 |
| - | GLOBAL | QE | train | 111/29/0 | 6.8961 | 8.1485 | [5.6474, 9.0003] | 1.49e-17 | 140 | 0.5857 |
| 8 | blobs | QE | holdout | 0/10/0 | -4.1422 | -3.9793 | [-5.2037, -2.9583] | 0.002 | 10 | -1 |
| 5 | breast_cancer | QE | holdout | 2/8/0 | -0.6423 | -0.4735 | [-1.2282, 0.3513] | 0.1309 | 10 | -0.6 |
| 7 | california_ housing | QE | holdout | 10/0/0 | 3.8135 | 3.8367 | [3.1391, 4.5531] | 0.002 | 10 | 1 |
| 9 | circles | QE | holdout | 0/10/0 | -2.5762 | -3.1314 | [-4.9792, -1.4919] | 0.002 | 10 | -1 |
| 14 | covertype | QE | holdout | 10/0/0 | 15.4626 | 15.3225 | [13.8381, 16.8307] | 0.002 | 10 | 1 |
| 4 | diabetes | QE | holdout | 4/6/0 | -0.1376 | 0.0531 | [-1.2671, 1.4709] | 1 | 10 | -0.2 |
| 6 | digits | QE | holdout | 10/0/0 | 2.5919 | 2.7034 | [2.2909, 3.2567] | 0.002 | 10 | 1 |
| 1 | iris | QE | holdout | 4/6/0 | 0.1297 | -0.4991 | [-4.9164, 4.9562] | 1 | 10 | -0.2 |
| 13 | kddcup99 | QE | holdout | 10/0/0 | 19.2029 | 19.4801 | [15.8968, 23.1041] | 0.002 | 10 | 1 |
| 10 | moons | QE | holdout | 1/9/0 | -3.6371 | -3.4648 | [-5.2819, -1.8559] | 0.0059 | 10 | -0.8 |

Supplementary Table S4 (continued)

| idx | dataset | metric | split | wins F/X/tie | median % | mean % | 95% CI | p | n | signed effect |
|---|---|---|---|---|---|---|---|---|---|---|
| 3 | olivetti_faces | QE | holdout | 10/0/0 | 1.915 | 1.8944 | [1.55, 2.2547] | 0.002 | 10 | 1 |
| 11 | s_curve | QE | holdout | 10/0/0 | 3.1294 | 3.2429 | [1.9458, 4.4824] | 0.002 | 10 | 1 |
| 12 | swiss_roll | QE | holdout | 10/0/0 | 5.1413 | 5.1294 | [3.9965, 6.4501] | 0.002 | 10 | 1 |
| 2 | wine | QE | holdout | 7/3/0 | 0.1337 | -0.0378 | [-1.2204, 0.9745] | 0.8457 | 10 | 0.4 |
| - | GLOBAL | QE | holdout | 88/52/0 | 1.7869 | 2.8626 | [0.8691, 2.7426] | 6.33e-05 | 140 | 0.2571 |
| 8 | blobs | QE_B | both | 0/10/0 | -3.91 | -3.7703 | [-4.93, -2.7417] | 0.002 | 10 | -1 |
| 5 | breast_cancer | QE_B | both | 10/0/0 | 2.4588 | 2.4478 | [1.8439, 2.987] | 0.002 | 10 | 1 |
| 7 | california_ housing | QE_B | both | 10/0/0 | 3.9644 | 4.0771 | [3.6418, 4.7324] | 0.002 | 10 | 1 |
| 9 | circles | QE_B | both | 0/10/0 | -2.0639 | -2.6605 | [-4.3931, -1.3869] | 0.002 | 10 | -1 |
| 14 | covertype | QE_B | both | 10/0/0 | 15.3975 | 15.3352 | [13.8864, 16.8854] | 0.002 | 10 | 1 |
| 4 | diabetes | QE_B | both | 10/0/0 | 3.1657 | 3.1796 | [2.2103, 4.1719] | 0.002 | 10 | 1 |
| 6 | digits | QE_B | both | 10/0/0 | 3.997 | 4.0139 | [3.6762, 4.3228] | 0.002 | 10 | 1 |
| 1 | iris | QE_B | both | 10/0/0 | 9.1407 | 9.5616 | [5.9156, 13.6397] | 0.002 | 10 | 1 |
| 13 | kddcup99 | QE_B | both | 10/0/0 | 19.3064 | 19.5854 | [16.0187, 23.1588] | 0.002 | 10 | 1 |
| 10 | moons | QE_B | both | 1/9/0 | -3.4326 | -3.4102 | [-5.2683, -1.7163] | 0.0059 | 10 | -0.8 |
| 3 | olivetti_faces | QE_B | both | 10/0/0 | 4.8444 | 4.8685 | [4.365, 5.4342] | 0.002 | 10 | 1 |
| 11 | s_curve | QE_B | both | 10/0/0 | 3.292 | 3.3299 | [1.8567, 4.777] | 0.002 | 10 | 1 |
| 12 | swiss_roll | QE_B | both | 10/0/0 | 5.2033 | 5.1138 | [4.0306, 6.4161] | 0.002 | 10 | 1 |
| 2 | wine | QE_B | both | 10/0/0 | 7.8522 | 7.7725 | [6.9044, 8.686] | 0.002 | 10 | 1 |
| - | GLOBAL | QE_B | both | 111/29/0 | 4.2807 | 4.9603 | [3.4978, 5.2317] | 1.35e-13 | 140 | 0.5857 |
| 8 | blobs | train time | train | 0/10/0 | -730.1165 | -757.4247 | [-864.8227, -727.7904] | 0.002 | 10 | -1 |
| 5 | breast_cancer | train time | train | 0/10/0 | -1011.2684 | -1004.5725 | [-1016.3642, -976.4259] | 0.002 | 10 | -1 |
| 7 | california_ housing | train time | train | 0/10/0 | -972.0213 | -950.4554 | [-1011.7249, -888.6339] | 0.002 | 10 | -1 |
| 9 | circles | train time | train | 0/10/0 | -728.984 | -728.8823 | [-733.0182, -724.2827] | 0.002 | 10 | -1 |
| 14 | covertype | train time | train | 0/10/0 | -37.4988 | -37.8968 | [-42.3539, -32.87] | 0.002 | 10 | -1 |
| 4 | diabetes | train time | train | 0/10/0 | -1010.0585 | -1043.324 | [-1183.0245, -1002.8581] | 0.002 | 10 | -1 |

Continued on next page

| idx | dataset | metric | split | wins F/X/tie | median % | mean % | 95% CI | p | n | signed effect |
|---|---|---|---|---|---|---|---|---|---|---|
| 6 | digits | train time | train | 0/10/0 | -1004.8808 | -1005.1876 | [-1011.5931, -998.5865] | 0.002 | 10 | -1 |
| 1 | iris | train time | train | 0/10/0 | -1037.1907 | -1036.0797 | [-1042.8531, -1029.1162] | 0.002 | 10 | -1 |
| 13 | kddcup99 | train time | train | 0/10/0 | -36.9051 | -35.1756 | [-38.6464, -32.0512] | 0.002 | 10 | -1 |
| 10 | moons | train time | train | 0/10/0 | -734.37 | -734.8929 | [-740.0781, -731.5506] | 0.002 | 10 | -1 |
| 3 | olivetti_faces | train time | train | 0/10/0 | -1019.8923 | -1016.4567 | [-1030.2974, -1000.3188] | 0.002 | 10 | -1 |
| 11 | s_curve | train time | train | 0/10/0 | -736.215 | -736.5435 | [-741.9456, -732.008] | 0.002 | 10 | -1 |
| 12 | swiss_roll | train time | train | 1/9/0 | -726.5489 | -649.8592 | [-732.2277, -336.7629] | 0.0039 | 10 | -0.8 |
| 2 | wine | train time | train | 0/10/0 | -1030.4764 | -1064.4839 | [-1204.4846, -1025.0949] | 0.002 | 10 | -1 |
| - | GLOBAL | train time | train | 1/139/0 | -868.8976 | -771.5168 | [-875.2748, -760.3798] | 1.59e-24 | 140 | -0.9857 |

Under review as submission to TMLR

Supplementary Table S5: Untuned FloatSOM RNG versus hexagonal XPySOM.

The `dataset_index` column matches the numbered points in Supplementary Figure S2 panel D. Positive percentage values and positive signed effects favor FloatSOM; the compact embedded table lists wins as FloatSOM/XPySOM/ties.

| idx | dataset | metric | split | wins F/X/tie | median % | mean % | 95% CI | p | n | signed effect |
|---|---|---|---|---|---|---|---|---|---|---|
| 8 | blobs | QE | train | 0/10/0 | -2.7298 | -2.6927 | [-3.7585, -1.6597] | 0.002 | 10 | -1 |
| 5 | breast_cancer | QE | train | 10/0/0 | 5.4998 | 5.465 | [5.0315, 5.9377] | 0.002 | 10 | 1 |
| 7 | california_housing | QE | train | 10/0/0 | 3.6187 | 3.641 | [3.0746, 4.2359] | 0.002 | 10 | 1 |
| 9 | circles | QE | train | 1/9/0 | -1.7013 | -2.1091 | [-3.9805, -0.3778] | 0.0137 | 10 | -0.8 |
| 14 | covertype | QE | train | 10/0/0 | 13.6239 | 13.5016 | [10.9438, 15.887] | 0.002 | 10 | 1 |
| 4 | diabetes | QE | train | 10/0/0 | 6.2882 | 6.3332 | [5.2073, 7.5187] | 0.002 | 10 | 1 |
| 6 | digits | QE | train | 10/0/0 | 4.4749 | 4.5437 | [4.073, 5.0017] | 0.002 | 10 | 1 |
| 1 | iris | QE | train | 10/0/0 | 25.6985 | 25.9471 | [20.7469, 31.2163] | 0.002 | 10 | 1 |
| 13 | kddcup99 | QE | train | 10/0/0 | 18.8172 | 18.2227 | [14.5027, 22.0324] | 0.002 | 10 | 1 |
| 10 | moons | QE | train | 2/8/0 | -2.6397 | -2.3844 | [-4.0907, -0.5701] | 0.0195 | 10 | -0.6 |
| 3 | olivetti_faces | QE | train | 10/0/0 | 6.9757 | 7.4422 | [5.9654, 9.3791] | 0.002 | 10 | 1 |
| 11 | s_curve | QE | train | 10/0/0 | 3.0675 | 2.9986 | [1.5732, 4.5318] | 0.002 | 10 | 1 |
| 12 | swiss_roll | QE | train | 10/0/0 | 5.7819 | 5.3025 | [4.1229, 6.2356] | 0.002 | 10 | 1 |
| 2 | wine | QE | train | 10/0/0 | 18.7804 | 18.6217 | [15.9669, 21.416] | 0.002 | 10 | 1 |
| - | GLOBAL | QE | train | 113/27/0 | 6.1574 | 7.4881 | [5.0617, 8.1204] | 5.8e-18 | 140 | 0.6143 |
| 8 | blobs | QE | holdout | 0/10/0 | -3.0403 | -2.9146 | [-3.8854, -1.8988] | 0.002 | 10 | -1 |
| 5 | breast_cancer | QE | holdout | 3/7/0 | -0.1679 | -0.0809 | [-0.6344, 0.5026] | 0.4316 | 10 | -0.4 |
| 7 | california_housing | QE | holdout | 10/0/0 | 3.5473 | 3.4616 | [2.8161, 3.9656] | 0.002 | 10 | 1 |
| 9 | circles | QE | holdout | 2/8/0 | -2.435 | -2.5817 | [-4.2591, -0.7108] | 0.0098 | 10 | -0.6 |
| 14 | covertype | QE | holdout | 10/0/0 | 13.5339 | 13.4773 | [11.0872, 15.767] | 0.002 | 10 | 1 |
| 4 | diabetes | QE | holdout | 8/2/0 | 0.993 | 0.9845 | [0.0524, 1.9348] | 0.0488 | 10 | 0.6 |
| 6 | digits | QE | holdout | 10/0/0 | 2.6213 | 2.6133 | [2.0362, 3.1169] | 0.002 | 10 | 1 |
| 1 | iris | QE | holdout | 3/7/0 | -1.3579 | -1.2948 | [-4.6315, 2.0699] | 0.375 | 10 | -0.4 |
| 13 | kddcup99 | QE | holdout | 10/0/0 | 18.8346 | 18.178 | [14.6321, 21.9247] | 0.002 | 10 | 1 |
| 10 | moons | QE | holdout | 1/9/0 | -2.9592 | -2.5631 | [-4.2128, -0.9289] | 0.0137 | 10 | -0.8 |

Supplementary Table S5 (continued)

| idx | dataset | metric | split | wins F/X/tie | median % | mean % | 95% CI | p | n | signed effect |
|---|---|---|---|---|---|---|---|---|---|---|
| 3 | olivetti_faces | QE | holdout | 8/2/0 | 1.5548 | 1.5684 | [0.0193, 3.2179] | 0.0371 | 10 | 0.6 |
| 11 | s_curve | QE | holdout | 10/0/0 | 2.8467 | 3.0331 | [1.8244, 4.3726] | 0.002 | 10 | 1 |
| 12 | swiss_roll | QE | holdout | 10/0/0 | 5.5918 | 5.3626 | [3.9067, 6.5351] | 0.002 | 10 | 1 |
| 2 | wine | QE | holdout | 5/5/0 | -0.0877 | -0.2419 | [-1.2255, 0.6779] | 0.6953 | 10 | 0 |
| - | GLOBAL | QE | holdout | 90/50/0 | 1.7513 | 2.7858 | [0.9907, 2.6251] | 1.02e-05 | 140 | 0.2857 |
| 8 | blobs | QE_B | both | 0/10/0 | -2.8719 | -2.8032 | [-3.8029, -1.9134] | 0.002 | 10 | -1 |
| 5 | breast_cancer | QE_B | both | 10/0/0 | 2.4644 | 2.4959 | [2.075, 2.9683] | 0.002 | 10 | 1 |
| 7 | california_housing | QE_B | both | 10/0/0 | 3.6308 | 3.5512 | [2.9851, 4.109] | 0.002 | 10 | 1 |
| 9 | circles | QE_B | both | 2/8/0 | -2.1386 | -2.3461 | [-3.9949, -0.5829] | 0.0098 | 10 | -0.6 |
| 14 | covertype | QE_B | both | 10/0/0 | 13.6109 | 13.4894 | [11.0155, 15.826] | 0.002 | 10 | 1 |
| 4 | diabetes | QE_B | both | 10/0/0 | 3.3183 | 3.3817 | [2.369, 4.1991] | 0.002 | 10 | 1 |
| 6 | digits | QE_B | both | 10/0/0 | 3.5551 | 3.5512 | [3.034, 3.9851] | 0.002 | 10 | 1 |
| 1 | iris | QE_B | both | 10/0/0 | 7.914 | 8.1514 | [4.7671, 10.8707] | 0.002 | 10 | 1 |
| 13 | kddcup99 | QE_B | both | 10/0/0 | 18.8261 | 18.1997 | [14.5584, 21.9789] | 0.002 | 10 | 1 |
| 10 | moons | QE_B | both | 2/8/0 | -2.8707 | -2.4735 | [-4.1471, -0.7217] | 0.0195 | 10 | -0.6 |
| 3 | olivetti_faces | QE_B | both | 10/0/0 | 4.1052 | 4.1971 | [2.6366, 5.7848] | 0.002 | 10 | 1 |
| 11 | s_curve | QE_B | both | 10/0/0 | 2.9385 | 3.0161 | [1.6726, 4.3754] | 0.002 | 10 | 1 |
| 12 | swiss_roll | QE_B | both | 10/0/0 | 5.7209 | 5.3327 | [4.0146, 6.334] | 0.002 | 10 | 1 |
| 2 | wine | QE_B | both | 10/0/0 | 7.1548 | 7.0442 | [5.7434, 8.3772] | 0.002 | 10 | 1 |
| - | GLOBAL | QE_B | both | 114/26/0 | 3.9446 | 4.6277 | [3.2069, 4.8515] | 1.27e-14 | 140 | 0.6286 |
| 8 | blobs | train time | train | 0/10/0 | -725.754 | -726.3513 | [-729.2884, -724.0566] | 0.002 | 10 | -1 |
| 5 | breast_cancer | train time | train | 0/10/0 | -1028.7409 | -1021.2214 | [-1033.9516, -990.4514] | 0.002 | 10 | -1 |
| 7 | california_housing | train time | train | 0/10/0 | -959.1137 | -935.4774 | [-997.3973, -863.1072] | 0.002 | 10 | -1 |
| 9 | circles | train time | train | 0/10/0 | -728.9236 | -729.3583 | [-734.228, -723.7248] | 0.002 | 10 | -1 |
| 14 | covertype | train time | train | 0/10/0 | -38.0038 | -37.1495 | [-41.0157, -32.7859] | 0.002 | 10 | -1 |
| 4 | diabetes | train time | train | 0/10/0 | -1029.3356 | -1028.7558 | [-1034.5884, -1022.6478] | 0.002 | 10 | -1 |

| idx | dataset | metric | split | wins F/X/tie | median % | mean % | 95% CI | p | n | signed effect |
|---|---|---|---|---|---|---|---|---|---|---|
| 6 | digits | train time | train | 0/10/0 | -1029.1819 | -1028.6713 | [-1034.196, -1022.846] | 0.002 | 10 | -1 |
| 1 | iris | train time | train | 0/10/0 | -1030.5871 | -1029.7524 | [-1035.7818, -1023.3311] | 0.002 | 10 | -1 |
| 13 | kddcup99 | train time | train | 0/10/0 | -36.6803 | -35.3473 | [-39.0245, -32.0055] | 0.002 | 10 | -1 |
| 10 | moons | train time | train | 0/10/0 | -731.1551 | -730.6761 | [-733.0006, -727.7656] | 0.002 | 10 | -1 |
| 3 | olivetti_faces | train time | train | 0/10/0 | -1054.7323 | -1050.5264 | [-1064.173, -1034.3248] | 0.002 | 10 | -1 |
| 11 | s_curve | train time | train | 0/10/0 | -749.2016 | -800.7466 | [-1011.2814, -744.996] | 0.002 | 10 | -1 |
| 12 | swiss_roll | train time | train | 1/9/0 | -741.1606 | -661.2161 | [-745.7479, -339.5738] | 0.0039 | 10 | -0.8 |
| 2 | wine | train time | train | 0/10/0 | -1040.5015 | -1039.5076 | [-1043.8579, -1034.1717] | 0.002 | 10 | -1 |
| - | GLOBAL | train time | train | 1/139/0 | -877.7086 | -775.3398 | [-884.2881, -753.573] | 1.59e-24 | 140 | -0.9857 |

Supplementary Table S6: Hexagonal FloatSOM–XPySOM implementation calibration.

The `dataset_index` column matches the numbered points in Supplementary Figure S3 panel D. Positive percentage values and positive signed effects favor FloatSOM; the compact embedded table lists wins as FloatSOM/XPySOM/ties.

| idx | dataset | metric | split | wins F/X/tie | median % | mean % | 95% CI | p | n | signed effect |
|---|---|---|---|---|---|---|---|---|---|---|
| 8 | blobs | QE | train | 7/3/0 | 0.0042 | 0.0034 | [-0.0023, 0.0089] | 0.1602 | 10 | 0.4 |
| 5 | breast_cancer | QE | train | 1/1/8 | 1.46e-08 | 2.92e-09 | [-1.04e-05, 1.04e-05] | 1 | 10 | 0 |
| 7 | california_ housing | QE | train | 7/1/2 | 1.13e-05 | 0.0016 | [-0.0046, 0.0095] | 0.1484 | 10 | 0.75 |
| 9 | circles | QE | train | 6/4/0 | -0.0156 | -0.0142 | [-0.0446, 0.0107] | 0.5566 | 10 | 0.2 |
| 14 | covertype | QE | train | 3/7/0 | -0.000333 | -0.0021 | [-0.0152, 0.0059] | 0.3223 | 10 | -0.4 |
| 4 | diabetes | QE | train | 2/2/6 | -8.76e-08 | -3.5e-08 | [-8.88e-06, 8.68e-06] | 0.625 | 10 | 0 |
| 6 | digits | QE | train | 0/2/8 | -1.08e-05 | -2.16e-06 | [-1.08e-05, -1.08e-05] | 0.5 | 10 | -1 |
| 1 | iris | QE | train | 3/1/6 | 7.85e-06 | 1.66e-06 | [-7.64e-06, 8.43e-06] | 0.25 | 10 | 0.5 |
| 13 | kddcup99 | QE | train | 4/6/0 | -0.037 | -0.2112 | [-0.5893, 0.0103] | 0.1309 | 10 | -0.2 |
| 10 | moons | QE | train | 8/2/0 | 0.0021 | 0.0114 | [-0.000313, 0.0481] | 0.084 | 10 | 0.6 |
| 3 | olivetti_faces | QE | train | 2/4/4 | -0.6875 | -0.3985 | [-1.5487, 0.3296] | 0.2188 | 10 | -0.3333 |
| 11 | s_curve | QE | train | 9/1/0 | 0.0024 | 0.0041 | [1.78e-05, 0.0092] | 0.0195 | 10 | 0.8 |
| 12 | swiss_roll | QE | train | 6/4/0 | 0.0019 | 0.0019 | [-0.0046, 0.0087] | 0.625 | 10 | 0.2 |
| 2 | wine | QE | train | 0/1/9 | -1.02e-05 | -1.02e-06 | [-1.02e-05, -1.02e-05] | 1 | 10 | -1 |
| - | GLOBAL | QE | train | 58/39/43 | 1.51e-05 | -0.0431 | [-0.000899, 0.0016] | 0.484 | 140 | 0.1959 |
| 8 | blobs | QE | holdout | 7/3/0 | 0.0026 | 0.0032 | [-0.001, 0.0074] | 0.2324 | 10 | 0.4 |
| 5 | breast_cancer | QE | holdout | 1/1/8 | -2.38e-07 | -4.76e-08 | [-9.3e-06, 8.82e-06] | 1 | 10 | 0 |
| 7 | california_ housing | QE | holdout | 7/2/1 | 7.32e-06 | -0.0011 | [-0.0113, 0.0069] | 0.4961 | 10 | 0.5556 |
| 9 | circles | QE | holdout | 6/4/0 | 0.0017 | 0.0011 | [-0.0504, 0.0522] | 0.9219 | 10 | 0.2 |
| 14 | covertype | QE | holdout | 3/7/0 | -0.000325 | -0.0022 | [-0.0154, 0.0059] | 0.3223 | 10 | -0.4 |
| 4 | diabetes | QE | holdout | 0/2/8 | -7.11e-06 | -1.42e-06 | [-7.17e-06, -7.05e-06] | 0.5 | 10 | -1 |
| 6 | digits | QE | holdout | 1/1/8 | 6.39e-08 | 1.28e-08 | [-1.05e-05, 1.06e-05] | 1 | 10 | 0 |
| 1 | iris | QE | holdout | 4/2/4 | 5.77e-06 | 3.63e-06 | [-8.4e-06, 1.94e-05] | 0.4375 | 10 | 0.3333 |
| 13 | kddcup99 | QE | holdout | 4/6/0 | -0.0381 | -0.2277 | [-0.6099, 0.0098] | 0.1602 | 10 | -0.2 |
| 10 | moons | QE | holdout | 6/3/1 | 0.0026 | 0.0049 | [-0.009, 0.0308] | 0.4961 | 10 | 0.3333 |

| idx | dataset | metric | split | wins F/X/tie | median % | mean % | 95% CI | p | n | signed effect |
|---|---|---|---|---|---|---|---|---|---|---|
| 3 | olivetti_faces | QE | holdout | 4/2/4 | 0.2291 | 0.1813 | [-0.3845, 1.4392] | 0.3125 | 10 | 0.3333 |
| 11 | s_curve | QE | holdout | 10/0/0 | 0.003 | 0.0034 | [0.000108, 0.0061] | 0.002 | 10 | 1 |
| 12 | swiss_roll | QE | holdout | 6/4/0 | 0.000823 | 0.0011 | [-0.0056, 0.007] | 0.625 | 10 | 0.2 |
| 2 | wine | QE | holdout | 4/1/5 | 6.17e-06 | 2.54e-06 | [-1.18e-05, 1.31e-05] | 0.4375 | 10 | 0.6 |
| - | GLOBAL | QE | holdout | 63/38/39 | 0.000105 | -0.0026 | [-0.000154, 0.0021] | 0.1633 | 140 | 0.2475 |
| 8 | blobs | QE_B | both | 8/2/0 | 0.0025 | 0.0033 | [0.000283, 0.0064] | 0.0488 | 10 | 0.6 |
| 5 | breast_cancer | QE_B | both | 2/2/6 | -1.11e-08 | -4.43e-09 | [-4.92e-06, 4.92e-06] | 1 | 10 | 0 |
| 7 | california_ housing | QE_B | both | 9/1/0 | 7.42e-06 | 0.000257 | [-0.0069, 0.0082] | 0.0645 | 10 | 0.8 |
| 9 | circles | QE_B | both | 6/4/0 | -0.0012 | -0.0065 | [-0.0372, 0.0231] | 0.9219 | 10 | 0.2 |
| 14 | covertype | QE_B | both | 3/7/0 | -0.000329 | -0.0022 | [-0.0153, 0.0059] | 0.3223 | 10 | -0.4 |
| 4 | diabetes | QE_B | both | 1/3/6 | -3.92e-06 | -7.82e-07 | [-3.93e-06, 3.95e-06] | 0.875 | 10 | -0.5 |
| 6 | digits | QE_B | both | 1/2/7 | -3.98e-06 | -1.06e-06 | [-1.06e-05, 5.35e-06] | 0.75 | 10 | -0.3333 |
| 1 | iris | QE_B | both | 4/2/4 | 4.93e-06 | 2.86e-06 | [-7.76e-06, 1.4e-05] | 0.2188 | 10 | 0.3333 |
| 13 | kddcup99 | QE_B | both | 4/6/0 | -0.0376 | -0.2194 | [-0.5996, 0.0099] | 0.1602 | 10 | -0.2 |
| 10 | moons | QE_B | both | 7/3/0 | 0.0018 | 0.0081 | [-0.003, 0.0377] | 0.2754 | 10 | 0.4 |
| 3 | olivetti_faces | QE_B | both | 4/3/3 | -0.0802 | -0.0794 | [-0.789, 0.4756] | 0.9375 | 10 | 0.1429 |
| 11 | s_curve | QE_B | both | 10/0/0 | 0.0037 | 0.0038 | [8.5e-05, 0.0073] | 0.002 | 10 | 1 |
| 12 | swiss_roll | QE_B | both | 6/4/0 | 0.000814 | 0.0015 | [-0.0051, 0.0079] | 0.6953 | 10 | 0.2 |
| 2 | wine | QE_B | both | 4/2/4 | 2.01e-06 | 1.15e-06 | [-7.44e-06, 7.84e-06] | 0.6875 | 10 | 0.3333 |
| - | GLOBAL | QE_B | both | 69/41/30 | 1.58e-05 | -0.0208 | [-3.92e-06, 0.0011] | 0.0923 | 140 | 0.2545 |
| 8 | blobs | train time | train | 0/10/0 | -82.1972 | -82.2698 | [-82.616, -81.8815] | 0.002 | 10 | -1 |
| 5 | breast_cancer | train time | train | 0/10/0 | -124.4887 | -123.0875 | [-125.5044, -117.1834] | 0.002 | 10 | -1 |
| 7 | california_ housing | train time | train | 0/10/0 | -112.7705 | -107.7269 | [-120.1093, -94.1388] | 0.002 | 10 | -1 |
| 9 | circles | train time | train | 0/10/0 | -82.011 | -82.0468 | [-82.6543, -81.5143] | 0.002 | 10 | -1 |
| 14 | covertype | train time | train | 8/2/0 | 4.5544 | 3.1364 | [-0.2416, 4.7034] | 0.1055 | 10 | 0.6 |
| 4 | diabetes | train time | train | 0/10/0 | -124.7165 | -130.8947 | [-156.4431, -123.3642] | 0.002 | 10 | -1 |
| 6 | digits | train time | train | 0/10/0 | -124.2555 | -124.4075 | [-126.3659, -122.7873] | 0.002 | 10 | -1 |

| idx | dataset | metric | split | wins F/X/tie | median % | mean % | 95% CI | p | n | signed effect |
|---|---|---|---|---|---|---|---|---|---|---|
| 1 | iris | train time | train | 0/10/0 | -126.2679 | -127.1989 | [-132.6232, -124.6813] | 0.002 | 10 | -1 |
| 13 | kddcup99 | train time | train | 10/0/0 | 12.295 | 12.5548 | [8.0787, 15.7426] | 0.002 | 10 | 1 |
| 10 | moons | train time | train | 0/10/0 | -82.0957 | -88.2408 | [-112.9703, -81.6979] | 0.002 | 10 | -1 |
| 3 | olivetti_faces | train time | train | 0/10/0 | -200.8962 | -199.5774 | [-203.2204, -194.7869] | 0.002 | 10 | -1 |
| 11 | s_curve | train time | train | 0/10/0 | -83.1004 | -83.8438 | [-87.0546, -82.4761] | 0.002 | 10 | -1 |
| 12 | swiss_roll | train time | train | 1/9/0 | -81.642 | -65.9893 | [-82.6048, -3.0855] | 0.0039 | 10 | -0.8 |
| 2 | wine | train time | train | 0/10/0 | -126.5213 | -126.5147 | [-127.7591, -125.2633] | 0.002 | 10 | -1 |
| - | GLOBAL | train time | train | 19/121/0 | -102.7688 | -94.7219 | [-103.9162, -95.6444] | 1.02e-22 | 140 | -0.7286 |

Under review as submission to TMLR

Supplementary Table S7: Paired topology-comparison effects for hexagonal versus MST and hexagonal versus RNG across Balanced QE, Holdout QE, and Train QE.

Rows list metric/dataset entries, including the OVERALL row. Effect columns report the paired mean percent improvement of hexagonal over the comparator topology, using hexagonal QE as the reference denominator; positive values favor hexagonal and negative values favor MST or RNG. The CI columns give the corresponding 95% paired $t$-test confidence intervals. Raw p-values are retained for audit, and q-values report Benjamini-Hochberg adjustment across the 42 dataset-level tests in each topology-comparison family (14 datasets by three QE endpoints, separately for MST and RNG); OVERALL rows are pooled summaries and retain q=NA. The embedded table is reproduced from `assets/tables/supp_table_topology_hex_vs_mst_rng_pvalues.tsv`.

| metric | dataset | MST_effect_pct | MST_95pct_CI | MST_p | MST_q | RNG_effect_pct | RNG_95pct_CI | RNG_p | RNG_q |
|---|---|---|---|---|---|---|---|---|---|
| Balanced QE | blobs | -0.234 | [-0.4136, -0.0543] | p=0.0163 | q=0.0403 | -0.4278 | [-0.6395, -0.2161] | p=0.00134 | q=0.00353 |
| Balanced QE | circles | -0.0117 | [-0.1666, 0.1432] | p=0.868 | q=0.889 | -0.1507 | [-0.28, -0.02145] | p=0.027 | q=0.0494 |
| Balanced QE | moons | 0.04526 | [-0.09841, 0.1889] | p=0.494 | q=0.576 | -0.1834 | [-0.3183, -0.04856] | p=0.0132 | q=0.0277 |
| Balanced QE | s_curve | -0.2258 | [-0.324, -0.1277] | p=0.000559 | q=0.00261 | -0.2554 | [-0.3406, -0.1702] | p=8.08e-05 | q=0.000565 |
| Balanced QE | swiss_roll | -0.2335 | [-0.6601, 0.1931] | p=0.247 | q=0.358 | -0.3258 | [-0.6783, 0.02675] | p=0.0661 | q=0.0992 |
| Balanced QE | breast_cancer | 0.2996 | [-0.1996, 0.7987] | p=0.208 | q=0.336 | -0.1832 | [-0.6168, 0.2505] | p=0.364 | q=0.413 |
| Balanced QE | wine | -3.999 | [-5.789, -2.209] | p=0.000686 | q=0.00288 | -5.319 | [-7.359, -3.28] | p=0.000229 | q=0.0012 |
| Balanced QE | iris | -1.004 | [-4.964, 2.955] | p=0.58 | q=0.641 | -4.83 | [-7.089, -2.572] | p=0.000924 | q=0.00299 |
| Balanced QE | digits | -0.1745 | [-0.4361, 0.08697] | p=0.165 | q=0.298 | -0.1445 | [-0.4203, 0.1314] | p=0.266 | q=0.32 |
| Balanced QE | olivetti_faces | -1.265 | [-2.634, 0.1046] | p=0.0663 | q=0.146 | -1.23 | [-2.492, 0.03187] | p=0.0549 | q=0.0887 |
| Balanced QE | diabetes | 0.1492 | [-0.5331, 0.8315] | p=0.633 | q=0.681 | -0.3543 | [-1.031, 0.3227] | p=0.267 | q=0.32 |

| metric | dataset | MST_ effect_ pct | MST_95pct_ CI | MST_p | MST_q | RNG_ effect_ pct | RNG_95pct_ CI | RNG_p | RNG_q |
|---|---|---|---|---|---|---|---|---|---|
| Balanced QE | california_ housing | -0.28 | [-0.3942, -0.1659] | p=0.000356 | q=0.00187 | -0.184 | [-0.2426, -0.1254] | p=5.65e-05 | q=0.000565 |
| Balanced QE | covertype | -0.3486 | [-1.411, 0.7143] | p=0.477 | q=0.574 | -0.1093 | [-0.857, 0.6383] | p=0.748 | q=0.786 |
| Balanced QE | kddcup99 | -5.859 | [-6.623, -5.096] | p=3.14e-08 | q=6.6e-07 | -4.466 | [-5.768, -3.165] | p=2.81e-05 | q=0.000394 |
| Balanced QE | OVERALL | -0.9387 | [-1.346, -0.5316] | p=1.12e-05 | q=NA | -1.297 | [-1.685, -0.9097] | p=7.4e-10 | q=NA |
| Holdout QE | blobs | -0.1188 | [-0.4239, 0.1864] | p=0.401 | q=0.544 | -0.4212 | [-0.8076, -0.03478] | p=0.0358 | q=0.0627 |
| Holdout QE | circles | 0.2031 | [-0.05154, 0.4578] | p=0.105 | q=0.209 | -0.02455 | [-0.2726, 0.2235] | p=0.828 | q=0.828 |
| Holdout QE | moons | 0.1236 | [-0.08663, 0.3339] | p=0.216 | q=0.336 | -0.2158 | [-0.3211, -0.1106] | p=0.00122 | q=0.00342 |
| Holdout QE | s_curve | -0.1582 | [-0.2432, -0.07327] | p=0.00226 | q=0.00792 | -0.1842 | [-0.2673, -0.1011] | p=0.000724 | q=0.00299 |
| Holdout QE | swiss_roll | -0.303 | [-0.6657, 0.05966] | p=0.0913 | q=0.192 | -0.3654 | [-0.6648, -0.06609] | p=0.0221 | q=0.0421 |
| Holdout QE | breast_cancer | 0.4924 | [0.1735, 0.8113] | p=0.0068 | q=0.0204 | 0.2997 | [-0.0605, 0.6599] | p=0.0925 | q=0.129 |
| Holdout QE | wine | 1.582 | [0.9599, 2.203] | p=0.000275 | q=0.00165 | 1.148 | [0.524, 1.772] | p=0.00244 | q=0.0057 |
| Holdout QE | iris | 2.725 | [0.7442, 4.705] | p=0.0125 | q=0.0349 | 2.032 | [0.06363, 4.001] | p=0.0444 | q=0.0745 |
| Holdout QE | digits | -0.2882 | [-0.5927, 0.01621] | p=0.0609 | q=0.142 | -0.1447 | [-0.4001, 0.1107] | p=0.232 | q=0.305 |
| Holdout QE | olivetti_faces | -1.322 | [-1.723, -0.9199] | p=3.93e-05 | q=0.000412 | -1.713 | [-2.152, -1.275] | p=9.96e-06 | q=0.000333 |
| Holdout QE | diabetes | 0.003896 | [-0.4209, 0.4287] | p=0.984 | q=0.984 | -0.273 | [-0.901, 0.355] | p=0.351 | q=0.41 |
| Holdout QE | california_ housing | -0.2211 | [-0.3483, -0.09394] | p=0.00344 | q=0.0111 | -0.1404 | [-0.247, -0.03388] | p=0.0154 | q=0.0308 |

Supplementary Table S7 (continued)

| metric | dataset | MST_effect_pct | MST_95pct_CI | MST_p | MST_q | RNG_effect_pct | RNG_95pct_CI | RNG_p | RNG_q |
|---|---|---|---|---|---|---|---|---|---|
| Holdout QE | covertype | -0.3532 | [-1.419, 0.7122] | p=0.472 | q=0.574 | -0.08568 | [-0.8426, 0.6712] | p=0.804 | q=0.823 |
| Holdout QE | kddcup99 | -5.871 | [-6.673, -5.068] | p=4.8e-08 | q=6.72e-07 | -4.643 | [-5.902, -3.383] | p=1.58e-05 | q=0.000333 |
| Holdout QE | OVERALL | -0.2504 | [-0.5924, 0.09159] | p=0.15 | q=NA | -0.3379 | [-0.6291, -0.0468] | p=0.0232 | q=NA |
| Train QE | blobs | -0.2851 | [-0.5038, -0.06639] | p=0.0162 | q=0.0403 | -0.4429 | [-0.6468, -0.2391] | p=0.00083 | q=0.00299 |
| Train QE | circles | -0.1236 | [-0.2862, 0.03895] | p=0.119 | q=0.228 | -0.1656 | [-0.2318, -0.09945] | p=0.000309 | q=0.00144 |
| Train QE | moons | -0.04045 | [-0.1551, 0.07423] | p=0.445 | q=0.574 | -0.157 | [-0.2696, -0.04434] | p=0.0117 | q=0.0258 |
| Train QE | s_curve | -0.2144 | [-0.2961, -0.1327] | p=0.000219 | q=0.00153 | -0.2643 | [-0.3584, -0.1702] | p=0.000132 | q=0.000794 |
| Train QE | swiss_roll | -0.2022 | [-0.6787, 0.2742] | p=0.362 | q=0.507 | -0.3216 | [-0.6867, 0.04341] | p=0.0774 | q=0.112 |
| Train QE | breast_cancer | 0.9588 | [-0.4954, 2.413] | p=0.17 | q=0.298 | -0.263 | [-1.543, 1.016] | p=0.653 | q=0.722 |
| Train QE | wine | -17.32 | [-23.63, -11.02] | p=0.000156 | q=0.00131 | -22.4 | [-33.21, -11.6] | p=0.00114 | q=0.00341 |
| Train QE | iris | -5.888 | [-28.17, 16.4] | p=0.565 | q=0.641 | -25.13 | [-36.77, -13.48] | p=0.000872 | q=0.00299 |
| Train QE | digits | -0.2587 | [-0.7176, 0.2002] | p=0.234 | q=0.351 | -0.3143 | [-0.6485, 0.01992] | p=0.0623 | q=0.0969 |
| Train QE | olivetti_faces | -1.791 | [-4.796, 1.215] | p=0.211 | q=0.336 | -1.553 | [-4.04, 0.9338] | p=0.191 | q=0.259 |
| Train QE | diabetes | 0.2611 | [-1.533, 2.055] | p=0.75 | q=0.787 | -0.8799 | [-2.463, 0.7033] | p=0.24 | q=0.306 |
| Train QE | california_housing | -0.3185 | [-0.4766, -0.1604] | p=0.00137 | q=0.00523 | -0.2556 | [-0.3834, -0.1278] | p=0.00144 | q=0.00355 |
| Train QE | covertype | -0.3468 | [-1.407, 0.7137] | p=0.478 | q=0.574 | -0.1341 | [-0.8722, 0.6039] | p=0.691 | q=0.744 |
| Train QE | kddcup99 | -5.804 | [-6.474, -5.134] | p=1.09e-08 | q=4.56e-07 | -4.183 | [-5.557, -2.809] | p=7.18e-05 | q=0.000565 |
| Train QE | OVERALL | -2.241 | [-3.842, -0.6405] | p=0.0064 | q=NA | -4.033 | [-5.707, -2.36] | p=4.69e-06 | q=NA |

Supplementary Table S8: Figure 15 deployment comparison percent summary for tuned FloatSOM RNG versus untuned hexagonal XPySOM across $QE_B$, $QE_H$, and $QE_T$.

Rows list per-dataset and `GLOBAL_OVERALL` entries with the plotted median percent change and 95% confidence interval.

| metric | dataset | median % change | 95% CI |
|--------|---------|-----------------|--------|
| QE_B | blobs | 5.4588 | [3.8282, 7.0895] |
| QE_B | circles | 5.6542 | [4.552, 6.7563] |
| QE_B | moons | 4.4217 | [2.6294, 6.2141] |
| QE_B | s_curve | 10.0599 | [9.2501, 10.8696] |
| QE_B | swiss_roll | 19.7271 | [18.4881, 20.9661] |
| QE_B | breast_cancer | 7.2273 | [6.3913, 8.0633] |
| QE_B | wine | 20.5163 | [19.3744, 21.6583] |
| QE_B | iris | 23.2197 | [20.569, 25.8704] |
| QE_B | digits | 8.3021 | [7.8283, 8.7758] |
| QE_B | olivetti_faces | 10.9797 | [10.0517, 11.9077] |
| QE_B | diabetes | 9.2085 | [8.1874, 10.2296] |
| QE_B | california_housing | 9.4344 | [9.0497, 9.8191] |
| QE_B | covertype | 30.3468 | [29.1115, 31.5821] |
| QE_B | kddcup99 | 38.2338 | [35.8971, 40.5705] |
| QE_B | GLOBAL_OVERALL | 14.485 | [12.7836, 16.1865] |
| QE_H | blobs | 4.8943 | [3.3372, 6.4515] |
| QE_H | circles | 4.98 | [3.8583, 6.1016] |
| QE_H | moons | 3.8767 | [1.9564, 5.797] |
| QE_H | s_curve | 9.4896 | [8.5416, 10.4377] |
| QE_H | swiss_roll | 19.2924 | [17.9583, 20.6264] |
| QE_H | breast_cancer | 0.1143 | [-0.6627, 0.8913] |
| QE_H | wine | -1.8427 | [-3.2635, -0.4218] |
| QE_H | iris | 0.4853 | [-2.8955, 3.8661] |
| QE_H | digits | 4.9934 | [4.5072, 5.4796] |
| QE_H | olivetti_faces | 2.6384 | [1.2928, 3.9839] |
| QE_H | diabetes | 1.3763 | [-0.3528, 3.1055] |
| QE_H | california_housing | 8.5235 | [8.0055, 9.0414] |
| QE_H | covertype | 30.3196 | [29.0984, 31.5408] |
| QE_H | kddcup99 | 37.9291 | [35.5645, 40.2937] |
| QE_H | GLOBAL_OVERALL | 9.0765 | [7.1203, 11.0326] |
| QE_T | blobs | 6.0226 | [4.2924, 7.7527] |
| QE_T | circles | 6.3297 | [5.1892, 7.4702] |
| QE_T | moons | 4.968 | [3.2744, 6.6616] |
| QE_T | s_curve | 10.6309 | [9.9339, 11.3279] |
| QE_T | swiss_roll | 20.1631 | [18.9915, 21.3347] |
| QE_T | breast_cancer | 15.8475 | [14.5024, 17.1926] |
| QE_T | wine | 56.1875 | [54.54, 57.8351] |
| QE_T | iris | 63.2439 | [60.2553, 66.2326] |
| QE_T | digits | 11.8094 | [11.1554, 12.4634] |
| QE_T | olivetti_faces | 21.029 | [19.7281, 22.3298] |
| QE_T | diabetes | 18.9602 | [18.1282, 19.7921] |

Supplementary Table S8 (continued)

| metric | dataset | median % change | 95% CI |
|---|---|---|---|
| QE_T | california_housing | 10.3513 | [9.7384, 10.9642] |
| QE_T | covertype | 30.3738 | [29.1225, 31.6251] |
| QE_T | kddcup99 | 38.5352 | [36.2102, 40.8602] |
| QE_T | GLOBAL_OVERALL | 22.4609 | [19.4613, 25.4605] |

Supplementary Table S9: Supplementary Figure S7 deployment comparison percent summary for tuned FloatSOM hexagonal versus untuned hexagonal XPySOM across $QE_B$, $QE_H$, and $QE_T$.

Rows list per-dataset and `GLOBAL_OVERALL` entries with the plotted median percent change and 95% confidence interval.

| metric | dataset | median % change | 95% CI |
|---|---|---|---|
| QE_B | blobs | 4.5418 | [3.0375, 6.0462] |
| QE_B | circles | 4.5998 | [3.6022, 5.5974] |
| QE_B | moons | 3.5895 | [2.051, 5.128] |
| QE_B | s_curve | 9.7727 | [9.0814, 10.464] |
| QE_B | swiss_roll | 17.6702 | [16.7526, 18.5878] |
| QE_B | breast_cancer | 6.399 | [5.5895, 7.2084] |
| QE_B | wine | 15.0518 | [13.113, 16.9905] |
| QE_B | iris | 19.6158 | [17.7567, 21.475] |
| QE_B | digits | 7.7568 | [7.3648, 8.1489] |
| QE_B | olivetti_faces | 10.367 | [9.5746, 11.1594] |
| QE_B | diabetes | 9.133 | [8.2954, 9.9707] |
| QE_B | california_housing | 8.9193 | [8.4319, 9.4067] |
| QE_B | covertype | 29.6621 | [28.3242, 31] |
| QE_B | kddcup99 | 35.5053 | [34.0328, 36.9779] |
| QE_B | GLOBAL_OVERALL | 13.0417 | [11.4631, 14.6204] |
| QE_H | blobs | 3.8869 | [2.4806, 5.2931] |
| QE_H | circles | 4.1388 | [3.1274, 5.1503] |
| QE_H | moons | 2.9919 | [1.3076, 4.6761] |
| QE_H | s_curve | 9.2419 | [8.4744, 10.0095] |
| QE_H | swiss_roll | 17.3207 | [16.3909, 18.2504] |
| QE_H | breast_cancer | -0.997 | [-1.683, -0.311] |
| QE_H | wine | -2.8868 | [-4.4124, -1.3613] |
| QE_H | iris | -0.4904 | [-2.1865, 1.2058] |
| QE_H | digits | 4.5103 | [4.0172, 5.0034] |
| QE_H | olivetti_faces | 1.9861 | [0.769, 3.2032] |
| QE_H | diabetes | 2.245 | [1.1198, 3.3702] |
| QE_H | california_housing | 8.0301 | [7.4144, 8.6458] |
| QE_H | covertype | 29.6665 | [28.3026, 31.0305] |
| QE_H | kddcup99 | 35.1253 | [33.6747, 36.5759] |
| QE_H | GLOBAL_OVERALL | 8.1978 | [6.3289, 10.0667] |
| QE_T | blobs | 5.1961 | [3.5689, 6.8233] |

Supplementary Table S9 (continued)

| metric | dataset | median % change | 95% CI |
|--------|---------|-----------------|--------|
| QE_T | circles | 5.0606 | [4.0085, 6.1127] |
| QE_T | moons | 4.1898 | [2.7833, 5.5963] |
| QE_T | s_curve | 10.3044 | [9.6672, 10.9416] |
| QE_T | swiss_roll | 18.021 | [17.101, 18.9411] |
| QE_T | breast_cancer | 15.3534 | [14.1184, 16.5884] |
| QE_T | wine | 43.6818 | [39.113, 48.2507] |
| QE_T | iris | 55.1557 | [50.9804, 59.331] |
| QE_T | digits | 11.2012 | [10.7183, 11.6841] |
| QE_T | olivetti_faces | 20.4609 | [19.2364, 21.6853] |
| QE_T | diabetes | 17.7191 | [16.8369, 18.6013] |
| QE_T | california_housing | 9.8167 | [9.2321, 10.4012] |
| QE_T | covertype | 29.6576 | [28.3443, 30.9708] |
| QE_T | kddcup99 | 35.8819 | [34.3431, 37.4206] |
| QE_T | GLOBAL_OVERALL | 20.1214 | [17.5722, 22.6707] |

Supplementary Table S10: Supplementary Figure S8 deployment comparison percent summary for tuned FloatSOM MST versus untuned hexagonal XPySOM across $QE_B$, $QE_H$, and $QE_T$.

Rows list per-dataset and GLOBAL_OVERALL entries with the plotted median percent change and 95% confidence interval.

| metric | dataset | median % change | 95% CI |
|--------|---------|-----------------|--------|
| QE_B | blobs | 5.2243 | [3.6173, 6.8314] |
| QE_B | circles | 5.5236 | [4.4413, 6.6059] |
| QE_B | moons | 4.2266 | [2.4056, 6.0477] |
| QE_B | s_curve | 9.968 | [9.2244, 10.7116] |
| QE_B | swiss_roll | 19.5015 | [18.1673, 20.8356] |
| QE_B | breast_cancer | 7.2275 | [6.4917, 7.9633] |
| QE_B | wine | 18.0952 | [16.4439, 19.7464] |
| QE_B | iris | 17.3842 | [13.8051, 20.9634] |
| QE_B | digits | 8.5168 | [7.9287, 9.105] |
| QE_B | olivetti_faces | 11.9472 | [10.795, 13.0994] |
| QE_B | diabetes | 9.2961 | [8.2097, 10.3825] |
| QE_B | california_housing | 9.474 | [9.0518, 9.8963] |
| QE_B | covertype | 32.2987 | [30.9976, 33.5997] |
| QE_B | kddcup99 | 40.8705 | [39.5821, 42.159] |
| QE_B | GLOBAL_OVERALL | 14.2539 | [12.4871, 16.0207] |
| QE_H | blobs | 4.5705 | [3.0306, 6.1104] |
| QE_H | circles | 4.8064 | [3.6639, 5.9489] |
| QE_H | moons | 3.768 | [1.7919, 5.744] |
| QE_H | s_curve | 9.4507 | [8.6251, 10.2763] |
| QE_H | swiss_roll | 19.0489 | [17.593, 20.5048] |
| QE_H | breast_cancer | -0.7772 | [-1.3919, -0.1624] |

Supplementary Table S10 (continued)

| metric | dataset | median % change | 95% CI |
|---|---|---|---|
| QE_H | wine | -2.2046 | [-3.9272, -0.4821] |
| QE_H | iris | -0.872 | [-3.7238, 1.9798] |
| QE_H | digits | 5.2512 | [4.6436, 5.8587] |
| QE_H | olivetti_faces | 2.5542 | [0.9211, 4.1873] |
| QE_H | diabetes | 1.059 | [-0.5135, 2.6316] |
| QE_H | california_housing | 8.4673 | [7.9248, 9.0099] |
| QE_H | covertype | 32.2708 | [30.9787, 33.5629] |
| QE_H | kddcup99 | 40.3684 | [39.0453, 41.6915] |
| QE_H | GLOBAL_OVERALL | 9.1258 | [7.0316, 11.2201] |
| QE_T | blobs | 5.8778 | [4.178, 7.5776] |
| QE_T | circles | 6.2415 | [5.1655, 7.3175] |
| QE_T | moons | 4.6869 | [3.0023, 6.3715] |
| QE_T | s_curve | 10.486 | [9.7854, 11.1866] |
| QE_T | swiss_roll | 19.9554 | [18.7303, 21.1804] |
| QE_T | breast_cancer | 16.9227 | [15.7815, 18.0639] |
| QE_T | wine | 50.5129 | [47.2136, 53.8122] |
| QE_T | iris | 49.5872 | [43.4267, 55.7476] |
| QE_T | digits | 11.9794 | [11.277, 12.6817] |
| QE_T | olivetti_faces | 23.2531 | [21.9339, 24.5724] |
| QE_T | diabetes | 19.5575 | [18.2819, 20.8331] |
| QE_T | california_housing | 10.4893 | [9.9385, 11.04] |
| QE_T | covertype | 32.3264 | [31.0143, 33.6385] |
| QE_T | kddcup99 | 41.3715 | [40.0794, 42.6636] |
| QE_T | GLOBAL_OVERALL | 21.6605 | [19.0539, 24.2672] |

Supplementary Table S11: Figure 14 topology runtime summary at the largest common 8-GPU axis value for the dimension, sample, and grid size scaling workloads.

Rows report the plotted 8-GPU mean runtimes for hexagonal, MST, and RNG, together with the fastest and slowest topology at that axis value and the maximum pairwise runtime spread.

| axis | axis value | hexagonal s | MST s | RNG s | fastest | slowest | spread % |
|---|---|---|---|---|---|---|---|
| Dimension Scaling | 5000 | 404.17 | 423.15 | 409.94 | hexagonal | mst | 4.7 |
| Sample Scaling | 1e+09 | 363.58 | 375.44 | 369.41 | hexagonal | mst | 3.26 |
| Grid-Size Scaling | 64 | 32.54 | 266.45 | 880.83 | hexagonal | rng | 2606.61 |

Supplementary Table S12: Matched topology diagnostic paired summaries across tuned and untuned profiles.

Rows report pooled paired effects from the final matched random-seed topology diagnostic benchmark. `Effect favoring comparator` is oriented so positive values favor the comparator after applying each metric's directionality; lower is better for QE, MTR, and dead-node fraction, while higher is better for node utilization. Because these are pre-specified pooled paired summaries rather than dataset-level test families, the table reports raw paired-test p-values. `comp/ref/tie` gives the number of comparator-favoring, reference-favoring, and tied matched pairs. Metric suffixes denote holdout (`_H`), train (`_T`), and balanced train-holdout (`_B`) summaries. The tuned profile uses the deployable tuned configurations reported in the main text.

| Comparison | Metric | Better | n | Effect favoring comparator | 95% CI | dz | raw p | comp/ref/tie |
|---|---|---|---|---|---|---|---|---|
| Untuned hex vs untuned MST | QE_H | lower | 280 | 0.1523 | [0.1110, 0.1937] | 0.4330 | 4.22e-12 | 185/95/0 |
| Untuned hex vs untuned MST | QE_T | lower | 280 | 0.3862 | [0.2780, 0.4944] | 0.4199 | 1.63e-11 | 226/54/0 |
| Untuned hex vs untuned MST | QE_B | lower | 280 | 0.2693 | [0.1975, 0.3410] | 0.4415 | 1.73e-12 | 222/58/0 |
| Untuned hex vs untuned MST | MTR_H | lower | 280 | -0.6164 | [-0.9036, -0.3292] | -0.2525 | 3.24e-05 | 125/155/0 |
| Untuned hex vs untuned MST | MTR_T | lower | 280 | 0.2075 | [-0.0155, 0.4304] | 0.1095 | 0.068 | 167/113/0 |
| Untuned hex vs untuned MST | MTR_B | lower | 280 | -0.2045 | [-0.4382, 0.0293] | -0.1029 | 0.086 | 139/141/0 |
| Untuned hex vs untuned MST | Util_H | higher | 280 | 0.0187 | [0.0115, 0.0260] | 0.3056 | 5.88e-07 | 110/73/97 |
| Untuned hex vs untuned MST | Util_T | higher | 280 | 0.0414 | [0.0345, 0.0483] | 0.7049 | 2.55e-26 | 147/17/116 |
| Untuned hex vs untuned MST | Util_B | higher | 280 | 0.0301 | [0.0237, 0.0364] | 0.5534 | 5.62e-18 | 144/46/90 |
| Untuned hex vs untuned MST | Dead_H | lower | 280 | 0.0188 | [0.0115, 0.0260] | 0.3056 | 5.88e-07 | 110/73/97 |
| Untuned hex vs untuned MST | Dead_T | lower | 280 | 0.0414 | [0.0345, 0.0483] | 0.7049 | 2.55e-26 | 147/17/116 |

Continued on next page

Supplementary Table S12 (continued)

| Comparison | Metric | Better | n | Effect favoring comparator | 95% CI | dz | raw p | comp/ref/tie |
|---|---|---|---|---|---|---|---|---|
| Untuned hex vs untuned MST | Dead_B | lower | 280 | 0.0301 | [0.0237, 0.0364] | 0.5534 | 5.62e-18 | 144/46/90 |
| Untuned hex vs untuned RNG | QE_H | lower | 280 | 0.1523 | [0.1094, 0.1953] | 0.4174 | 2.10e-11 | 206/74/0 |
| Untuned hex vs untuned RNG | QE_T | lower | 280 | 0.3464 | [0.2480, 0.4447] | 0.4143 | 2.88e-11 | 246/34/0 |
| Untuned hex vs untuned RNG | QE_B | lower | 280 | 0.2494 | [0.1808, 0.3179] | 0.4281 | 6.99e-12 | 241/39/0 |
| Untuned hex vs untuned RNG | MTR_H | lower | 280 | 2.3952 | [2.1221, 2.6683] | 1.0316 | 6.30e-46 | 256/24/0 |
| Untuned hex vs untuned RNG | MTR_T | lower | 280 | 2.4838 | [2.2596, 2.7080] | 1.3032 | 3.18e-62 | 272/8/0 |
| Untuned hex vs untuned RNG | MTR_B | lower | 280 | 2.4395 | [2.2033, 2.6757] | 1.2152 | 5.16e-57 | 269/11/0 |
| Untuned hex vs untuned RNG | Util_H | higher | 280 | 0.0225 | [0.0157, 0.0294] | 0.3886 | 3.65e-10 | 125/55/100 |
| Untuned hex vs untuned RNG | Util_T | higher | 280 | 0.0432 | [0.0361, 0.0504] | 0.7162 | 5.62e-27 | 157/10/113 |
| Untuned hex vs untuned RNG | Util_B | higher | 280 | 0.0329 | [0.0266, 0.0392] | 0.6130 | 3.69e-21 | 157/30/93 |
| Untuned hex vs untuned RNG | Dead_H | lower | 280 | 0.0225 | [0.0157, 0.0294] | 0.3886 | 3.65e-10 | 125/55/100 |
| Untuned hex vs untuned RNG | Dead_T | lower | 280 | 0.0432 | [0.0361, 0.0504] | 0.7162 | 5.62e-27 | 157/10/113 |
| Untuned hex vs untuned RNG | Dead_B | lower | 280 | 0.0329 | [0.0266, 0.0392] | 0.6130 | 3.69e-21 | 157/30/93 |
| Untuned MST vs untuned RNG | QE_H | lower | 280 | -5.54e-06 | [-0.0198, 0.0198] | -3.29e-05 | 1.000 | 138/142/0 |
| Untuned MST vs untuned RNG | QE_T | lower | 280 | -0.0398 | [-0.0616, -0.0181] | -0.2159 | 3.59e-04 | 107/173/0 |
| Untuned MST vs untuned RNG | QE_B | lower | 280 | -0.0199 | [-0.0385, -0.0013] | -0.1261 | 0.036 | 120/160/0 |

Supplementary Table S12 (continued)

| Comparison | Metric | Better | n | Effect favoring comparator | 95% CI | dz | raw p | comp/ref/tie |
|---|---|---|---|---|---|---|---|---|
| Untuned MST vs untuned RNG | MTR_H | lower | 280 | 3.0116 | [2.6987, 3.3245] | 1.1324 | 4.89e-52 | 238/42/0 |
| Untuned MST vs untuned RNG | MTR_T | lower | 280 | 2.2763 | [2.0287, 2.5240] | 1.0815 | 5.90e-49 | 233/47/0 |
| Untuned MST vs untuned RNG | MTR_B | lower | 280 | 2.6440 | [2.3810, 2.9070] | 1.1826 | 4.64e-55 | 240/40/0 |
| Untuned MST vs untuned RNG | Util_H | higher | 280 | 0.0038 | [7.90e-05, 0.0075] | 0.1201 | 0.045 | 93/64/123 |
| Untuned MST vs untuned RNG | Util_T | higher | 280 | 0.0019 | [-0.0012, 0.0050] | 0.0712 | 0.235 | 82/58/140 |
| Untuned MST vs untuned RNG | Util_B | higher | 280 | 0.0028 | [-9.28e-05, 0.0058] | 0.1139 | 0.058 | 101/69/110 |
| Untuned MST vs untuned RNG | Dead_H | lower | 280 | 0.0038 | [7.90e-05, 0.0075] | 0.1201 | 0.045 | 93/64/123 |
| Untuned MST vs untuned RNG | Dead_T | lower | 280 | 0.0019 | [-0.0012, 0.0050] | 0.0712 | 0.235 | 82/58/140 |
| Untuned MST vs untuned RNG | Dead_B | lower | 280 | 0.0028 | [-9.28e-05, 0.0058] | 0.1139 | 0.058 | 101/69/110 |
| Tuned hex vs tuned MST | QE_H | lower | 280 | 0.0404 | [0.0199, 0.0608] | 0.2323 | 1.27e-04 | 199/81/0 |
| Tuned hex vs tuned MST | QE_T | lower | 280 | 0.0872 | [0.0537, 0.1207] | 0.3063 | 5.54e-07 | 246/34/0 |
| Tuned hex vs tuned MST | QE_B | lower | 280 | 0.0638 | [0.0379, 0.0897] | 0.2897 | 2.07e-06 | 226/54/0 |
| Tuned hex vs tuned MST | MTR_H | lower | 280 | 22.2866 | [21.4521, 23.1212] | 3.1417 | 8.86e-147 | 280/0/0 |
| Tuned hex vs tuned MST | MTR_T | lower | 280 | 23.1240 | [22.2558, 23.9922] | 3.1331 | 1.77e-146 | 280/0/0 |
| Tuned hex vs tuned MST | MTR_B | lower | 280 | 22.7053 | [21.8739, 23.5368] | 3.2126 | 3.06e-149 | 280/0/0 |
| Tuned hex vs tuned MST | Util_H | higher | 280 | 0.0092 | [0.0055, 0.0129] | 0.2905 | 1.95e-06 | 116/63/101 |

Continued on next page

| Comparison | Metric | Better | n | Effect favoring comparator | 95% CI | dz | raw p | comp/ref/tie |
|---|---|---|---|---|---|---|---|---|
| Tuned hex vs tuned MST | Util_T | higher | 280 | 0.0222 | [0.0185, 0.0260] | 0.6975 | 6.79e-26 | 150/13/117 |
| Tuned hex vs tuned MST | Util_B | higher | 280 | 0.0157 | [0.0125, 0.0189] | 0.5798 | 2.30e-19 | 139/47/94 |
| Tuned hex vs tuned MST | Dead_H | lower | 280 | 0.0092 | [0.0055, 0.0129] | 0.2905 | 1.95e-06 | 116/63/101 |
| Tuned hex vs tuned MST | Dead_T | lower | 280 | 0.0222 | [0.0185, 0.0260] | 0.6975 | 6.79e-26 | 150/13/117 |
| Tuned hex vs tuned MST | Dead_B | lower | 280 | 0.0157 | [0.0125, 0.0189] | 0.5798 | 2.30e-19 | 140/47/93 |
| Tuned hex vs tuned RNG | QE_H | lower | 280 | 0.0360 | [0.0186, 0.0533] | 0.2434 | 6.04e-05 | 210/70/0 |
| Tuned hex vs tuned RNG | QE_T | lower | 280 | 0.0671 | [0.0409, 0.0932] | 0.3014 | 8.25e-07 | 245/35/0 |
| Tuned hex vs tuned RNG | QE_B | lower | 280 | 0.0515 | [0.0318, 0.0712] | 0.3077 | 4.94e-07 | 240/40/0 |
| Tuned hex vs tuned RNG | MTR_H | lower | 280 | 25.4035 | [24.5013, 26.3057] | 3.3124 | 1.24e-152 | 280/0/0 |
| Tuned hex vs tuned RNG | MTR_T | lower | 280 | 25.9296 | [25.0046, 26.8545] | 3.2979 | 3.81e-152 | 280/0/0 |
| Tuned hex vs tuned RNG | MTR_B | lower | 280 | 25.6665 | [24.7693, 26.5638] | 3.3652 | 2.17e-154 | 280/0/0 |
| Tuned hex vs tuned RNG | Util_H | higher | 280 | 0.0117 | [0.0076, 0.0158] | 0.3348 | 5.07e-08 | 110/59/111 |
| Tuned hex vs tuned RNG | Util_T | higher | 280 | 0.0276 | [0.0226, 0.0326] | 0.6496 | 3.52e-23 | 143/12/125 |
| Tuned hex vs tuned RNG | Util_B | higher | 280 | 0.0197 | [0.0158, 0.0235] | 0.6025 | 1.38e-20 | 138/40/102 |
| Tuned hex vs tuned RNG | Dead_H | lower | 280 | 0.0117 | [0.0076, 0.0158] | 0.3348 | 5.07e-08 | 110/59/111 |
| Tuned hex vs tuned RNG | Dead_T | lower | 280 | 0.0276 | [0.0226, 0.0326] | 0.6496 | 3.52e-23 | 143/12/125 |

Supplementary Table S12 (continued)

| Comparison | Metric | Better | n | Effect favoring comparator | 95% CI | dz | raw p | comp/ref/tie |
|---|---|---|---|---|---|---|---|---|
| Tuned hex vs tuned RNG | Dead_B | lower | 280 | 0.0197 | [0.0158, 0.0235] | 0.6025 | 1.38e-20 | 138/40/102 |
| Tuned MST vs tuned RNG | QE_H | lower | 280 | -0.0044 | [-0.0218, 0.0130] | -0.0298 | 0.619 | 161/119/0 |
| Tuned MST vs tuned RNG | QE_T | lower | 280 | -0.0201 | [-0.0395, -8.07e-04] | -0.1225 | 0.041 | 161/119/0 |
| Tuned MST vs tuned RNG | QE_B | lower | 280 | -0.0123 | [-0.0280, 0.0035] | -0.0916 | 0.126 | 167/113/0 |
| Tuned MST vs tuned RNG | MTR_H | lower | 280 | 3.1169 | [2.8270, 3.4067] | 1.2650 | 5.69e-60 | 254/26/0 |
| Tuned MST vs tuned RNG | MTR_T | lower | 280 | 2.8056 | [2.5596, 3.0516] | 1.3416 | 1.81e-64 | 257/23/0 |
| Tuned MST vs tuned RNG | MTR_B | lower | 280 | 2.9612 | [2.7049, 3.2175] | 1.3591 | 1.75e-65 | 260/20/0 |
| Tuned MST vs tuned RNG | Util_H | higher | 280 | 0.0025 | [-3.62e-04, 0.0054] | 0.1029 | 0.086 | 75/79/126 |
| Tuned MST vs tuned RNG | Util_T | higher | 280 | 0.0054 | [0.0027, 0.0081] | 0.2361 | 9.90e-05 | 69/51/160 |
| Tuned MST vs tuned RNG | Util_B | higher | 280 | 0.0040 | [0.0017, 0.0062] | 0.2073 | 6.07e-04 | 87/80/113 |
| Tuned MST vs tuned RNG | Dead_H | lower | 280 | 0.0025 | [-3.62e-04, 0.0054] | 0.1029 | 0.086 | 75/79/126 |
| Tuned MST vs tuned RNG | Dead_T | lower | 280 | 0.0054 | [0.0027, 0.0081] | 0.2361 | 9.90e-05 | 69/51/160 |
| Tuned MST vs tuned RNG | Dead_B | lower | 280 | 0.0040 | [0.0017, 0.0062] | 0.2073 | 6.07e-04 | 87/80/113 |
| Tuned vs untuned hex | QE_H | lower | 280 | 0.1651 | [0.1227, 0.2075] | 0.4580 | 2.99e-13 | 230/50/0 |
| Tuned vs untuned hex | QE_T | lower | 280 | 0.6969 | [0.4891, 0.9047] | 0.3945 | 2.04e-10 | 280/0/0 |
| Tuned vs untuned hex | QE_B | lower | 280 | 0.4310 | [0.3117, 0.5503] | 0.4249 | 9.78e-12 | 280/0/0 |
| Tuned vs untuned hex | MTR_H | lower | 280 | -24.2851 | [-25.2036, -23.3666] | -3.1103 | 1.12e-145 | 0/280/0 |
| Tuned vs untuned hex | MTR_T | lower | 280 | -24.3956 | [-25.3383, -23.4529] | -3.0444 | 2.51e-143 | 0/280/0 |

Continued on next page

Supplementary Table S12 (continued)

| Comparison | Metric | Better | n | Effect favoring comparator | 95% CI | dz | raw p | comp/ref/tie |
|---|---|---|---|---|---|---|---|---|
| Tuned vs untuned hex | MTR_B | lower | 280 | -24.3403 | [-25.2543, -23.4263] | -3.1328 | 1.82e-146 | 0/280/0 |
| Tuned vs untuned hex | Util_H | higher | 280 | 0.0087 | [0.0023, 0.0150] | 0.1610 | 0.007 | 98/93/89 |
| Tuned vs untuned hex | Util_T | higher | 280 | 0.0380 | [0.0321, 0.0439] | 0.7533 | 3.85e-29 | 150/17/113 |
| Tuned vs untuned hex | Util_B | higher | 280 | 0.0233 | [0.0180, 0.0287] | 0.5149 | 5.24e-16 | 139/53/88 |
| Tuned vs untuned hex | Dead_H | lower | 280 | 0.0087 | [0.0023, 0.0150] | 0.1610 | 0.007 | 98/93/89 |
| Tuned vs untuned hex | Dead_T | lower | 280 | 0.0380 | [0.0321, 0.0439] | 0.7533 | 3.85e-29 | 150/17/113 |
| Tuned vs untuned hex | Dead_B | lower | 280 | 0.0233 | [0.0180, 0.0287] | 0.5149 | 5.24e-16 | 139/53/88 |
| Tuned vs untuned MST | QE_H | lower | 280 | 0.0531 | [0.0319, 0.0744] | 0.2939 | 1.49e-06 | 224/56/0 |
| Tuned vs untuned MST | QE_T | lower | 280 | 0.3979 | [0.2681, 0.5276] | 0.3607 | 5.03e-09 | 280/0/0 |
| Tuned vs untuned MST | QE_B | lower | 280 | 0.2255 | [0.1546, 0.2963] | 0.3744 | 1.41e-09 | 276/4/0 |
| Tuned vs untuned MST | MTR_H | lower | 280 | -1.3820 | [-1.6350, -1.1290] | -0.6427 | 8.56e-23 | 55/225/0 |
| Tuned vs untuned MST | MTR_T | lower | 280 | -1.4791 | [-1.6529, -1.3052] | -1.0006 | 4.89e-44 | 27/253/0 |
| Tuned vs untuned MST | MTR_B | lower | 280 | -1.4305 | [-1.6184, -1.2427] | -0.8959 | 1.14e-37 | 39/241/0 |
| Tuned vs untuned MST | Util_H | higher | 280 | -8.93e-04 | [-0.0052, 0.0034] | -0.0247 | 0.680 | 75/100/105 |
| Tuned vs untuned MST | Util_T | higher | 280 | 0.0189 | [0.0145, 0.0232] | 0.5097 | 9.49e-16 | 121/37/122 |
| Tuned vs untuned MST | Util_B | higher | 280 | 0.0090 | [0.0054, 0.0126] | 0.2935 | 1.55e-06 | 103/79/98 |
| Tuned vs untuned MST | Dead_H | lower | 280 | -8.93e-04 | [-0.0052, 0.0034] | -0.0247 | 0.680 | 75/100/105 |
| Tuned vs untuned MST | Dead_T | lower | 280 | 0.0189 | [0.0145, 0.0232] | 0.5097 | 9.49e-16 | 121/37/122 |
| Tuned vs untuned MST | Dead_B | lower | 280 | 0.0090 | [0.0054, 0.0126] | 0.2935 | 1.55e-06 | 103/79/98 |
| Tuned vs untuned RNG | QE_H | lower | 280 | 0.0487 | [0.0261, 0.0713] | 0.2532 | 3.07e-05 | 223/57/0 |
| Tuned vs untuned RNG | QE_T | lower | 280 | 0.4176 | [0.2918, 0.5434] | 0.3905 | 3.02e-10 | 280/0/0 |
| Tuned vs untuned RNG | QE_B | lower | 280 | 0.2331 | [0.1681, 0.2982] | 0.4215 | 1.39e-11 | 280/0/0 |
| Tuned vs untuned RNG | MTR_H | lower | 280 | -1.2768 | [-1.4669, -1.0867] | -0.7902 | 2.56e-31 | 35/245/0 |
| Tuned vs untuned RNG | MTR_T | lower | 280 | -0.9498 | [-1.0619, -0.8378] | -0.9973 | 7.77e-44 | 21/259/0 |
| Tuned vs untuned RNG | MTR_B | lower | 280 | -1.1133 | [-1.2485, -0.9781] | -0.9686 | 4.33e-42 | 28/252/0 |
| Tuned vs untuned RNG | Util_H | higher | 280 | -0.0021 | [-0.0064, 0.0021] | -0.0599 | 0.317 | 83/92/105 |
| Tuned vs untuned RNG | Util_T | higher | 280 | 0.0224 | [0.0180, 0.0268] | 0.6043 | 1.10e-20 | 134/19/127 |
| Tuned vs untuned RNG | Util_B | higher | 280 | 0.0101 | [0.0067, 0.0136] | 0.3465 | 1.82e-08 | 116/68/96 |
| Tuned vs untuned RNG | Dead_H | lower | 280 | -0.0021 | [-0.0064, 0.0021] | -0.0599 | 0.317 | 83/92/105 |
| Tuned vs untuned RNG | Dead_T | lower | 280 | 0.0224 | [0.0180, 0.0268] | 0.6043 | 1.10e-20 | 134/19/127 |

Supplementary Table S12 (continued)

| Comparison | Metric | Better | n | Effect favoring comparator | 95% CI | dz | raw p | comp/ref/tie |
|---|---|---|---|---|---|---|---|---|
| Tuned vs untuned RNG | Dead_B | lower | 280 | 0.0101 | [0.0067, 0.0136] | 0.3465 | 1.82e-08 | 117/68/95 |

Supplementary Table S13: Full matched topology diagnostic means
by profile, dataset, and topology.

Rows report the mean value across the final matched random seeds for each profile, dataset, topology, and full-sampling setting. `untuned` denotes the untuned reference profile using XPySOM-like default hyperparameters and `tuned` denotes the fixed QE-tuned profile. Lower is better for QE, MTR, and dead-node fraction; higher is better for node utilization. Metric suffixes denote holdout (`_H`), train (`_T`), and balanced train-holdout (`_B`) summaries.

| Profile | Dataset | Topology | n | QE_H | QE_T | QE_B | MTR_H | MTR_T | MTR_B |
|---------|---------|----------|---|------|------|------|-------|-------|-------|
| untuned | blobs | hexagonal | 20 | 0.1321 | 0.1313 | 0.1317 | 3.5132 | 3.5212 | 3.5172 |
| untuned | blobs | mst | 20 | 0.1341 | 0.1324 | 0.1332 | 5.2326 | 5.1119 | 5.1723 |
| untuned | blobs | rng | 20 | 0.1325 | 0.1307 | 0.1316 | 2.7535 | 2.7181 | 2.7358 |
| untuned | breast_cancer | hexagonal | 20 | 2.6474 | 2.2132 | 2.4303 | 9.9298 | 6.7921 | 8.3610 |
| untuned | breast_cancer | mst | 20 | 2.6624 | 2.0732 | 2.3678 | 13.4902 | 7.3257 | 10.4079 |
| untuned | breast_cancer | rng | 20 | 2.6515 | 2.0799 | 2.3657 | 7.7404 | 4.1496 | 5.9450 |
| untuned | california_housing | hexagonal | 20 | 0.7910 | 0.7856 | 0.7883 | 8.4626 | 8.3360 | 8.3993 |
| untuned | california_housing | mst | 20 | 0.7600 | 0.7451 | 0.7526 | 7.8776 | 7.6936 | 7.7856 |
| untuned | california_housing | rng | 20 | 0.7646 | 0.7522 | 0.7584 | 4.5736 | 4.4653 | 4.5195 |
| untuned | circles | hexagonal | 20 | 0.1611 | 0.1590 | 0.1601 | 3.5646 | 3.5441 | 3.5543 |
| untuned | circles | mst | 20 | 0.1653 | 0.1620 | 0.1636 | 7.2056 | 6.9061 | 7.0558 |
| untuned | circles | rng | 20 | 0.1627 | 0.1599 | 0.1613 | 2.8684 | 2.8279 | 2.8481 |
| untuned | covertype | hexagonal | 20 | 3.2499 | 3.2515 | 3.2507 | 3.6170 | 3.6102 | 3.6136 |
| untuned | covertype | mst | 20 | 2.4536 | 2.4535 | 2.4536 | 2.3593 | 2.3616 | 2.3604 |
| untuned | covertype | rng | 20 | 2.6068 | 2.6067 | 2.6068 | 2.4747 | 2.4737 | 2.4742 |
| untuned | diabetes | hexagonal | 20 | 1.6952 | 1.3289 | 1.5121 | 13.1182 | 7.7649 | 10.4416 |
| untuned | diabetes | mst | 20 | 1.6718 | 1.2181 | 1.4450 | 14.5970 | 7.9549 | 11.2760 |
| untuned | diabetes | rng | 20 | 1.6803 | 1.2362 | 1.4583 | 9.7368 | 4.1324 | 6.9346 |
| untuned | digits | hexagonal | 20 | 4.5856 | 4.2735 | 4.4296 | 7.6155 | 6.3931 | 7.0043 |
| untuned | digits | mst | 20 | 4.4317 | 3.9977 | 4.2147 | 7.2430 | 5.5079 | 6.3754 |
| untuned | digits | rng | 20 | 4.4481 | 4.0470 | 4.2475 | 5.0653 | 3.8881 | 4.4767 |
| untuned | iris | hexagonal | 20 | 0.3657 | 0.1823 | 0.2740 | 7.8578 | 5.3890 | 6.6234 |
| untuned | iris | mst | 20 | 0.3609 | 0.1138 | 0.2374 | 5.7900 | 1.8655 | 3.8277 |
| untuned | iris | rng | 20 | 0.3563 | 0.1149 | 0.2356 | 3.4344 | 1.7062 | 2.5703 |
| untuned | kddcup99 | hexagonal | 20 | 0.4678 | 0.4662 | 0.4670 | 3.1121 | 3.1058 | 3.1089 |
| untuned | kddcup99 | mst | 20 | 0.3386 | 0.3365 | 0.3376 | 1.7243 | 1.7225 | 1.7234 |

| Profile | Dataset | Topology | n | QE_H | QE_T | QE_B | MTR_H | MTR_T | MTR_B |
|---|---|---|---|---|---|---|---|---|---|
| untuned | kddcup99 | rng | 20 | 0.3730 | 0.3712 | 0.3721 | 2.9179 | 2.9130 | 2.9155 |
| untuned | moons | hexagonal | 20 | 0.1355 | 0.1346 | 0.1351 | 3.1842 | 3.1787 | 3.1815 |
| untuned | moons | mst | 20 | 0.1376 | 0.1356 | 0.1366 | 4.8419 | 4.7384 | 4.7902 |
| untuned | moons | rng | 20 | 0.1357 | 0.1342 | 0.1349 | 2.6343 | 2.6125 | 2.6234 |
| untuned | olivetti_faces | hexagonal | 20 | 42.6773 | 34.2471 | 38.4622 | 8.9338 | 5.0171 | 6.9754 |
| untuned | olivetti_faces | mst | 20 | 41.6810 | 30.7001 | 36.1905 | 10.5162 | 4.4246 | 7.4704 |
| untuned | olivetti_faces | rng | 20 | 41.5030 | 30.9796 | 36.2413 | 6.9887 | 2.7613 | 4.8750 |
| untuned | s_curve | hexagonal | 20 | 0.3334 | 0.3319 | 0.3327 | 7.8608 | 7.8535 | 7.8571 |
| untuned | s_curve | mst | 20 | 0.3193 | 0.3155 | 0.3174 | 7.0170 | 6.7520 | 6.8845 |
| untuned | s_curve | rng | 20 | 0.3211 | 0.3180 | 0.3195 | 2.9213 | 2.8719 | 2.8966 |
| untuned | swiss_roll | hexagonal | 20 | 0.2902 | 0.2889 | 0.2896 | 8.7591 | 8.7911 | 8.7751 |
| untuned | swiss_roll | mst | 20 | 0.2725 | 0.2695 | 0.2710 | 9.4331 | 9.1974 | 9.3152 |
| untuned | swiss_roll | rng | 20 | 0.2721 | 0.2695 | 0.2708 | 2.8013 | 2.7562 | 2.7788 |
| untuned | wine | hexagonal | 20 | 1.9354 | 1.1334 | 1.5344 | 7.9569 | 3.7968 | 5.8769 |
| untuned | wine | mst | 20 | 1.9459 | 0.8674 | 1.4066 | 8.7875 | 2.6270 | 5.7073 |
| untuned | wine | rng | 20 | 1.9270 | 0.8784 | 1.4027 | 7.0421 | 2.0440 | 4.5430 |
| tuned | blobs | hexagonal | 20 | 0.1284 | 0.1258 | 0.1271 | 27.4818 | 27.5131 | 27.4974 |
| tuned | blobs | mst | 20 | 0.1285 | 0.1255 | 0.1270 | 6.7794 | 6.6317 | 6.7055 |
| tuned | blobs | rng | 20 | 0.1283 | 0.1254 | 0.1269 | 3.6686 | 3.6021 | 3.6353 |
| tuned | breast_cancer | hexagonal | 20 | 2.6571 | 1.9210 | 2.2891 | 35.9418 | 32.7551 | 34.3485 |
| tuned | breast_cancer | mst | 20 | 2.6625 | 1.8673 | 2.2649 | 15.7822 | 10.7114 | 13.2468 |
| tuned | breast_cancer | rng | 20 | 2.6530 | 1.8688 | 2.2609 | 10.5715 | 6.2888 | 8.4302 |
| tuned | california_housing | hexagonal | 20 | 0.7416 | 0.7197 | 0.7307 | 36.2493 | 36.0668 | 36.1580 |
| tuned | california_housing | mst | 20 | 0.7384 | 0.7150 | 0.7267 | 10.4011 | 10.1778 | 10.2894 |
| tuned | california_housing | rng | 20 | 0.7389 | 0.7162 | 0.7275 | 6.3594 | 6.2047 | 6.2821 |
| tuned | circles | hexagonal | 20 | 0.1560 | 0.1526 | 0.1543 | 31.0386 | 30.9675 | 31.0030 |
| tuned | circles | mst | 20 | 0.1566 | 0.1525 | 0.1545 | 9.9500 | 9.5644 | 9.7572 |
| tuned | circles | rng | 20 | 0.1562 | 0.1523 | 0.1542 | 3.7441 | 3.6653 | 3.7047 |
| tuned | covertype | hexagonal | 20 | 2.2611 | 2.2604 | 2.2608 | 27.3364 | 27.3276 | 27.3320 |
| tuned | covertype | mst | 20 | 2.2210 | 2.2207 | 2.2209 | 3.1564 | 3.1564 | 3.1564 |
| tuned | covertype | rng | 20 | 2.1854 | 2.1835 | 2.1844 | 2.8206 | 2.8178 | 2.8192 |
| tuned | diabetes | hexagonal | 20 | 1.6791 | 1.0986 | 1.3888 | 41.7611 | 38.9399 | 40.3505 |

| Profile | Dataset | Topology | n | QE_H | QE_T | QE_B | MTR_H | MTR_T | MTR_B |
|---|---|---|---|---|---|---|---|---|---|
| tuned | diabetes | mst | 20 | 1.6773 | 1.0906 | 1.3840 | 16.6558 | 9.5850 | 13.1204 |
| tuned | diabetes | rng | 20 | 1.6769 | 1.0967 | 1.3868 | 12.2733 | 6.2114 | 9.2424 |
| tuned | digits | hexagonal | 20 | 4.4170 | 3.8544 | 4.1357 | 36.8784 | 36.8120 | 36.8452 |
| tuned | digits | mst | 20 | 4.3875 | 3.8143 | 4.1009 | 7.6847 | 6.6297 | 7.1572 |
| tuned | digits | rng | 20 | 4.3877 | 3.8152 | 4.1014 | 6.3308 | 5.2171 | 5.7740 |
| tuned | iris | hexagonal | 20 | 0.3650 | 0.0874 | 0.2262 | 30.7106 | 24.6950 | 27.7028 |
| tuned | iris | mst | 20 | 0.3688 | 0.0814 | 0.2251 | 5.6300 | 2.0910 | 3.8605 |
| tuned | iris | rng | 20 | 0.3552 | 0.0611 | 0.2081 | 4.5694 | 1.8345 | 3.2020 |
| tuned | kddcup99 | hexagonal | 20 | 0.3047 | 0.2982 | 0.3014 | 4.9441 | 4.9339 | 4.9390 |
| tuned | kddcup99 | mst | 20 | 0.2893 | 0.2826 | 0.2860 | 2.0398 | 2.0344 | 2.0371 |
| tuned | kddcup99 | rng | 20 | 0.2934 | 0.2881 | 0.2908 | 1.8173 | 1.8160 | 1.8166 |
| tuned | moons | hexagonal | 20 | 0.1317 | 0.1292 | 0.1304 | 28.7197 | 28.5455 | 28.6326 |
| tuned | moons | mst | 20 | 0.1315 | 0.1288 | 0.1301 | 6.9319 | 6.7384 | 6.8351 |
| tuned | moons | rng | 20 | 0.1315 | 0.1285 | 0.1300 | 3.5884 | 3.5222 | 3.5553 |
| tuned | olivetti_faces | hexagonal | 20 | 41.7747 | 27.3023 | 34.5385 | 43.2448 | 41.6179 | 42.4314 |
| tuned | olivetti_faces | mst | 20 | 41.3166 | 26.3826 | 33.8496 | 11.8723 | 6.8992 | 9.3857 |
| tuned | olivetti_faces | rng | 20 | 41.4380 | 26.7705 | 34.1043 | 8.7106 | 4.3695 | 6.5400 |
| tuned | s_curve | hexagonal | 20 | 0.3070 | 0.3016 | 0.3043 | 27.9434 | 27.9687 | 27.9560 |
| tuned | s_curve | mst | 20 | 0.3060 | 0.3005 | 0.3032 | 8.9435 | 8.6914 | 8.8175 |
| tuned | s_curve | rng | 20 | 0.3059 | 0.3005 | 0.3032 | 3.9253 | 3.8508 | 3.8881 |
| tuned | swiss_roll | hexagonal | 20 | 0.2472 | 0.2430 | 0.2451 | 32.9706 | 32.9121 | 32.9413 |
| tuned | swiss_roll | mst | 20 | 0.2437 | 0.2394 | 0.2415 | 9.2023 | 8.9054 | 9.0538 |
| tuned | swiss_roll | rng | 20 | 0.2427 | 0.2388 | 0.2407 | 3.6234 | 3.5686 | 3.5960 |
| tuned | wine | hexagonal | 20 | 1.9856 | 0.6770 | 1.3313 | 32.2560 | 27.5770 | 29.9165 |
| tuned | wine | mst | 20 | 1.9633 | 0.5493 | 1.2563 | 10.4343 | 3.0798 | 6.7570 |
| tuned | wine | rng | 20 | 1.9598 | 0.4868 | 1.2233 | 9.8250 | 2.6490 | 6.2370 |

Under review as submission to TMLR

| Profile | Dataset | Topology | n | Util_H | Util_T | Util_B | Dead_H | Dead_T | Dead_B |
|---|---|---|---|---|---|---|---|---|---|
| untuned | blobs | hexagonal | 20 | 1.0000 | 1.0000 | 1.0000 | 0.0000 | 0.0000 | 0.0000 |
| untuned | blobs | mst | 20 | 1.0000 | 1.0000 | 1.0000 | 0.0000 | 0.0000 | 0.0000 |
| untuned | blobs | rng | 20 | 1.0000 | 1.0000 | 1.0000 | 0.0000 | 0.0000 | 0.0000 |
| untuned | breast_cancer | hexagonal | 20 | 0.7930 | 0.9930 | 0.8930 | 0.2070 | 0.0070 | 0.1070 |
| untuned | breast_cancer | mst | 20 | 0.7445 | 0.9935 | 0.8690 | 0.2555 | 0.0065 | 0.1310 |
| untuned | breast_cancer | rng | 20 | 0.7660 | 0.9940 | 0.8800 | 0.2340 | 0.0060 | 0.1200 |
| untuned | california_housing | hexagonal | 20 | 0.9995 | 1.0000 | 0.9998 | 5.00e-04 | 0.0000 | 2.50e-04 |
| untuned | california_housing | mst | 20 | 0.9930 | 1.0000 | 0.9965 | 0.0070 | 0.0000 | 0.0035 |
| untuned | california_housing | rng | 20 | 0.9935 | 1.0000 | 0.9967 | 0.0065 | 0.0000 | 0.0033 |
| untuned | circles | hexagonal | 20 | 1.0000 | 1.0000 | 1.0000 | 0.0000 | 0.0000 | 0.0000 |
| untuned | circles | mst | 20 | 1.0000 | 1.0000 | 1.0000 | 0.0000 | 0.0000 | 0.0000 |
| untuned | circles | rng | 20 | 1.0000 | 1.0000 | 1.0000 | 0.0000 | 0.0000 | 0.0000 |
| untuned | covertype | hexagonal | 20 | 0.6585 | 0.6635 | 0.6610 | 0.3415 | 0.3365 | 0.3390 |
| untuned | covertype | mst | 20 | 0.8460 | 0.8470 | 0.8465 | 0.1540 | 0.1530 | 0.1535 |
| untuned | covertype | rng | 20 | 0.8545 | 0.8550 | 0.8547 | 0.1455 | 0.1450 | 0.1453 |
| untuned | diabetes | hexagonal | 20 | 0.6970 | 0.9295 | 0.8133 | 0.3030 | 0.0705 | 0.1867 |
| untuned | diabetes | mst | 20 | 0.7015 | 0.9725 | 0.8370 | 0.2985 | 0.0275 | 0.1630 |
| untuned | diabetes | rng | 20 | 0.7065 | 0.9815 | 0.8440 | 0.2935 | 0.0185 | 0.1560 |
| untuned | digits | hexagonal | 20 | 0.9205 | 0.9665 | 0.9435 | 0.0795 | 0.0335 | 0.0565 |
| untuned | digits | mst | 20 | 0.9515 | 0.9950 | 0.9732 | 0.0485 | 0.0050 | 0.0268 |
| untuned | digits | rng | 20 | 0.9625 | 0.9980 | 0.9803 | 0.0375 | 0.0020 | 0.0198 |
| untuned | iris | hexagonal | 20 | 0.3315 | 0.6690 | 0.5003 | 0.6685 | 0.3310 | 0.4997 |
| untuned | iris | mst | 20 | 0.3330 | 0.7785 | 0.5557 | 0.6670 | 0.2215 | 0.4442 |
| untuned | iris | rng | 20 | 0.3385 | 0.7910 | 0.5647 | 0.6615 | 0.2090 | 0.4353 |
| untuned | kddcup99 | hexagonal | 20 | 0.8850 | 0.9140 | 0.8995 | 0.1150 | 0.0860 | 0.1005 |
| untuned | kddcup99 | mst | 20 | 0.9760 | 0.9770 | 0.9765 | 0.0240 | 0.0230 | 0.0235 |
| untuned | kddcup99 | rng | 20 | 0.9270 | 0.9310 | 0.9290 | 0.0730 | 0.0690 | 0.0710 |
| untuned | moons | hexagonal | 20 | 1.0000 | 1.0000 | 1.0000 | 0.0000 | 0.0000 | 0.0000 |
| untuned | moons | mst | 20 | 1.0000 | 1.0000 | 1.0000 | 0.0000 | 0.0000 | 0.0000 |
| untuned | moons | rng | 20 | 1.0000 | 1.0000 | 1.0000 | 0.0000 | 0.0000 | 0.0000 |
| untuned | olivetti_faces | hexagonal | 20 | 0.6115 | 0.8175 | 0.7145 | 0.3885 | 0.1825 | 0.2855 |
| untuned | olivetti_faces | mst | 20 | 0.6285 | 0.8775 | 0.7530 | 0.3715 | 0.1225 | 0.2470 |

| Profile | Dataset | Topology | n | Util_H | Util_T | Util_B | Dead_H | Dead_T | Dead_B |
|---------|---------|----------|---|--------|--------|--------|--------|--------|--------|
| untuned | olivetti_faces | rng | 20 | 0.6325 | 0.8845 | 0.7585 | 0.3675 | 0.1155 | 0.2415 |
| untuned | s_curve | hexagonal | 20 | 0.9990 | 1.0000 | 0.9995 | 0.0010 | 0.0000 | 5.00e-04 |
| untuned | s_curve | mst | 20 | 1.0000 | 1.0000 | 1.0000 | 0.0000 | 0.0000 | 0.0000 |
| untuned | s_curve | rng | 20 | 1.0000 | 1.0000 | 1.0000 | 0.0000 | 0.0000 | 0.0000 |
| untuned | swiss_roll | hexagonal | 20 | 0.9325 | 0.9385 | 0.9355 | 0.0675 | 0.0615 | 0.0645 |
| untuned | swiss_roll | mst | 20 | 0.9240 | 0.9265 | 0.9252 | 0.0760 | 0.0735 | 0.0747 |
| untuned | swiss_roll | rng | 20 | 0.9645 | 0.9655 | 0.9650 | 0.0355 | 0.0345 | 0.0350 |
| untuned | wine | hexagonal | 20 | 0.3885 | 0.7155 | 0.5520 | 0.6115 | 0.2845 | 0.4480 |
| untuned | wine | mst | 20 | 0.3810 | 0.8185 | 0.5998 | 0.6190 | 0.1815 | 0.4003 |
| untuned | wine | rng | 20 | 0.3865 | 0.8120 | 0.5992 | 0.6135 | 0.1880 | 0.4008 |
| tuned | blobs | hexagonal | 20 | 1.0000 | 1.0000 | 1.0000 | 0.0000 | 0.0000 | 0.0000 |
| tuned | blobs | mst | 20 | 1.0000 | 1.0000 | 1.0000 | 0.0000 | 0.0000 | 0.0000 |
| tuned | blobs | rng | 20 | 1.0000 | 1.0000 | 1.0000 | 0.0000 | 0.0000 | 0.0000 |
| tuned | breast_cancer | hexagonal | 20 | 0.7165 | 0.9910 | 0.8538 | 0.2835 | 0.0090 | 0.1462 |
| tuned | breast_cancer | mst | 20 | 0.6955 | 0.9925 | 0.8440 | 0.3045 | 0.0075 | 0.1560 |
| tuned | breast_cancer | rng | 20 | 0.7120 | 0.9985 | 0.8553 | 0.2880 | 0.0015 | 0.1447 |
| tuned | california_housing | hexagonal | 20 | 0.9835 | 1.0000 | 0.9918 | 0.0165 | 0.0000 | 0.0083 |
| tuned | california_housing | mst | 20 | 0.9800 | 1.0000 | 0.9900 | 0.0200 | 0.0000 | 0.0100 |
| tuned | california_housing | rng | 20 | 0.9790 | 1.0000 | 0.9895 | 0.0210 | 0.0000 | 0.0105 |
| tuned | circles | hexagonal | 20 | 1.0000 | 1.0000 | 1.0000 | 0.0000 | 0.0000 | 0.0000 |
| tuned | circles | mst | 20 | 1.0000 | 1.0000 | 1.0000 | 0.0000 | 0.0000 | 0.0000 |
| tuned | circles | rng | 20 | 1.0000 | 1.0000 | 1.0000 | 0.0000 | 0.0000 | 0.0000 |
| tuned | covertype | hexagonal | 20 | 0.7955 | 0.7965 | 0.7960 | 0.2045 | 0.2035 | 0.2040 |
| tuned | covertype | mst | 20 | 0.8725 | 0.8715 | 0.8720 | 0.1275 | 0.1285 | 0.1280 |
| tuned | covertype | rng | 20 | 0.8865 | 0.8870 | 0.8867 | 0.1135 | 0.1130 | 0.1132 |
| tuned | diabetes | hexagonal | 20 | 0.6960 | 0.9800 | 0.8380 | 0.3040 | 0.0200 | 0.1620 |
| tuned | diabetes | mst | 20 | 0.6835 | 0.9995 | 0.8415 | 0.3165 | 5.00e-04 | 0.1585 |
| tuned | diabetes | rng | 20 | 0.6825 | 0.9970 | 0.8397 | 0.3175 | 0.0030 | 0.1603 |
| tuned | digits | hexagonal | 20 | 0.9050 | 0.9735 | 0.9393 | 0.0950 | 0.0265 | 0.0607 |
| tuned | digits | mst | 20 | 0.9415 | 0.9970 | 0.9692 | 0.0585 | 0.0030 | 0.0307 |
| tuned | digits | rng | 20 | 0.9365 | 0.9955 | 0.9660 | 0.0635 | 0.0045 | 0.0340 |
| tuned | iris | hexagonal | 20 | 0.3215 | 0.7235 | 0.5225 | 0.6785 | 0.2765 | 0.4775 |

| Profile | Dataset | Topology | n | Util_H | Util_T | Util_B | Dead_H | Dead_T | Dead_B |
|---------|---------|----------|---|--------|--------|--------|--------|--------|--------|
| tuned | iris | mst | 20 | 0.3220 | 0.7600 | 0.5410 | 0.6780 | 0.2400 | 0.4590 |
| tuned | iris | rng | 20 | 0.3315 | 0.8020 | 0.5668 | 0.6685 | 0.1980 | 0.4332 |
| tuned | kddcup99 | hexagonal | 20 | 0.9540 | 0.9695 | 0.9617 | 0.0460 | 0.0305 | 0.0382 |
| tuned | kddcup99 | mst | 20 | 0.9620 | 0.9780 | 0.9700 | 0.0380 | 0.0220 | 0.0300 |
| tuned | kddcup99 | rng | 20 | 0.9625 | 0.9740 | 0.9682 | 0.0375 | 0.0260 | 0.0318 |
| tuned | moons | hexagonal | 20 | 1.0000 | 1.0000 | 1.0000 | 0.0000 | 0.0000 | 0.0000 |
| tuned | moons | mst | 20 | 1.0000 | 1.0000 | 1.0000 | 0.0000 | 0.0000 | 0.0000 |
| tuned | moons | rng | 20 | 1.0000 | 1.0000 | 1.0000 | 0.0000 | 0.0000 | 0.0000 |
| tuned | olivetti_faces | hexagonal | 20 | 0.6285 | 0.9430 | 0.7857 | 0.3715 | 0.0570 | 0.2142 |
| tuned | olivetti_faces | mst | 20 | 0.6380 | 0.9770 | 0.8075 | 0.3620 | 0.0230 | 0.1925 |
| tuned | olivetti_faces | rng | 20 | 0.6360 | 0.9640 | 0.8000 | 0.3640 | 0.0360 | 0.2000 |
| tuned | s_curve | hexagonal | 20 | 1.0000 | 1.0000 | 1.0000 | 0.0000 | 0.0000 | 0.0000 |
| tuned | s_curve | mst | 20 | 1.0000 | 1.0000 | 1.0000 | 0.0000 | 0.0000 | 0.0000 |
| tuned | s_curve | rng | 20 | 1.0000 | 1.0000 | 1.0000 | 0.0000 | 0.0000 | 0.0000 |
| tuned | swiss_roll | hexagonal | 20 | 0.9665 | 0.9665 | 0.9665 | 0.0335 | 0.0335 | 0.0335 |
| tuned | swiss_roll | mst | 20 | 0.9955 | 0.9955 | 0.9955 | 0.0045 | 0.0045 | 0.0045 |
| tuned | swiss_roll | rng | 20 | 0.9920 | 0.9920 | 0.9920 | 0.0080 | 0.0080 | 0.0080 |
| tuned | wine | hexagonal | 20 | 0.3710 | 0.7955 | 0.5833 | 0.6290 | 0.2045 | 0.4168 |
| tuned | wine | mst | 20 | 0.3760 | 0.8790 | 0.6275 | 0.6240 | 0.1210 | 0.3725 |
| tuned | wine | rng | 20 | 0.3835 | 0.9160 | 0.6498 | 0.6165 | 0.0840 | 0.3502 |

Supplementary Table S14: Matched local CuPy and Ray streaming execution-path diagnostics.

Values are paired differences computed as Ray streaming minus local CuPy for matched dataset, seed, topology, and tuned configuration. Each row summarizes the paired differences across 14 datasets and 10 seeds. The table reports the mean paired difference, standard deviation of the paired difference, 95% confidence interval, and Benjamini-Hochberg adjusted q-value.

| Topology | Metric | Mean difference | SD difference | 95% CI | BH q |
|----------|--------|-----------------|---------------|--------|------|
| hexagonal | Balanced QE | 3.08e-05 | 3.00e-04 | [-1.90e-05, 8.05e-05] | 0.7840 |
| hexagonal | Balanced MTR | -0.0121 | 0.1526 | [-0.0374, 0.0132] | 0.7840 |
| hexagonal | Balanced node utilization | -1.07e-04 | 0.0013 | [-3.17e-04, 1.03e-04] | 0.7840 |
| hexagonal | Balanced dead-node fraction | 1.07e-04 | 0.0013 | [-1.03e-04, 3.17e-04] | 0.7840 |
| MST | Balanced QE | -2.96e-04 | 0.0043 | [-0.0010, 4.15e-04] | 0.7840 |
| MST | Balanced MTR | -0.0153 | 0.5061 | [-0.0991, 0.0685] | 0.8317 |
| MST | Balanced node utilization | 3.21e-04 | 0.0052 | [-5.40e-04, 0.0012] | 0.7840 |
| MST | Balanced dead-node fraction | -3.21e-04 | 0.0052 | [-0.0012, 5.40e-04] | 0.7840 |
| RNG | Balanced QE | 0.0049 | 0.1260 | [-0.0160, 0.0258] | 0.8294 |
| RNG | Balanced MTR | -0.0823 | 0.5164 | [-0.1679, 0.0032] | 0.7840 |
| RNG | Balanced node utilization | -1.07e-04 | 0.0035 | [-6.90e-04, 4.76e-04] | 0.8317 |
| RNG | Balanced dead-node fraction | 1.07e-04 | 0.0035 | [-4.76e-04, 6.90e-04] | 0.8317 |

Supplementary Table S15: Post hoc matched cross-topology balanced-QE contrasts across the initial-radius sweep.

For each topology pair and radius, seed-level log $QE_B$ ratios were averaged within each dataset and then tested across the 14 equally weighted dataset effects. `QE ratio` is the exponentiated mean log ratio (comparator/reference), so values below 1 and positive improvement values favor the comparator. Confidence intervals are exponentiated 95% paired $t$ intervals. Benjamini–Hochberg q-values adjust across all 21 topology-pair–radius tests.

| Contrast | Radius | n datasets | n seed pairs | QE ratio | 95% CI | Improvement | raw p | BH q |
|---|---|---|---|---|---|---|---|---|
| MST vs Hex | 0.5 | 14 | 280 | 0.9969 | [0.9876, 1.0062] | 0.31% | 0.4801 | 0.5930 |
| RNG vs Hex | 0.5 | 14 | 280 | 0.9880 | [0.9726, 1.0036] | 1.20% | 0.1193 | 0.1927 |
| RNG vs MST | 0.5 | 14 | 280 | 0.9911 | [0.9803, 1.0020] | 0.89% | 0.1019 | 0.1784 |
| MST vs Hex | 0.75 | 14 | 280 | 0.9992 | [0.9880, 1.0105] | 0.08% | 0.8745 | 0.8745 |
| RNG vs Hex | 0.75 | 14 | 280 | 0.9930 | [0.9760, 1.0103] | 0.70% | 0.3961 | 0.5199 |
| RNG vs MST | 0.75 | 14 | 280 | 0.9938 | [0.9870, 1.0007] | 0.62% | 0.0751 | 0.1434 |
| MST vs Hex | 1.027 | 14 | 280 | 0.9849 | [0.9732, 0.9968] | 1.51% | 0.0172 | 0.0361 |
| RNG vs Hex | 1.027 | 14 | 280 | 0.9815 | [0.9674, 0.9957] | 1.85% | 0.0150 | 0.0351 |
| RNG vs MST | 1.027 | 14 | 280 | 0.9965 | [0.9847, 1.0084] | 0.35% | 0.5329 | 0.5943 |
| MST vs Hex | 1.5 | 14 | 280 | 0.9670 | [0.9458, 0.9887] | 3.30% | 0.0062 | 0.0162 |
| RNG vs Hex | 1.5 | 14 | 280 | 0.9633 | [0.9412, 0.9859] | 3.67% | 0.0041 | 0.0141 |
| RNG vs MST | 1.5 | 14 | 280 | 0.9962 | [0.9834, 1.0092] | 0.38% | 0.5377 | 0.5943 |
| MST vs Hex | 2 | 14 | 280 | 0.9536 | [0.9263, 0.9817] | 4.64% | 0.0036 | 0.0141 |
| RNG vs Hex | 2 | 14 | 280 | 0.9516 | [0.9248, 0.9791] | 4.84% | 0.0024 | 0.0141 |
| RNG vs MST | 2 | 14 | 280 | 0.9979 | [0.9870, 1.0090] | 0.21% | 0.6871 | 0.7214 |
| MST vs Hex | 3 | 14 | 280 | 0.9243 | [0.8793, 0.9716] | 7.57% | 0.0047 | 0.0141 |
| RNG vs Hex | 3 | 14 | 280 | 0.9340 | [0.8989, 0.9705] | 6.60% | 0.0020 | 0.0141 |
| RNG vs MST | 3 | 14 | 280 | 1.0105 | [0.9943, 1.0269] | -1.05% | 0.1873 | 0.2809 |
| MST vs Hex | 5 | 14 | 280 | 0.8842 | [0.8224, 0.9506] | 11.58% | 0.0028 | 0.0141 |
| RNG vs Hex | 5 | 14 | 280 | 0.8919 | [0.8416, 0.9452] | 10.81% | 0.0009 | 0.0141 |
| RNG vs MST | 5 | 14 | 280 | 1.0088 | [0.9923, 1.0255] | -0.88% | 0.2731 | 0.3824 |

**Supplementary Figure S1: FloatSOM vs XPySOM Calibration (Default Settings, MST)**

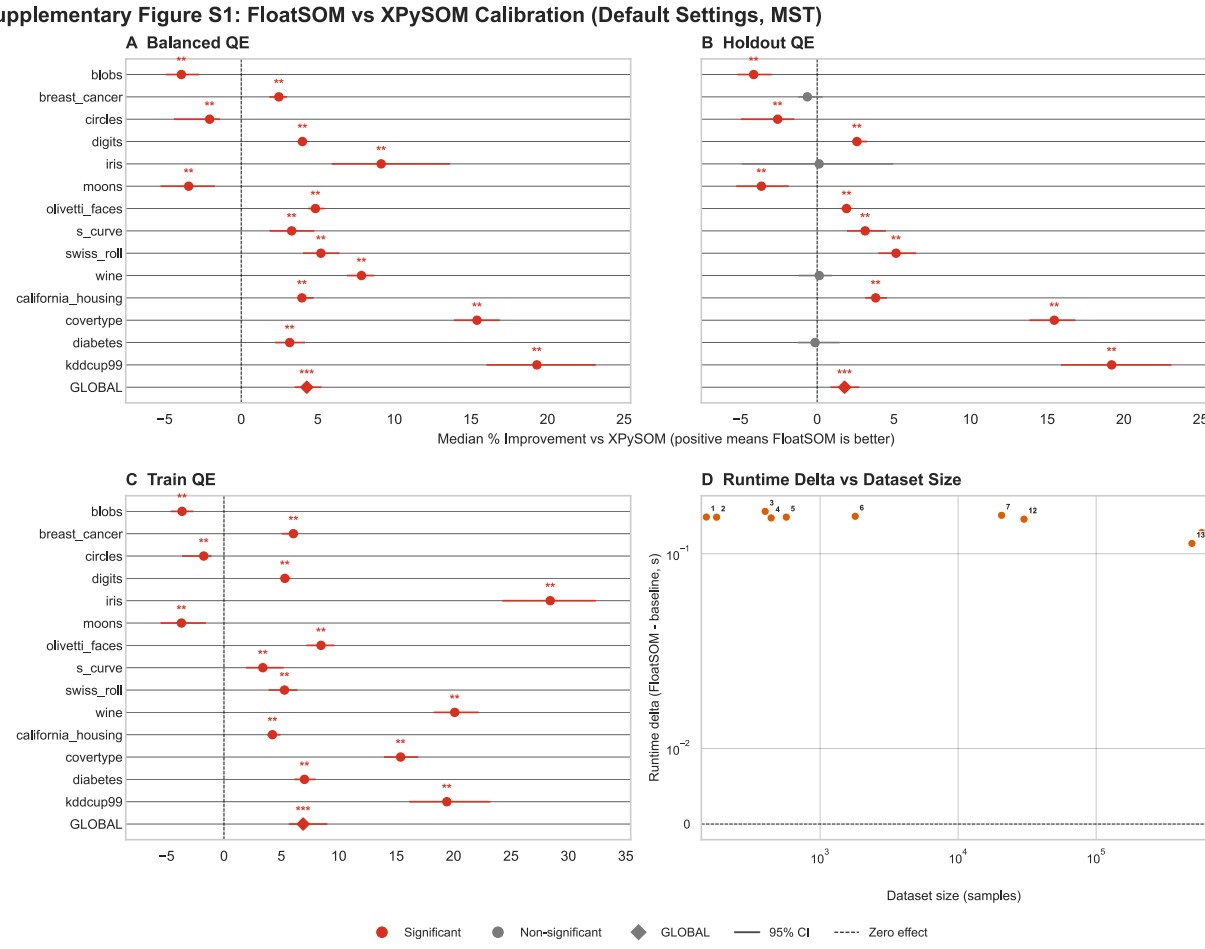

Supplementary Figure S1: Untuned FloatSOM MST versus hexagonal XPySOM. Panels A-C report paired $QE$ effects for $QE_B$, $QE_H$, and $QE_T$. Panel D reports dataset-level median runtime deltas against dataset size, where each numbered dot is the median matched-seed value of `FloatSOM time - XPySOM time`; negative values favor FloatSOM and positive values favor XPySOM. The point numbers map to Supplementary Table S4. Forest whiskers denote exact distribution-free 95% intervals for the paired Wilcoxon location estimate.

## B Supplementary Figures (End Matter)

**Supplementary Figure S2: FloatSOM vs XPySOM Calibration (Default Settings, RNG)**

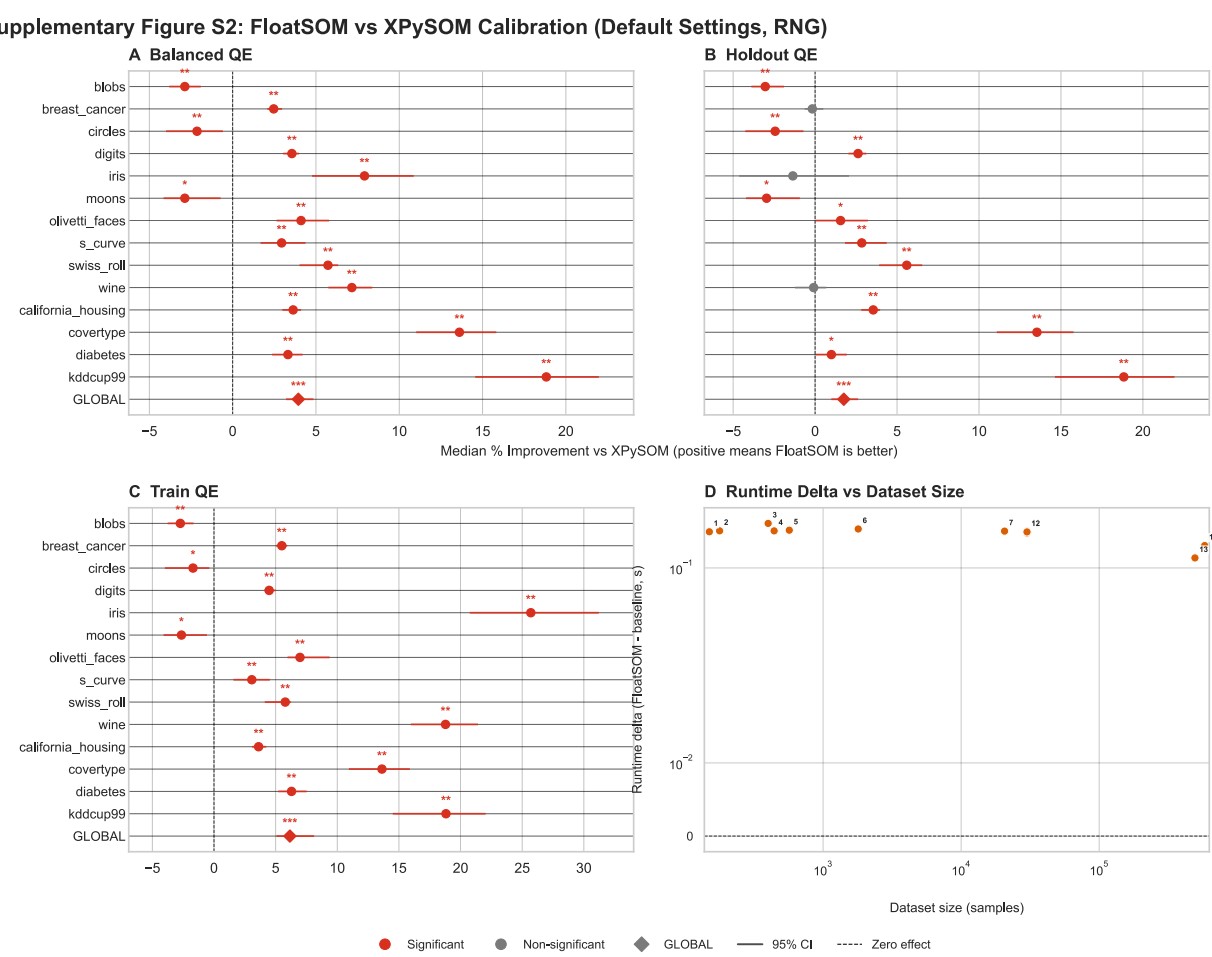

Supplementary Figure S2: Untuned FloatSOM RNG versus hexagonal XPySOM. Panels A-C report paired $QE$ effects for $QE_B$, $QE_H$, and $QE_T$. Panel D reports dataset-level median runtime deltas against dataset size, where each numbered dot is the median matched-seed value of `FloatSOM time - XPySOM time`; negative values favor FloatSOM and positive values favor XPySOM. The point numbers map to Supplementary Table S5. Forest whiskers denote exact distribution-free 95% intervals for the paired Wilcoxon location estimate.

**Supplementary Figure S3: FloatSOM vs XPySOM Calibration (Default Settings, Hexagonal)**

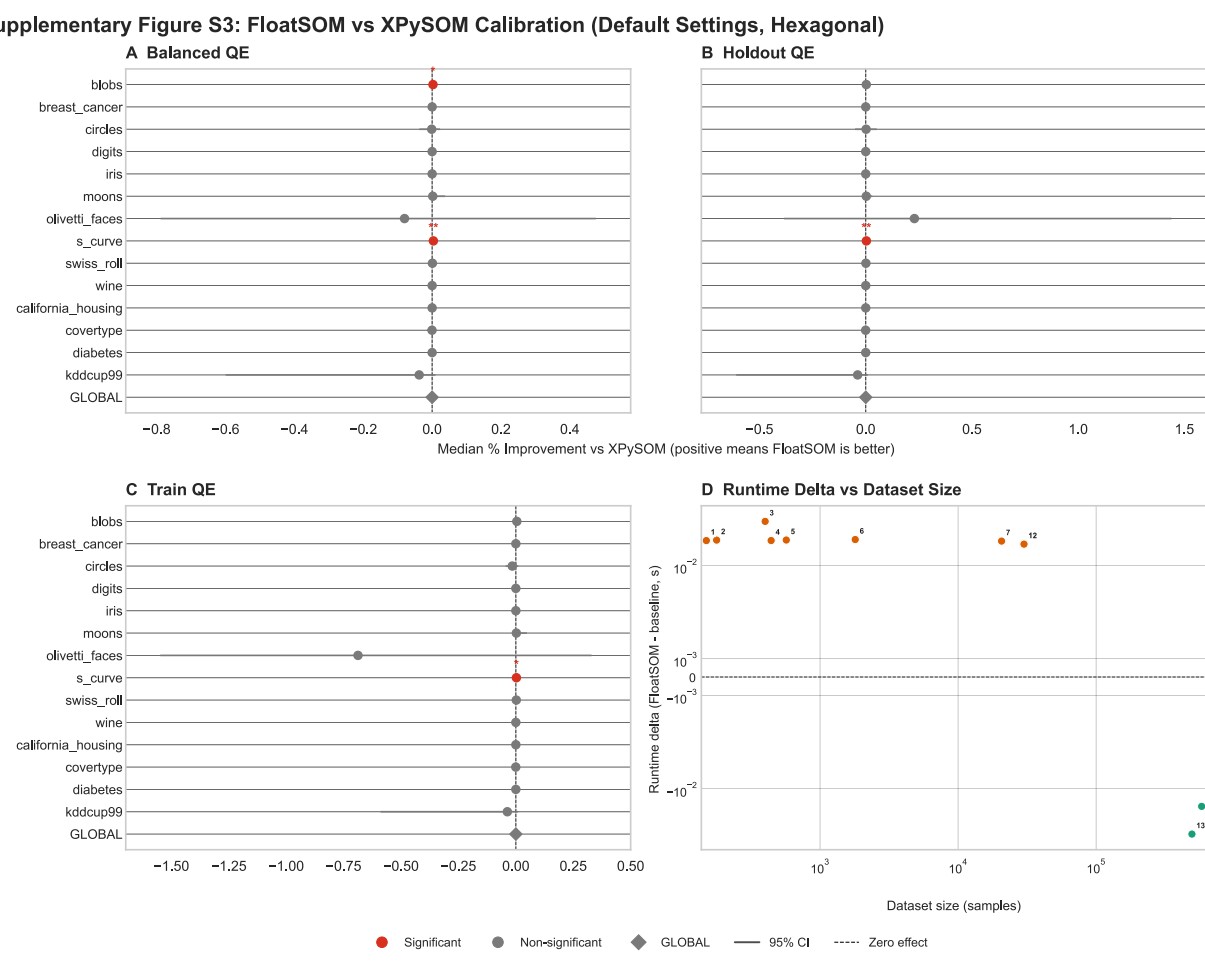

Supplementary Figure S3: Hexagonal FloatSOM–XPySOM implementation calibration. Panels A-C report paired $QE$ effects for $QE_B$, $QE_H$, and $QE_T$. Panel D reports dataset-level median runtime deltas against dataset size, where each numbered dot is the median matched-seed value of `FloatSOM time - XPySOM time`; negative values favor FloatSOM and positive values favor XPySOM. The point numbers map to Supplementary Table S6. Forest whiskers denote exact distribution-free 95% intervals for the paired Wilcoxon location estimate.

**Supplementary Figure S4: Topology Sensitivity by Contrast**

Supplementary Figure S4: Topology sensitivity analyses under full sampling. A: hexagonal versus MST; B: hexagonal versus RNG; C: MST versus RNG. Within each row, columns report matched top-$k$ paired sensitivity analyses for $QE_B$, $QE_H$, and $QE_T$ at $k \in \{1, 3, 5, 10\}$; directional labels indicate which topology is favored.

**Supplementary Figure S5: Tuned vs Untuned by Topology**

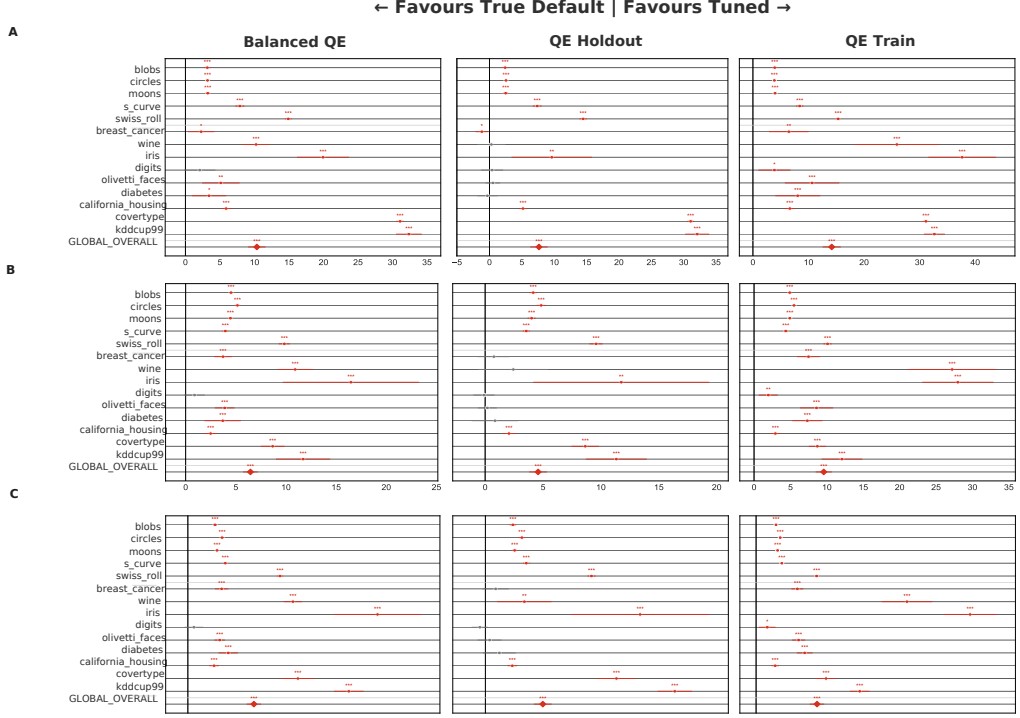

Supplementary Figure S5: Tuned versus untuned fixed-configuration comparisons by topology. A: hexagonal; B: MST; C: RNG. Within each row, columns report $QE_B$, $QE_H$, and $QE_T$ for 14 datasets and 20 seeds under matched dataset–seed–topology–split keys. Positive values indicate that the tuned configuration outperforms the untuned reference; profile definitions are given in Section 4.3.1.

**Supplementary Figure S6: GPU Scaling by Workload**
**(A-C: MST; D-F: Hexagonal)**

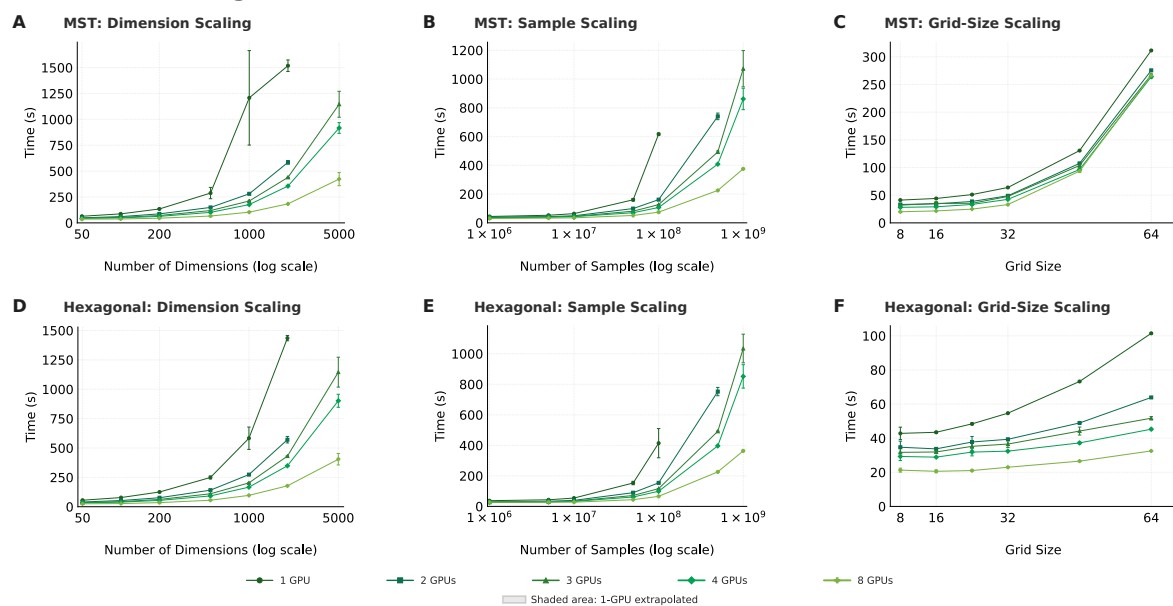

Supplementary Figure S6: Full GPU-count scaling context for MST and hexagonal under matched full sampling settings. A-C: MST dimension, sample, and grid-size scaling; D-F: hexagonal dimension, sample, and grid-size scaling. Curves correspond to $G \in \{1, 2, 4, 8\}$ GPUs and report mean wall-clock runtime (s) with $\pm 1$ standard-deviation error bars across $n = 3$ repeated runs per configuration.

**Supplementary Figure S7: Tuned Hexagonal vs XPySOM Default**

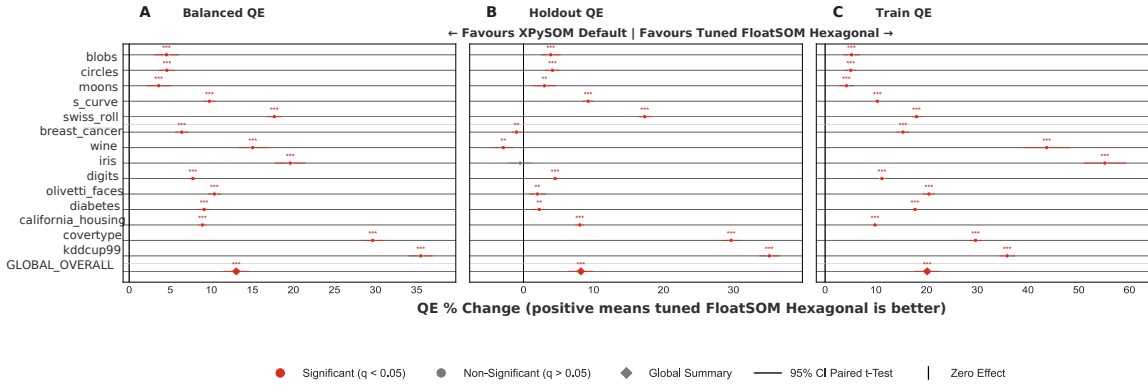

Supplementary Figure S7: Deployment comparison of untuned hexagonal XPySOM versus tuned FloatSOM hexagonal. A: $QE_B$; B: $QE_H$; C: $QE_T$ under the matched dataset/seed comparison keys. Positive values indicate tuned FloatSOM hexagonal outperforms untuned hexagonal XPySOM. Forest whiskers denote 95% paired $t$-test confidence intervals around the mean paired effect. Per-dataset and `GLOBAL_OVERALL` panel summaries are listed in Supplementary Table S9.

**Supplementary Figure S8: Tuned MST vs XPySOM Default**

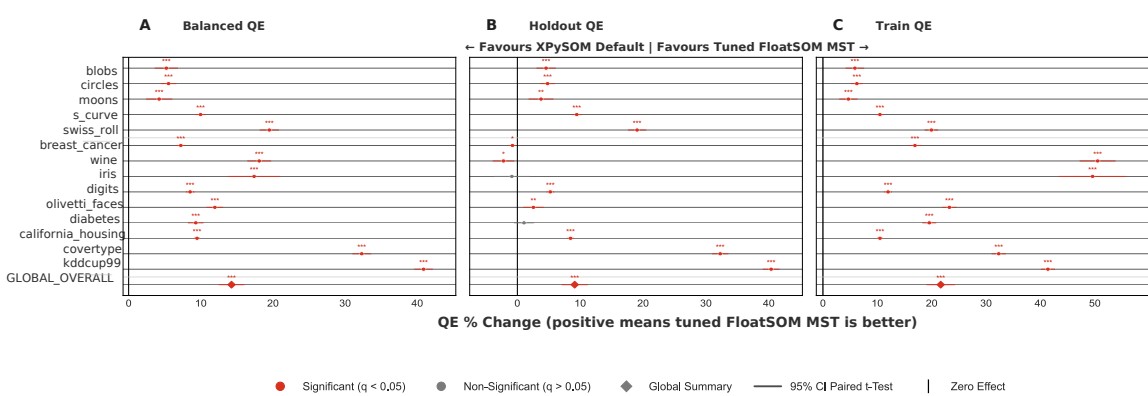

Supplementary Figure S8: Deployment comparison of untuned hexagonal XPySOM versus tuned FloatSOM MST. A: $QE_B$; B: $QE_H$; C: $QE_T$ under the matched dataset/seed comparison keys. Positive values indicate tuned FloatSOM MST outperforms untuned hexagonal XPySOM. Forest whiskers denote 95% paired $t$-test confidence intervals around the mean paired effect. Per-dataset and GLOBAL_OVERALL panel summaries are listed in Supplementary Table S10.

