# OpenReview forum: "FloatSOM: GPU Accelerated, Distributed, Topology- Flexible Self-Organizing Maps"
_TMLR — Under review for TMLR_

### Review · Reviewer_v2cg · 2026-07-04

**Summary Of Contributions:**

The paper introduces FloatSOM, a batch SOM framework for large-scale training and deployment. It has three parts. First, a systems contribution: distributed multi-GPU training with NCCL all-reduce over Ray actors, RAM spilling, and disk-backed streaming through chunked pinned-memory transfers overlapped with CUDA compute, demonstrated on 10^9 samples with 50 features on a 1024-node map in 6.16 minutes using 8 V100s across two HPC nodes. Second, graph-based topology flexibility through MST and RNG neighborhoods derived from the evolving node-weight geometry. Third, a large empirical study: an Optuna multi-objective tuning campaign, sampling comparisons, speed-scaling benchmarks, and a deployment comparison against XPySOM.

Strengths:
1. There is no widely used Python SOM that does distributed GPU training with out-of-core streaming, and the 10^9-sample run is a useful result.
2. The benchmark set is large, the statistics are paired with confidence intervals, and the XPySOM equivalence check in Section 5.1 is a good addition.
3. Limitations are included. Extrapolated 1-GPU baselines, the efficiency numbers above 100 percent that come from a memory-regime change rather than compute, and the fact that random sampling still stages full chunks.

Weaknesses:
1. The quality claim runs on QE alone. QE measures distance to the best-matching unit, not neighborhood preservation, which is the point of a SOM. No topographic error or trustworthiness, even though Forest et al., 2020 is cited and implements them.
2. Related work names prior graph and GPU SOMs but does not benchmark against any. aweSOM (Ha et al., JOSS 2025), a GPU Python SOM in the same N > 10^6 range, is not mentioned.
3. Quality numbers are not in the paper. Per-dataset effect sizes exist only as external .tsv paths (Tables S4 to S11 are captions pointing at files). They cannot be read off the forest plots either.

**Audience:**

Yes

**Audience Explanation:**

1. SOMs are still used in cytometry, single-cell work, and large tabular clustering. FlowSOM, GigaSOM, XPySOM, Somoclu, aweSOM.
2. A Python framework combining distributed GPU training with out-of-core streaming is usable in those areas.

**Broader Impact Concerns:**

Nothing.
May be the cost of the 168,000-run Optuna campaign. Not required though.

**Claims And Evidence:**

No

**Claims Explanation:**

1. Systems claims, mostly yes. Quality claims, no. These are two different things.
2. Topology comparison (Section 5.3): hexagonal, MST, RNG all inside FloatSOM under the same Optuna budget. Implementation and tuning are matched here. That part is fine.
3. The metric is not. QE drops when you loosen the lattice, and loosening the lattice is what MST and RNG do. Section 3.2.2 says the same thing, that RNG is expected to win through more freedom in its topology. That freedom is what folds the map and breaks preservation. A lower QE reads just as well as the map turning into a loose vector quantizer, which contradicts the preservation language in the introduction and Section 8.2. Not tested.
4. Deployment comparison (Section 7, Figure 13) is a different claim. Tuned FloatSOM RNG against untuned hexagonal XPySOM. One number, three changes: topology, tuning, implementation. The 14.5, 9.1, 22.5 percent gains cannot be credited to topology. Sections 5.1,  5.3, and 5.4 already have the pieces to separate them.
5. Some scaling points are extrapolated 1-GPU values, not measured (Section 4.2). Stated in the paper. Still limits the efficiency panels.

**Requested Changes:**

1. Report a preservation metric next to QE in every topology comparison. Forest et al., 2020 is already cited and has topographic error and trustworthiness.
2. Add a control that shrinks the hexagonal neighborhood radius. If a tighter-radius hexagonal map closes most of the QE gap, the graph topologies are winning on looseness, not structure.
3. Put the numbers in the paper. Embed the per-dataset effect-size tables instead of external .tsv paths.
4. Separate topology from tuning and implementation in the deployment comparison, or tune both systems.
5. Add recent GPU and distributed SOM baselines. Somoclu CUDA, GigaSOM, aweSOM. aweSOM is single-node CPU/GPU with ensemble stacking, so it is a quality baseline, not a distributed competitor.
6. Drop "novel" for the topologies. Jang et al., 2009, already in the references, put MSTs on SOMs. Frame this as the first version that scales on GPUs.

Following may be added:
1. Measure the runtime cost of the recommended topology. Section 8.5 recommends RNG. At grid size 64, RNG is 27x hexagonal, MST is 8x. The MST distances-to-CPU step for Kruskal is the cheap part. The all-pairs hop-distance step both share (line 10, Algorithms 1 and 2) plus RNG's blocker test is the expensive part. Put a number on the RNG cost.
2. State whether the paired t-tests were corrected for multiple comparisons. About 42 per topology, 14 datasets by 3 metrics. Seed counts and intervals are already there.
3. Add a dead-node or node-utilization count across the three topologies. Cheap, and it bears on the preservation question.
4. Keep the HDSSSOM results framed as the pilot they already are.

---

> ### Author Response · Authors · 2026-07-21
>
> We thank the reviewer for the careful reading and for separating the systems contribution from the quality/topology claims. We agree that the systems evidence is stronger in the submitted version than the topology-quality evidence, and the revision is structured to make that distinction explicit. Where the manuscript already contained the relevant analysis but did not state the point clearly enough, we have clarified the text. Where the reviewer identified a missing control or missing reporting detail, we have added the corresponding amendment below.
>
> ## 1. Quality and topology diagnostics
>
> > The quality claim runs on QE alone. QE measures distance to the best-matching unit, not neighborhood preservation, which is the point of a SOM. No topographic error or trustworthiness, even though Forest et al., 2020 is cited and implements them. Report a preservation metric next to QE in every topology comparison. Forest et al., 2020 is already cited and has topographic error and trustworthiness.
>
> We agree that $QE$ alone does not characterize SOM neighborhood ordering. We carefully considered the reviewer's suggestion to report topographic error. Topographic error records the proportion of samples for which the first and second best-matching units are not immediately adjacent. This binary criterion depends critically on the number of immediate neighbours and therefore does not provide a like-for-like comparison between maps with different topologies. López-Rubio and Díaz Ramos (2014) use MTR to compare alternative SOM grid topologies.
>
> We therefore added Mean Tied Rank (MTR), which groups non-winning units by their shortest-path hop distance from the first BMU and assigns units at the same hop distance their average ordinal rank. This construction accommodates the different adjacency rules of the three topologies: hexagonal adjacency is fixed by the lattice, MST adjacency is a sparse tree derived from the node weights, and RNG adjacency is weight-derived with variable node degree. In every case, MTR uses the map's own graph, counts the units fewer hops away, and accounts for the number of units tied at the second BMU's hop distance. It therefore provides a graded measure of local ordering across the three adjacency structures.
>
> The matched analysis in Table 1. found lower MTR for RNG than for hexagonal maps both with and without tuning. The fixed QE-tuned profiles had lower $QE$ but higher MTR than the untuned profiles, with a much larger MTR difference for hexagonal maps (24.34 tied-rank positions) than for MST (1.43) or RNG (1.11). The main text reports the principal effects, and Supplementary Tables S12-S13 provide the full results.
>
> We also expanded the geometric rationale. A fixed hexagonal lattice predetermines neighborhood coupling independently of the learned data-space geometry, so movement of one prototype can influence lattice neighbors that are not locally related in data space. MST minimizes this coupling and lets prototypes redistribute along irregular or elongated structures, but its tree constraint can omit additional useful local connections in dense regions. RNG is less free because nodes may influence several neighbors, yet those connections are supported by the evolving prototype geometry rather than imposed beforehand. It can therefore remain sparse where appropriate and become more mesh-like in concentrated regions. We present $QE$, MTR, node utilization, and dead-node fraction as evidence consistent with this interpretation, not as proof of a unique causal mechanism.
>
> To illustrate these neighborhood structures beyond a two-dimensional synthetic example, we expanded Fig. 5 to include matched hexagonal, MST, and RNG overlays for KDD Cup 99, trained in standardized 41-dimensional space and displayed using a shared two-dimensional PCA projection. The representative overlays also provide qualitative confirmation that the prototypes are deployed across the observed data structure. In the KDD Cup 99 panels, the hexagonal topology shows several apparently nonlocal connections across separated regions of the projected prototype distribution, whereas the MST and RNG connections more closely follow its local geometry. We interpret this cautiously because the two-dimensional PCA display can distort relationships in the original 41-dimensional space.

---

> ### Author Response · Authors · 2026-07-21
>
> ### Summary of Manuscript Changes
>
> We added MTR as a graded, topology-aware measure of local neighborhood ordering alongside $QE$, defined how it is calculated for fixed hexagonal and dynamically derived MST/RNG adjacency. Table 1 and Supplementary Tables S12--S13 now report the matched MTR and node-use diagnostics across 14 datasets and 20 seeds. The Results and Discussion now distinguish quantization fidelity from local ordering, report the stronger MTR trade-off observed for tuned hexagonal maps, and identify RNG as the strongest joint $QE$/MTR result while keeping the causal interpretation cautious. We also expanded Fig. 5 with matched KDD Cup 99 topology overlays and added a geometric explanation of fixed-lattice coupling versus graph neighborhoods derived from evolving prototypes.
>
> ## 2. Hexagonal neighborhood-radius control
>
> ### Reviewer Comment
>
> > Add a control that shrinks the hexagonal neighborhood radius. If a tighter-radius hexagonal map closes most of the QE gap, the graph topologies are winning on looseness, not structure.
>
> ### Response
>
> We agree that neighborhood radius is an important potential confound. In addition to the existing Optuna design, in which `initial_radius` was optimized independently for every topology over the same interval and search budget, we performed additional experiment where we iteratively increase the radius while keeping other settings fixed at their optimal value based on the optuna results. The results are illustrated in Figure 9.
>
> Across the entire radius range investigated, MST and RNG consistently attained lower normalised QE than hexagonal, with this being most obvious after $r=1.5$ (Fig. 9A). Regarding each topology's optimal $QE_B$ radius, RNG and MST's optimal occurred at $r=1.5$, which is larger than hexagonal at $r=0.75$. The graph-topology gains therefore cannot be explained by preferential radius treatment or by comparison with a broad default-radius hexagonal map. Even at this optimal point, hexagonal still demonstrated inferior $QE_B$ to both graph topologies, implying that decreased radius alone is not responsible for the observed $QE$ differential. Fig. 9B additionally shows that the sharp hexagonal $QE$--MTR trade-off is substantially attenuated for MST and RNG : RNG maintains lower MTR than MST while its $QE_B$ remains stable. Fig. 9C shows that MST and RNG also maintain high node utilization in this region, which does not support reduced use of map capacity as the explanation for their lower $QE_B$. In both the MTR and Node utilisation, MST and RNG perform consistently better than hexagonal.
>
> ### Summary of Manuscript Changes
>
> We clarified that `initial_radius` was optimized independently for all three topology families over the same 0.5--10.0 interval. We also added a matched seven-radius sensitivity experiment across 14 datasets and 20 seeds, holding the other settings fixed and applying multiplicity-corrected cross-topology tests. The new Fig. 9 summarizes normalized balanced $QE$, MTR, and node utilization; MST and RNG retain lower observed balanced $QE$ than hexagonal maps across the tested range, with adjusted differences from $r=1.027$ onward.

---

> > ### Author Response · Authors · 2026-07-21
> >
> > ## 3. Related work and external baselines
> >
> > ### Reviewer Comment
> >
> > > Related work names prior graph and GPU SOMs but does not benchmark against any. aweSOM (Ha et al., JOSS 2025), a GPU Python SOM in the same N > 10^6 range, is not mentioned.
> >
> > > Add recent GPU and distributed SOM baselines. Somoclu CUDA, GigaSOM, aweSOM. aweSOM is single-node CPU/GPU with ensemble stacking, so it is a quality baseline, not a distributed competitor.
> >
> > ### Response
> >
> > We agree that the baseline selection required clearer justification. We expanded the related-work discussion and attempted an aweSOM benchmark. All five aweSOM attempts reached the 1800-s timeout before completing 60% of $N$ online updates on the standard speed workload, as referenced in Section 4.2. Matching FloatSOM's 10 full batch iterations would require $10N$ pointwise updates.
> >
> > We now discuss Somoclu and GigaSOM as important parallel-systems references. Somoclu was not benchmarked against because Mancini et al. (2020) directly compared both Somoclu and SomocluGPU with XPySOM and found them to be more than 10 times slower in the reported benchmark. As we benchmark FloatSOM directly against XPySOM, we retained XPySOM as the controlled executable comparator. This published comparison does not replace a contemporary FloatSOM--Somoclu head-to-head benchmark, and we do not infer a transitive speedup from it. Similarly, GigaSOM is a CPU algorithm coded in Julia. Due to the differences in hardware (with a large CPU cluster, as opposed to our present GPU based hardware) and coding language, a true head-to-head comparison is not possible.
> >
> > ### Summary of Manuscript Changes
> >
> > We expanded the Introduction and related work to position MiniSom, aweSOM, XPySOM, Somoclu, and GigaSOM according to their training and systems models. Section 4.2 now documents the attempted aweSOM benchmark: all five runs reached the 1,800-second timeout before completing 60% of $N$ online updates, so aweSOM was not included in the timed comparison. We explain why XPySOM remains the closest executable Python/GPU batch comparator and treat Somoclu and GigaSOM as published systems context rather than fair head-to-head baselines. We also added the complete XPySOM calibration protocol and replaced the prior equivalence wording with the narrower finding that no implementation-associated hexagonal $QE$ difference was detected.

---

> > > ### Author Response · Authors · 2026-07-21
> > >
> > > ## 4. Numbers should be embedded in the paper
> > >
> > > ### Reviewer Comment
> > >
> > > > Quality numbers are not in the paper. Per-dataset effect sizes exist only as external table paths (Tables S4 to S11 are captions pointing at files). They cannot be read off the forest plots either.
> > >
> > > > Put the numbers in the paper. Embed the per-dataset effect-size tables instead of external paths.
> > >
> > > ### Response
> > >
> > > We agree. The submitted version included the tables as reproducibility artifacts but did not embed their numerical contents in the manuscript, which makes the paper harder to evaluate independently. We have replaced the path-only supplementary table captions with embedded tables for the XPySOM calibration summaries, topology effect and q-value summary, deployment effect summaries, topology runtime summary, and the new matched-radius cross-topology tests in Supplementary Table S15. The manuscript now contains the numerical values needed to evaluate the corresponding figures.
> > >
> > > ### Summary of Manuscript Changes
> > >
> > > We replaced the path-only supplementary-table captions with embedded numerical tables, including the XPySOM calibration summaries, topology effects and adjusted q-values, deployment effects, topology runtimes, and matched-radius comparisons. The revised supplement therefore contains the values needed to evaluate the corresponding plots directly, including the paired topology comparisons in Table S7 and the deployment summary in Table S8.
> > >
> > > ## 5. Deployment comparison separates topology, tuning, and implementation
> > >
> > > ### Reviewer Comment
> > >
> > > > Deployment comparison (Section 7, Figure 13) is a different claim. Tuned FloatSOM RNG against untuned hexagonal XPySOM. One number, three changes: topology, tuning, implementation. The 14.5, 9.1, 22.5 percent gains cannot be credited to topology. Sections 5.1, 5.3, and 5.4 already have the pieces to separate them.
> > >
> > > > Separate topology from tuning and implementation in the deployment comparison, or tune both systems.
> > >
> > > ### Response
> > >
> > > We agree that Fig. 15 should not be read as attributing the full gain to topology alone. It presents an integrated deployment comparison between an untuned XPySOM run and the recommended tuned FloatSOM RNG configuration. The component effects are evaluated separately in Sections 5.1, 5.3, and 5.4, but the submitted Section 7 did not make this decomposition sufficiently clear.
> > >
> > > We revised Section 7 and the Fig. 15 caption so that the 14.5%, 9.1%, and 22.5% improvements are described as an integrated deployment effect rather than a topology-only effect. We also point readers to the tuned-hexagonal and tuned-MST deployment figures in the Supplementary material.
> > >
> > > ### Summary of Manuscript Changes
> > >
> > > We revised Section 7 and the Fig. 15 caption to state explicitly that the deployment comparison combines implementation, tuning, and topology effects; the reported 14.5%, 9.1%, and 22.5% improvements are no longer presented as topology-only gains. The text points readers to Sections 5.1, 5.3, and 5.4 for the component analyses and to Supplementary Figs. S7--S8 for the corresponding tuned hexagonal and MST comparisons.
> > >
> > > ## 6. MST and RNG novelty claims
> > >
> > > ### Reviewer Comment
> > >
> > > > Drop "novel" for the topologies. Jang et al., 2009, already in the references, put MSTs on SOMs. Frame this as the first version that scales on GPUs.
> > >
> > > ### Response
> > >
> > > We agree that the submitted wording could imply that MSTs had not previously been associated with SOMs. Weno longer claim novelty for the graph object itself. Instead, the contribution is the scalable GPU implementation and large-scale evaluation of dynamically refreshed graph neighborhoods during SOM training. We retain only the narrower, qualified statement that we have not identified prior work using dynamically refreshed RNG neighborhoods for SOM training.
> > >
> > > ### Summary of Manuscript Changes
> > >
> > > We removed the broad claim that graph topologies themselves are novel and now describe the contribution as scalable, GPU-compatible implementation and large-scale evaluation of refreshed graph neighborhoods during SOM training. The related-work section distinguishes training-time MST neighborhoods from post hoc MST visualization while acknowledging prior training-time variants, and the Methods now state explicitly that MST/RNG graphs determine the update neighborhood. We retain only a qualified claim that we found no prior SOM work using dynamically refreshed RNG and MST training neighborhoods.

---

> > > > ### Author Response · Authors · 2026-07-21
> > > >
> > > > ## 7. Runtime cost of RNG recommendation
> > > >
> > > > ### Reviewer Comment
> > > >
> > > > > Measure the runtime cost of the recommended topology. Section 8.5 recommends RNG. At grid size 64, RNG is 27x hexagonal, MST is 8x. The MST distances-to-CPU step for Kruskal is the cheap part. The all-pairs hop-distance step both share plus RNG's blocker test is the expensive part. Put a number on the RNG cost.
> > > >
> > > > ### Response
> > > >
> > > > We agree that any recommendation of RNG must be paired with its runtime cost. Fig. 14 reports this cost: at grid size 64, the 8-GPU mean runtime was 32.54 s for hexagonal, 266.45 s for MST, and 880.83 s for RNG, corresponding to 8.19x and 27.07x the hexagonal runtime for MST and RNG, respectively. We now repeat these values at the point where RNG is recommended and make the grid-size limitation explicit.
> > > >
> > > > ### Summary of Manuscript Changes
> > > >
> > > > We revised the practical recommendation to make topology cost explicit and grid-size dependent. The Discussion now reports the measured 8-GPU runtimes at grid size 64—32.54 s for hexagonal, 266.45 s for MST, and 880.83 s for RNG—and recommends RNG for the strongest joint $QE$/MTR result only when its overhead is acceptable; MST is identified as the lower-cost graph option when $QE$ is the main objective or large graphs make RNG prohibitive.
> > > >
> > > > ## 8. Multiple-comparison correction
> > > >
> > > > ### Reviewer Comment
> > > >
> > > > > State whether the paired t-tests were corrected for multiple comparisons. About 42 per topology, 14 datasets by 3 metrics. Seed counts and intervals are already there.
> > > >
> > > > ### Response
> > > >
> > > > We agree. The manuscript now reports the multiple-comparison correction method used for the topology comparisons, and the associated figure and table descriptions have been updated accordingly. Only two dataset-level annotations changed from significant to non-significant: `blobs` in Fig. 7B changed from p=0.0358 to q=0.0627, and `iris` changed from p=0.0444 to q=0.0745; the remaining changes affected star levels only.
> > > >
> > > > ### Summary of Manuscript Changes
> > > >
> > > > We added a multiple-comparison policy for the topology analyses. Each hexagonal-versus-graph contrast treats the 42 dataset-level tests (14 datasets by three $QE$ metrics) as one Benjamini--Hochberg family, reports q-values alongside raw p-values, and uses q-values for figure markers and significance counts; pooled overall tests remain separate. The affected figure captions and Supplementary Table S7 were updated accordingly.
> > > >
> > > > ## 9. Dead-node and node-utilization reporting
> > > >
> > > > ### Reviewer Comment
> > > >
> > > > > Add a dead-node or node-utilization count across the three topologies. Cheap, and it bears on the preservation question.
> > > >
> > > > ### Response
> > > >
> > > > We agree, and thank the reviewer for pointing this out. Node utilization is a useful confirmatory diagnostic for checking whether the MTR result is accompanied by broadly used map capacity rather than uneven allocation in which some nodes are effectively unused. Considering this, we added node-utilization and dead-node-fraction diagnostics to the matched topology benchmark outputs, using the same matched units as the topology comparison.
> > > >
> > > > The matched tuned-profile result favored RNG rather than indicating poorer map use. Relative to tuned hexagonal maps, tuned RNG increased balanced node utilization by 0.0197 (95% CI 0.0158 to 0.0235; p=1.38e-20) and reduced balanced dead-node fraction by the same amount. We report the split-specific and balanced diagnostics separately and present node utilization as a confirmatory diagnostic alongside MTR.
> > > >
> > > > Quantitative node-utilization and dead-node evidence is shown in Fig. 9 and Supplementary Tables S12-S13. The representative overlays in Fig. 5 additionally provide qualitative confirmation that prototypes are deployed across the observed data structures, although they do not classify individual nodes as active or dead. Together, these results are consistent with the interpretation that geometry-derived neighborhoods avoid some unnecessary fixed-lattice coupling while retaining map capacity; they do not establish that interpretation as a unique causal explanation.
> > > >
> > > > ### Summary of Manuscript Changes
> > > >
> > > > We added node utilization and dead-node fraction as post hoc diagnostics on the training and holdout splits, balanced in the same way as $QE$. Table 1 and Supplementary Tables S12--S13 now report these measures with the matched topology diagnostics, allowing the topology comparison to account for whether lower $QE$ is accompanied by reduced map-capacity use.

---

> > > > > ### Author Response · Authors · 2026-07-21
> > > > >
> > > > > ## 10. HDSSSOM framing
> > > > >
> > > > > ### Reviewer Comment
> > > > >
> > > > > > Keep the HDSSSOM results framed as the pilot they already are.
> > > > >
> > > > > ### Response
> > > > >
> > > > > We agree and preserve the current framing. The submitted manuscript already describes HDSSSOM as a focused screening pilot and not as part of the main full-versus-random sampling benchmark. We kept that language and added a sentence to avoid broadening the claim beyond the pilot configuration.
> > > > >
> > > > > ### Summary of Manuscript Changes
> > > > >
> > > > > We retained HDSSSOM as a focused screening pilot and strengthened that qualification in Section 5.2. The revision now states explicitly that the smaller pilot configuration is an elimination screen under the tested schedule, not a comprehensive evaluation of all possible HDSSSOM schedules.
> > > > >
> > > > > ## 11. Scaling efficiency and extrapolated 1-GPU baselines
> > > > >
> > > > > ### Reviewer Comment
> > > > >
> > > > > > Some scaling points are extrapolated 1-GPU values, not measured (Section 4.2). Stated in the paper. Still limits the efficiency panels.
> > > > >
> > > > > ### Response
> > > > >
> > > > > We agree that extrapolated 1-GPU denominators limit the efficiency analysis. The affected 1-GPU jobs did not complete within the fixed benchmark timeout, particularly for workloads entering the disk-backed execution regime, so no measured $T_1$ was available at those points. Because scaling efficiency requires a 1-GPU denominator, we used local extrapolation from the last successful 1-GPU measurement to provide a provisional reference for plotting. These extrapolated values are not measured runtimes and should not be interpreted as validated estimates of single-GPU performance.
> > > > >
> > > > > We have therefore moderated the interpretation of the efficiency panels. Points with directly measured 1-GPU baselines provide the primary evidence for scaling efficiency. Points using extrapolated baselines are treated as descriptive estimates only, and are not used to support a general claim that efficiency improves across the full workload range. Absolute runtime and directly measured speedup remain the primary scaling results. Efficiencies above 100\% are interpreted as reflecting parallelism together with changes in memory and data-staging regime, rather than superlinear computation.
> > > > >
> > > > > ### Summary of Manuscript Changes
> > > > >
> > > > > We clarified which 1-GPU scaling denominators were measured and which were locally extrapolated because runs exceeded the benchmark timeout. Extrapolated efficiency values are now presented only as descriptive plotting references and are not used to support general scaling claims. The revised interpretation prioritizes absolute runtimes and directly measured baselines, and notes that efficiencies above 100% may reflect a change in memory or data-staging regime as well as parallel throughput.

---

### Review · Reviewer_QCRL · 2026-07-07

**Summary Of Contributions:**

FloatSOM is a GPU-accelerated framework for training Self-Organizing Maps (SOMs) at scale, combining three contributions: (1) distributed, out-of-core execution via Ray and NCCL for datasets that exceed device memory; (2) topology-flexible training using Minimum Spanning Tree (MST) and Relative Neighborhood Graph (RNG) neighborhood structures in place of fixed rectangular or hexagonal lattices; and (3) systematic Optuna-driven hyperparameter tuning applied consistently across topology and sampling configurations. The empirical program is substantial: 14 datasets, 10 seeds, 200 Optuna trials per configuration, and paired statistical testing throughout. The headline scaling result -- training a 1024-node SOM on 1,000,000,000 samples across 8 GPUs in roughly 6 minutes -- is a genuine systems achievement that existing open SOM libraries such as XPySOM, GigaSOM, and Somoclu do not match.
Overall, the paper is best understood as a systems/tooling contribution validated by careful empirical benchmarking, rather than as a paper introducing a fundamentally new algorithmic idea: MST- and RNG-based SOM topologies date back to Jang et al. (2009) and Toussaint (1980) respectively, and the paper's own framing acknowledges this. That positioning raises the bar on rigor, since every claim needs to be fully substantiated by the evidence shown, with no gap between what is asserted in the main text and what is actually demonstrated. On this front, the manuscript falls short in several places, some of which affect claims central to the paper's contribution rather than incidental details.

**Additional Comments:**

Assessment: The systems contribution and empirical scope are solid and appropriate. However, several claims central to the paper's argument (particularly the topology ordering and the joint quality/scalability claim) currently rest on evidence that is incomplete, appendix-only, or shown on a different execution path than the one being generalized about. I do not think the manuscript meets the bar for acceptance as it stands, but I believe the gaps are addressable. I would expect a revision to include: (a) the direct MST-versus-RNG comparison moved into the main text, (b) the execution-path discrepancy addressed or reconciled, (c) at least a discussion-level comparison against Somoclu/GigaSOM or an explicit justification for their absence, and (d) a stated position on multiple-comparisons correction.
Reference (acronyms used above): SOM = Self-Organizing Map; GPU = Graphics Processing Unit; MST = Minimum Spanning Tree; RNG = Relative Neighborhood Graph; NCCL = NVIDIA Collective Communications Library; QE = Quantization Error; HDSSSOM = Hierarchical Dynamic Subset Selection SOM.

**Audience:**

Yes

**Audience Explanation:**

Scalable SOM training is a practical need for practitioners working with large-scale clustering, visualization, and dimensionality-reduction workloads, and the systems engineering here (multi-GPU, out-of-core, disk-backed execution) addresses a real gap in the current open-source ecosystem (XPySOM, GigaSOM, Somoclu). The topology-flexible (MST/RNG) approach and the topology-aware hyperparameter tuning methodology would also be of interest to the broader unsupervised-learning and representation-learning community. Provided the soundness gaps above are addressed, the empirical scope and the scaling results would be a useful reference point for future SOM tooling work.

**Broader Impact Concerns:**

I do not have specific ethical concerns about this work. It is a systems/tooling contribution for unsupervised learning infrastructure, and I do not see a need for a dedicated Broader Impact Statement beyond what would be standard for general-purpose ML tooling papers.

**Claims And Evidence:**

No

**Claims Explanation:**

Several claims central to the paper's argument currently rest on evidence that is incomplete, appendix-only, or shown on a different execution path than the one being generalized about.
1. Central claims rely on evidence that is incomplete or absent in the main text. The paper's principal empirical conclusion is that RNG is the strongest topology, yet the main text never directly establishes that ordering. Section 5.3 compares MST and RNG independently against the hexagonal baseline, but the direct MST-versus-RNG comparison appears only in the supplementary material (Figs. S4-S7). Because the Discussion ultimately recommends RNG over the other topology options, this comparison is central to the paper's argument and should be presented in the main text rather than relegated to the appendix. Separately, Section 6.2 claims "broadly consistent scaling... regardless of topology," citing Figs. 11 and S11, but Fig. 11 itself shows RNG only -- the hexagonal/MST curves substantiating this sit in the supplement. More importantly, "regardless of topology" could be read as runtime parity, yet Section 6.3 reports MST and RNG running 8.19x and 27.07x slower than hexagonal at large grid sizes. If Section 6.2 means only that scaling-efficiency shape is similar across topologies (not absolute runtime), that should be stated explicitly, since as written the two sections read as contradictory.
2. Two additional soundness gaps that reorganizing the appendix would not resolve. No system named as closest related work (Somoclu, GigaSOM) is empirically compared against; only single-GPU XPySOM is benchmarked, despite the introduction's "systems gap" argument being framed explicitly around these tools. In addition, the quality benchmarks (Figs. 3-9) run on the in-memory batch path, while the scaling benchmarks (Figs. 10-13) run on the distributed path; the joint claim of "better QE and better scalability" is never shown on the same execution path, and this decoupling is not discussed.
3. Statistical reporting. With a large number of paired significance tests reported (14 datasets x 3 metrics x multiple comparisons), there is no correction for multiple comparisons and no statement of whether per-dataset significance stars are intended as primary evidence or as descriptive/exploratory annotation.
4. A screening decision is made on a lower-powered protocol. HDSSSOM is eliminated from the rest of the study (Section 5.2) using a smaller pilot (10 datasets/5 seeds) than the main protocol (14/10) used everywhere else. This should be validated at full scale, or at minimum flagged as provisional, before being used to justify dropping a whole sampling method from subsequent analysis.
5. A genuine practical limitation is underweighted in the recommendation. RNG's 27x runtime penalty over hexagonal at large grid sizes (Section 6.3) is reduced to a brief caveat in the Discussion's general recommendation to use RNG (Section 8.5); given the size of the effect, this deserves a clearer statement of where the recommendation breaks down.

**Requested Changes:**

Critical (needed for acceptance):
a) Move the direct MST-versus-RNG comparison (currently Figs. S4-S7) into the main text, since it underlies the paper's central topology-ordering claim.
b) Reconcile the apparent contradiction between Section 6.2 ("broadly consistent scaling... regardless of topology") and Section 6.3 (8.19x/27.07x runtime differences at large grid sizes). Either qualify the 6.2 language to make clear it refers to scaling-efficiency shape rather than absolute runtime, or bring the S11 hexagonal/MST panels into the main text alongside Fig. 11.
c) Add at least a discussion-level comparison against Somoclu and/or GigaSOM, or provide an explicit justification for why they are not empirically benchmarked, given that the introduction's systems-gap argument is framed explicitly around these tools.
d) State a position on multiple-comparisons correction for the large number of paired significance tests reported (14 datasets x 3 metrics x multiple comparisons), or clarify explicitly that per-dataset stars are intended as descriptive/exploratory rather than primary confirmatory evidence.
Strengthening (not critical, but would improve the work):
e) Address the decoupling between the in-memory batch path used for the quality benchmarks (Figs. 3-9) and the distributed path used for the scaling benchmarks (Figs. 10-13); the joint "better QE and better scalability" claim is never demonstrated on a single execution path, and this should at least be discussed as a limitation.
f) Validate the HDSSSOM elimination (Section 5.2) at the full 14-dataset/10-seed protocol used elsewhere, or explicitly flag the pilot-based elimination (10 datasets/5 seeds) as provisional.
g) Give RNG's 27x runtime penalty over hexagonal at large grid sizes (Section 6.3) more weight in the Discussion's recommendation (Section 8.5) rather than treating it as a brief caveat, since the effect size is large enough to change the practical recommendation for grid-size-dominated workloads.
h) Presentation: Sections 5.3-5.4 and especially Discussion Section 8 (8.1-8.5) fragment an otherwise continuous, well-sequenced argument (sampling -> topology -> tuning -> stability -> systems limits) into short subsections of 2-3 sentences each. Consolidating into flowing prose with topic-sentence transitions, particularly in the Discussion, would let the paper's coherent narrative come through more clearly.
i) Appendix redundancy: several supplementary figures (S5-S7, S8-S10) largely re-confirm robustness (top-k sensitivity, per-topology tuning benefit) without adding distinct conclusions, and could be compressed into single multi-panel summaries or a single in-text sentence pointing to a consolidated figure. This would help offset the length added by moving the load-bearing results identified above (a, b) into the main text.

---

> ### Author Response · Authors · 2026-07-21
>
> We thank the reviewer for the careful and constructive assessment. We agree with the central framing: FloatSOM is primarily a systems and tooling contribution, and the manuscript must keep the scope of its claims aligned with the evidence shown in the main text. In revision, we have focused on making the topology, scaling, and statistical claims more explicit, and on separating evidence that was directly demonstrated from evidence that should be treated as supporting context or a limitation.
>
> ## 1. Direct MST-versus-RNG evidence in the main text
>
> ### Reviewer Comment
>
> > The paper's principal empirical conclusion is that RNG is the strongest topology, yet the main text never directly establishes that ordering. Section 5.3 compares MST and RNG independently against the hexagonal baseline, but the direct MST-versus-RNG comparison appears only in the supplementary material (Figs. S4-S7). Because the Discussion ultimately recommends RNG over the other topology options, this comparison is central to the paper's argument and should be presented in the main text rather than relegated to the appendix.
>
> ### Response
>
> We agree that the direct MST-versus-RNG comparison is necessary to support our topology recommendation, and we have promoted it from the supplement to the main topology results as Fig. 8. In the matched full-sampling Optuna top-$k$ analysis, RNG achieves significantly lower balanced and train $QE$ than MST, while no significant holdout-$QE$ difference is detected. The hyperparameter analysis also gives RNG the lowest overall stability score across seeds and datasets.
>
> Taken together, these results support RNG as our quality-first default: it provides the strongest attainable $QE$ performance and the most reproducible tuned settings. MST remains a lower-cost alternative for runtime-constrained workloads, particularly at large grid sizes. We therefore do not claim that RNG dominates every endpoint or every deployment regime.
>
> We also revised the geometric explanation so that the RNG preference is not reduced to graph density alone. MST minimizes neighborhood coupling and permits substantial redistribution along irregular structures, but a tree cannot represent multiple locally appropriate connections in a dense region. RNG permits such connections only when supported by the evolving prototype geometry, whereas the hexagonal lattice imposes a uniform neighbor pattern before learning. The observed $QE$ results are presented as consistent with this account rather than as identifying a unique causal mechanism.
>
> ### Summary of Manuscript Changes
>
> We moved the direct MST-versus-RNG comparison into the main text as Fig. 8. Section 5.3 now reports that RNG has lower balanced and train $QE$ than MST in the matched Optuna top-$k$ analysis, while no holdout-$QE$ difference was detected, providing direct support for the stated topology ordering.
>
> ## 2. Section 6.2 versus Section 6.3 scaling language
>
> ### Reviewer Comment
>
> > Section 6.2 claims "broadly consistent scaling... regardless of topology," citing Figs. 11 and S11, but Fig. 11 itself shows RNG only -- the hexagonal/MST curves substantiating this sit in the supplement. More importantly, "regardless of topology" could be read as runtime parity, yet Section 6.3 reports MST and RNG running 8.19x and 27.07x slower than hexagonal at large grid sizes. If Section 6.2 means only that scaling-efficiency shape is similar across topologies (not absolute runtime), that should be stated explicitly, since as written the two sections read as contradictory.
>
> ### Response
>
> We agree and have qualified Section 6.2 to distinguish dimension and sample scaling from grid-size scaling. Across the dimension and sample workloads, the topologies show similar runtime and GPU-efficiency profiles, whereas increasing the number of SOM nodes produces large topology-dependent runtime differences. This makes explicit that the dimension- and sample-scaling patterns and the grid-size runtime penalties are compatible findings.
>
> ### Summary of Manuscript Changes
>
> We revised the scaling language to distinguish similarity in scaling behavior from parity in absolute runtime. Section 6.2 now points to the topology-specific curves and reports that the maximum pairwise runtime spread remained 4.70% at 5,000 dimensions and 3.26% at one billion samples, while Section 6.3 separately presents the much larger topology-dependent costs that arise as grid size increases.

---

> > ### Author Response · Authors · 2026-07-21
> >
> > ## 3. Somoclu and GigaSOM comparison
> >
> > ### Reviewer Comment
> >
> > > No system named as closest related work (Somoclu, GigaSOM) is empirically compared against; only single-GPU XPySOM is benchmarked, despite the introduction's "systems gap" argument being framed explicitly around these tools.
> >
> > ### Response
> >
> > We agree that the baseline selection required clearer justification. We expanded the related-work discussion, attempted an aweSOM benchmark, and now explain why XPySOM was retained as the executable comparator. Somoclu was not benchmarked against because Mancini et al. (2020) directly compared both Somoclu and SomocluGPU with XPySOM and found them to be more than 10 times slower in the reported benchmark. As XPySOM most closely matches FloatSOM's Python/GPU batch-training pathway, we retained it as the controlled executable comparator. Regarding GigaSOM, differences in execution model, language, hardware, and published benchmark design likewise prevent a controlled direct comparison in the present study. The manuscript now states these limitations explicitly.
> >
> > ### Summary of Manuscript Changes
> >
> > We expanded the related-work and Discussion sections to explain the baseline selection. XPySOM was retained as the controlled executable comparator because it most closely matches FloatSOM's Python/GPU batch pathway. We also now note that XPySOM has already been compared directly with Somoclu and SomocluGPU with favourable findings to XPySOM. Finally, we note the GigaSOM differences in execution model, language, hardware, training regime, and workload design preventing a fair direct comparison with FloatSOM.
> >
> > ## 4. Multiple-comparison correction
> >
> > ### Reviewer Comment
> >
> > > With a large number of paired significance tests reported (14 datasets x 3 metrics x multiple comparisons), there is no correction for multiple comparisons and no statement of whether per-dataset significance stars are intended as primary evidence or as descriptive/exploratory annotation.
> >
> > ### Response
> >
> > We agree that the statistical reporting should state the correction policy. The revised manuscript now reports Benjamini-Hochberg adjusted q-values for dataset-level families of related paired tests, while retaining raw p-values for audit. For the topology comparisons, the q-values are computed over exactly the family raised by the reviewer: 42 non-global dataset-level tests per topology comparison, corresponding to 14 datasets across $QE_B$, $QE_H$, and $QE_T$, separately for hexagonal-versus-MST and hexagonal-versus-RNG. Dataset-level figure significance markers and dataset-level significance counts use the adjusted q-values. Global pooled rows are reported separately as overall summaries and are not included in the dataset-level adjustment family.
> >
> > The added radius-control analysis applies a separate prespecified correction across 54 within-topology radius--metric tests. Its post hoc cross-topology analysis applies Benjamini--Hochberg correction across all 21 topology-pair--radius $QE_B$ contrasts, reported with estimates and confidence intervals in Supplementary Table S15.
> >
> > ### Summary of Manuscript Changes
> >
> > We now report Benjamini--Hochberg adjusted q-values alongside raw p-values for the dataset-level tests. Each topology contrast uses a separate family of 42 tests (14 datasets across balanced, holdout, and train $QE$), and the adjusted q-values determine dataset-level figure markers and significance counts. Pooled overall tests are reported separately.

---

> > > ### Author Response · Authors · 2026-07-21
> > >
> > > ## 5. In-memory quality path versus distributed scaling path
> > >
> > > ### Reviewer Comment
> > >
> > > > The quality benchmarks (Figs. 3-9) run on the in-memory batch path, while the scaling benchmarks (Figs. 10-13) run on the distributed path; the joint claim of "better QE and better scalability" is never shown on the same execution path, and this decoupling is not discussed.
> > >
> > > ### Response
> > >
> > > We agree that the submitted version did not make this execution-path distinction prominent enough. The revised Methods now explicitly state that the Optuna quality runs use the standard in-memory batch path, whereas the speed-scaling results use the Ray-orchestrated distributed execution layer. We also revised the deployment comparison language so that Fig. 15 is described as an integrated deployment comparison rather than a pure topology-only attribution.
> > >
> > > To test whether the diagnostic conclusions depended on the execution path, we added a matched benchmark comparing local CuPy with Ray streaming under identical datasets, seeds, topologies, and fixed tuned configurations. No comparison reached statistical significance before or after Benjamini-Hochberg correction, and the estimated differences were small (Supplementary Table S14). We report this diagnostic after the tuning and stability results and before the Ray-pathway performance benchmarks.
> > >
> > > ### Summary of Manuscript Changes
> > >
> > > We now identify the execution pathway used by each experiment: the quality and Optuna analyses use the in-memory batch path, whereas the scaling and XPySOM runtime experiments use Ray distributed execution. We added a matched concordance analysis across 14 datasets, 10 seeds, three topologies, and identical tuned configurations; the small Ray-minus-local differences were not significant before or after correction. Section 5.5 and Supplementary Table S14 report the protocol, paired differences, confidence intervals, and adjusted q-values, while Section 7 separately retains the caveat that the deployment comparison combines implementation, tuning, and topology effects.
> > >
> > > ## 6. HDSSSOM pilot scope
> > >
> > > ### Reviewer Comment
> > >
> > > > HDSSSOM is eliminated from the rest of the study (Section 5.2) using a smaller pilot (10 datasets/5 seeds) than the main protocol (14/10) used everywhere else. This should be validated at full scale, or at minimum flagged as provisional, before being used to justify dropping a whole sampling method from subsequent analysis.
> > >
> > > ### Response
> > >
> > > We agree that the smaller protocol should not be presented as a full-scale elimination study. The revised manuscript frames HDSSSOM as a focused screening pilot and explicitly states that the result should be interpreted under the stated pilot configuration rather than as a comprehensive evaluation of all possible HDSSSOM schedules.
> > >
> > > ### Summary of Manuscript Changes
> > >
> > > We strengthened the qualification in Section 5.2 so that HDSSSOM is described as a focused pilot screen under the tested smaller configuration, rather than a full-scale elimination study or a comprehensive assessment of all possible schedules.
> > >
> > > ## 7. RNG runtime penalty in the practical recommendation
> > >
> > > ### Reviewer Comment
> > >
> > > > RNG's 27x runtime penalty over hexagonal at large grid sizes (Section 6.3) is reduced to a brief caveat in the Discussion's general recommendation to use RNG (Section 8.5); given the size of the effect, this deserves a clearer statement of where the recommendation breaks down.
> > >
> > > ### Response
> > >
> > > We agree that the RNG recommendation must be explicitly conditional on runtime budget and grid size. The revised Discussion identifies RNG as the default topology for most datasets, while qualifying this recommendation for very large grids. It identifies MST as the preferable graph alternative when RNG's topology overhead is prohibitive and reports the measured hexagonal, MST, and RNG runtimes at the largest tested grid size.
> > >
> > > The geometric rationale makes that conditional recommendation more precise: RNG retains more geometry-supported local connections than MST, which may benefit concentrated regions, but the same additional topology work contributes to its higher grid-size cost. Thus MST's greater freedom and lower graph cost make it the practical compromise when RNG's mesh-like local connectivity is not worth the runtime penalty.
> > >
> > > ### Summary of Manuscript Changes
> > >
> > > We made the RNG recommendation conditional on runtime budget and grid size. The Discussion now gives the measured 8-GPU runtimes at grid size 64—32.54 s for hexagonal, 266.45 s for MST, and 880.83 s for RNG—and recommends MST when RNG's graph-construction and path-calculation overhead is prohibitive on large grids. The geometric discussion also connects RNG's additional local connectivity to both its potential quality benefit and its higher cost.

---

> > > > ### Author Response · Authors · 2026-07-21
> > > >
> > > > ## 8. Discussion structure
> > > >
> > > > ### Reviewer Comment
> > > >
> > > > > Sections 5.3-5.4 and especially Discussion Section 8 (8.1-8.5) fragment an otherwise continuous, well-sequenced argument into short subsections of 2-3 sentences each. Consolidating into flowing prose with topic-sentence transitions, particularly in the Discussion, would let the paper's coherent narrative come through more clearly.
> > > >
> > > > ### Response
> > > >
> > > > We agree that the Discussion should read as a continuous argument rather than a list of short independent subsections. We consolidated it into a continuous sequence covering sampling, topology, tuning, stability, and systems limits, while removing repeated interpretation already given in the Results.
> > > >
> > > > ### Summary of Manuscript Changes
> > > >
> > > > We consolidated the former short Discussion subsections into a continuous sequence of topic-led paragraphs covering sampling, topology, tuning, stability, and systems limitations. Repeated Results interpretation was removed, the execution-path validation was placed in Section 5.5 before the scaling results, and the topology discussion now progresses from fixed-lattice coupling through MST and RNG before turning to the measured runtime trade-offs.
> > > >
> > > > ## 9. Appendix redundancy
> > > >
> > > > ### Reviewer Comment
> > > >
> > > > > Several supplementary figures (S5-S7, S8-S10) largely re-confirm robustness without adding distinct conclusions, and could be compressed into single multi-panel summaries or a single in-text sentence pointing to a consolidated figure. This would help offset the length added by moving the load-bearing results identified above into the main text.
> > > >
> > > > ### Response
> > > >
> > > > We agree that the supplement should prioritize results that materially support the main claims. In revision, we compressed the redundant robustness material into two consolidated supplementary figures: Supplementary Fig. S4 for the topology top-$k$ sensitivity analyses and Supplementary Fig. S5 for the topology-stratified tuned-versus-untuned comparisons. We also moved the direct MST-versus-RNG comparison into the main text as Fig. 8, so the central topology-ordering evidence is no longer relegated to the appendix.
> > > >
> > > > ### Summary of Manuscript Changes
> > > >
> > > > We consolidated the redundant supplementary robustness plots into two multi-panel figures: Supplementary Fig. S4 for topology top-$k$ sensitivity and Supplementary Fig. S5 for topology-stratified tuned-versus-untuned comparisons. The direct MST-versus-RNG comparison was promoted to main-text Fig. 8, preserving the supplementary audit trail while removing repeated figure-level interpretation.

---

### Review · Reviewer_aZCA · 2026-07-14

**Summary Of Contributions:**

Summary and Contributions:
The paper introduces FloatSOM, a framework designed to scale Self-Organizing Maps to massive datasets. The authors address memory limits by implementing multi-GPU distributed training and out-of-core disk streaming. Additionally, they replace traditional fixed lattices like hexagonal grids with dynamic graph-based topologies, specifically Minimum Spanning Trees and Relative Neighborhood Graphs. The authors conduct extensive empirical evaluations across 14 datasets using automated hyperparameter optimization. Their results demonstrate that FloatSOM achieves lower quantization error and scales efficiently to extremely large workloads, such as processing one billion samples on eight GPUs.

Key Strengths:
The engineering effort is highly commendable. The experimental design is rigorous, featuring extensive hyperparameter tuning and fair baseline comparisons against existing libraries. The paper provides highly actionable insights regarding sampling trade-offs and topology selection for real-world deployments.

Key Weaknesses:
The manuscript leans heavily on empirical results and lacks a deeper theoretical discussion explaining the geometric reasons behind the superior performance of graph-based topologies on specific data manifolds. Furthermore, the computational overhead for complex topologies becomes a bottleneck at very large grid sizes.

**Additional Comments:**

None

**Audience:**

Yes

**Audience Explanation:**

The TMLR audience includes many researchers focused on large-scale machine learning systems and unsupervised learning algorithms. Self-Organizing Maps remain valuable for clustering and high-dimensional data visualization. Demonstrating how to modernize this classic algorithm using contemporary GPU orchestration techniques to handle billion-scale datasets is highly relevant. The intersection of system-level engineering and algorithmic topology improvements aligns perfectly with the journal scope regarding practical learning methods and their empirical validation.

**Broader Impact Concerns:**

This research focuses on foundational unsupervised learning algorithms and computational scaling frameworks. The work does not involve sensitive personal data collection or target applications prone to ethical controversies such as surveillance or biased decision-making. Consequently, I do not foresee any direct negative societal impacts or ethical concerns arising from this methodology. A dedicated broader impact statement is not strictly necessary for this specific contribution.

**Claims And Evidence:**

Yes

**Claims Explanation:**

The claims are exceptionally well supported. The authors evaluate their framework across 14 diverse datasets and perform over 160,000 optimization trials. They use rigorous statistical methods, including paired t-tests and confidence intervals, to validate their findings. The scalability claims are backed by clear runtime and efficiency scaling curves demonstrating successful execution on massive workloads. Furthermore, the authors establish a fair baseline by first proving their implementation matches the performance of an existing library under default settings before demonstrating their proposed improvements.

**Requested Changes:**

I recommend a few adjustments to strengthen the manuscript. First, I suggest expanding the discussion section to include a deeper geometric intuition regarding why the Relative Neighborhood Graph topology outperforms fixed grids on real-world datasets. Providing visual examples of how these dynamic graphs adapt to complex low-dimensional manifolds would be highly beneficial. Second, the authors should provide more quantitative details regarding the memory and communication overhead required to synchronize complex topology states across multiple GPUs, especially for large grid sizes. This would help clarify the sharp runtime increases observed in the grid-size scaling experiments. Finally, improving the legibility of the forest plots is necessary, as the data points and error bars are currently difficult to read without zooming in significantly.

---

> ### Author Response · Authors · 2026-07-21
>
> We sincerely thank the reviewer for the careful, constructive, and encouraging assessment of our work. We especially appreciate the reviewer highlighting the engineering contribution, the rigor of the empirical evaluation, and the practical relevance of modernizing SOM training for billion-scale workloads. The requested changes were thoughtful and helped us improve both the explanatory depth and the presentation of the manuscript. In response, we have expanded the systems accounting, strengthened the geometric explanation of the topology choices, added a real high-dimensional example to the representative-topology figure, and improved the legibility and auditability of the quantitative results.
>
> ## 1. Geometric intuition and expanded Figure 5
>
> ### Reviewer Comment
>
> > First, I suggest expanding the discussion section to include a deeper geometric intuition regarding why the Relative Neighborhood Graph topology outperforms fixed grids on real-world datasets. Providing visual examples of how these dynamic graphs adapt to complex low-dimensional manifolds would be highly beneficial.
>
> ### Response
>
> We agree that the topology motivation benefits from a more concrete geometric account and from an example beyond a two-dimensional synthetic dataset.
>
> A fixed hexagonal lattice assigns neighborhood relationships before learning and independently of the evolving prototype geometry. Movement of one prototype can therefore influence lattice neighbors that are not locally related in learned data space, potentially displacing otherwise useful prototypes and increasing dead nodes. MST minimizes this coupling and gives prototypes greater freedom to redistribute along irregular or elongated structures. That freedom is also MST's limitation: a tree cannot retain several locally appropriate connections where a dense region is better described by a mesh-like neighborhood.
>
> RNG occupies an intermediate position. It is less free than MST because nodes may influence several neighbors, but those additional connections are supported by the evolving prototype geometry rather than uniformly imposed beforehand. RNG can remain sparse where appropriate and form multiple connections in concentrated regions, influencing nodes that should be influenced without the fixed lattice's uniform coupling.
>
> We expanded Fig. 5 from one row to a 2x3 layout. Panels A--C show circles in native 2D, and panels D--F show KDD Cup 99 trained on standardized 41-dimensional data and displayed using a shared two-dimensional PCA projection. In the KDD Cup 99 display, the hexagonal topology contains several apparently nonlocal connections spanning separated regions of the projected prototype distribution, whereas the MST and RNG connections more closely follow its local geometry. Quantitative node-utilization and dead-node evidence remains in Fig. 9 and the matched diagnostic tables.
>
> ### Summary of Manuscript Changes
>
> We expanded Fig. 5 to a 2x3 comparison of matched hexagonal, MST, and RNG maps on both the native two-dimensional circles dataset and standardized 41-dimensional KDD Cup 99 data shown through a shared PCA projection. The Results now describe the apparently nonlocal fixed-lattice connections visible in the KDD projection, with an explicit warning that PCA can distort the original geometry. The Methods distinguish fixed regular-lattice adjacency from graph neighborhoods refreshed from evolving prototypes, and the Discussion develops the corresponding geometric interpretation: MST reduces predetermined coupling but is restricted to a tree, whereas RNG can retain several geometry-supported local connections.

---

> > ### Author Response · Authors · 2026-07-21
> >
> > ## 2. Distributed memory, communication, and large-grid cost
> >
> > ### Reviewer Comment
> >
> > > Second, the authors should provide more quantitative details regarding the memory and communication overhead required to synchronize complex topology states across multiple GPUs, especially for large grid sizes. This would help clarify the sharp runtime increases observed in the grid-size scaling experiments.
> >
> > ### Response
> >
> > We thank the reviewer for identifying where the original systems explanation needed more quantitative detail. FloatSOM does not communicate observations between workers during an iteration. Each worker processes its own shard in bounded chunks and accumulates a node-by-feature update numerator and a node-wise normalization denominator. Those two arrays are synchronized once per iteration by NCCL all-reduce; the updated prototypes then remain resident and identical on every worker.
> >
> > For $P$ nodes and $d$ features stored as float32, the two accumulators contain $Pd+P$ values. Their logical all-reduce payload is therefore $4P(d+1)$ bytes per worker per iteration, plus a 4-byte sample-count reduction. For a ring all-reduce over $G$ workers, each worker sends and receives approximately $2(G-1)/G$ times that payload. This communication term is independent of the number of observations $N$: increasing $N$ increases local BMU/update work and data staging, whereas increasing $P$ increases both the synchronized update and the graph-topology work.
> >
> > The memory explanation now separates the source of the cost from the mechanism used to manage it. With chunks of at most $C$ rows, the dominant bounded worker arrays scale as $O(Cd+Pd)$ rather than requiring the full $O(Nd)$ observations in GPU memory. Graph-distance and influence structures add both computation and storage terms that scale with $P^2$; for MST and especially RNG, topology construction, all-pairs path calculation, and influence-cache updates therefore become more expensive as node count increases. FloatSOM tiles these structures and can spill them from VRAM to system RAM when necessary. Spilling is a memory-management response that can add transfer overhead, but it is not the sole explanation for the runtime increase. The measured grid-size results show the combined practical consequence. At grid size 64 on 8 GPUs, hexagonal, MST, and RNG required 32.54, 266.45, and 880.83 s, respectively. Thus, the distributed pathway scales much more favorably in sample count than in node count, especially for RNG.
> >
> > To further examine the RNG runtime spike in grid-size scaling, we compared the topology-construction operations. For each of the $\binom{P}{2}$ candidate node pairs, RNG checks every possible third node in its relative-neighborhood blocker test, giving up to $\binom{P}{2}(P-2)=O(P^3)$ pair--blocker comparisons per topology refresh. At $P=4096$, this is approximately 34.3 billion unordered pair--blocker checks. MST instead constructs and sorts approximately $\binom{4096}{2}=8.39$ million candidate edges using Kruskal's algorithm, an $O(P^2\log P)$ edge-construction step after pairwise distances are calculated. Both topologies then perform the shared graph-distance and influence calculations. The additional $O(P^3)$ RNG blocker-test work therefore provides a quantitative algorithmic explanation for the sharp RNG cost increase at large grid sizes.
> >
> > ### Summary of Manuscript Changes
> >
> > We added communication and memory accounting to Section 3.3.1. For $P$ nodes, $d$ float32 features, and $G$ workers, the manuscript now gives the per-iteration logical all-reduce payload as $4P(d+1)$ bytes per worker plus the sample-count reduction, and the ring all-reduce traffic factor as approximately $2(G-1)/G$. It also distinguishes bounded worker storage, $O(Cd+Pd)$ for chunk size $C$, from the $P^2$ graph-distance and influence structures and explains tiling and RAM spill behavior.
> >
> > Section 6.3 now reports the grid-size-64 runtimes on 8 GPUs—32.54 s for hexagonal, 266.45 s for MST, and 880.83 s for RNG—and quantifies the construction cost: at $P=4096$, RNG may perform about 34.3 billion pair--blocker checks, compared with approximately 8.39 million candidate MST edges before the shared graph-distance calculations. The Discussion uses these controlled internal measurements to qualify the topology recommendation and treats Somoclu and GigaSOM only as non-comparable published systems context.
> >
> > ## 3. Forest-plot legibility
> >
> > ### Reviewer Comment
> >
> > > Finally, improving the legibility of the forest plots is necessary, as the data points and error bars are currently difficult to read without zooming in significantly.
> >
> > ### Response
> >
> > We thank the reviewer for drawing our attention to this presentation issue. We increased the confidence-interval line widths and point-marker sizes by 200% across all forest plots to improve legibility at manuscript scale.